# Hadronic Light-by-Light Corrections to the Muon Anomalous Magnetic Moment

**Daniel Melo** [1,*] , **Edilson Reyes** [2] **and Raffaele Fazio** [1]

1 Departamento de Física, Universidad Nacional de Colombia, Bogotá 111321, Colombia; arfazio@unal.edu.co
2 Departamento de Física, Universidad de Pamplona, Pamplona 543050, Colombia; eareyesro@unal.edu.co
* Correspondence: dgmelop@unal.edu.co

**Abstract:** We review the hadronic light-by-light (HLbL) contribution to the muon anomalous magnetic moment. Upcoming measurements will reduce the experimental uncertainty of this observable by a factor of four; therefore, the theoretical precision must improve accordingly to fully harness such an experimental breakthrough. With regards to the HLbL contribution, this implies a study of the high-energy intermediate states that are neglected in dispersive estimates. We focus on the maximally symmetric high-energy regime and in-quark loop approximation of perturbation theory, following the method of the OPE with background fields proposed by Bijnens et al. in 2019 and 2020. We confirm their results regarding the contributions to the muon $g - 2$. For this, we use an alternative computational method based on a reduction in the full quark loop amplitude, instead of projecting on a supposedly complete system of tensor structures motivated by first principles. Concerning scalar coefficients, mass corrections have been obtained by hypergeometric representations of Mellin–Barnes integrals. By our technique, the completeness of such kinematic singularity/zero-free tensor decomposition of the HLbL amplitude is explicitly checked.

**Keywords:** magnetic moment of the muon; hadronic light-by-light scattering; Mellin–Barnes; hypergeometric series; multivariate residues

## 1. Introduction

The Standard Model (SM) is the current theoretical paradigm for particle physics at its most fundamental level. This fact is rooted in the SM's mathematical consistency and especially in its highly accurate predictions for precision experiments. In fact, one of the most precisely verified theoretical predictions in the history of physics and the true triumph of quantum field theory is the SM magnetic moment of the electron [1,2] $\vec{\mu} = g\left(\frac{e}{2m}\right)\vec{S}$, with $m$ being the electron mass and $\vec{S}$ its spin operator. The so-called anomalous part is expressed by the quantity $a = \frac{g-2}{2}$, quantifying the deviation of the Landé factor from the classical value $g = 2$, and is entirely due to quantum-mechanical phenomena; the "cloud" of virtual particles with which the electron is constantly interacting slightly changes the way it interacts with a classical magnetic field (see Figure 1). Therefore, the measurement of the anomalous part of a particle's magnetic moment makes possible to test which kinds of other particles it interacts with and the strength of the interaction. Consequently, this quantity is of the utmost interest for theoretical physicists when testing the SM itself, and also theories referred to as Beyond the Standard Model (BSM). For the electron, this anomalous part has been computed to $O(\alpha^5)$ in QED, for weak contributions the uncertainty is $\sim 10^{-16}$, and for hadronic contributions it is $\sim 10^{-14}$ [2]. The discrepancy of the SM prediction with the latest and most precise measurement is either $10.2(2.7) \times 10^{-13}$ [3] or $-3.4(1.6) \times 10^{-13}$ [4] (The numbers we cite here were obtained only considering the five-loop QED estimate from [5], but an alternate computation of such contribution can be found in [6]) depending on whether the value of the fine structure

constant (necessary input for the QED contribution) is taken from measurements of the recoil velocity of caesium or rubidium atoms, respectively, when they absorb a photon.

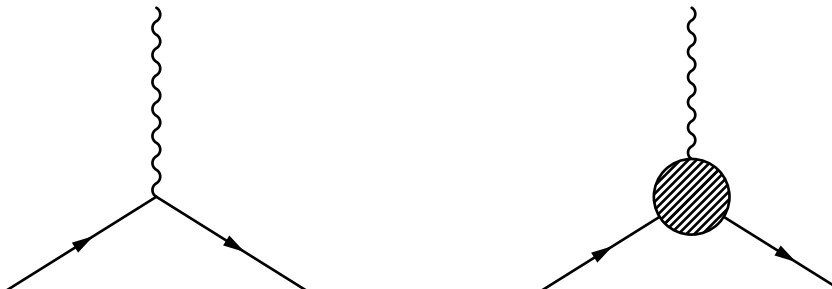

**Figure 1.** Interaction of a fermion with a classical electromagnetic field at tree level (**left**) vs. corrections due to virtual particles (**right**).

For the muon, the tension between the SM theoretical prediction for the anomalous magnetic moment $a_\mu$ and its experimental measurement is bigger. Therefore, it has attracted much attention since the Brookhaven National Laboratory (BNL) experiment results shed light on the issue in 2004 [7]. Taking into account the latest results from Fermilab (FNAL) [8,9] in addition to the BNL ones, the experimental value $a_\mu^{\text{exp}}$ is:

$$a_\mu^{\text{BNL}} = 116\ 592\ 089(63) \times 10^{-11}\ , \tag{1}$$

$$a_\mu^{\text{Fermilab}} = 116\ 592\ 055(24) \times 10^{-11}\ , \tag{2}$$

$$a_\mu^{\text{exp}} = 116\ 592\ 059(22) \times 10^{-11}\ . \tag{3}$$

As usual, the combination of the two measurements is obtained from the principle of maximum likelihood, which is mathematically realized by the method of least squares due to the Gaussian probability distribution and provides the above weighted average.

In contrast, the most recent consensus SM prediction $a_\mu^{\text{SM}}$ is:

$$a_\mu^{\text{SM}} = 116\ 591\ 810(43) \times 10^{-11} \tag{4}$$

which has been obtained by the "Muon $g-2$ Theory Initiative" and is described in [10]. Consequently, the tension between the SM value and the measurement is:

$$\Delta a_\mu = a_\mu^{\text{exp}} - a_\mu^{\text{SM}} = 249(48) \times 10^{-11} \text{ or } 5.2\ \sigma\ . \tag{5}$$

Although this discrepancy is already beyond the five-sigma discovery threshold, it is still not considered to be a sign of New Physics because there are unresolved inconsistencies between the dispersive and lattice estimations of the Hadronic Vacuum Polarization (HVP) contribution (see Figure 2), which we will describe below.

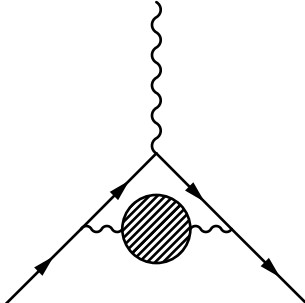

**Figure 2.** Hadronic vacuum polarization contribution to the anomalous magnetic moment of the muon. The blob contains only strongly interacting virtual particles.

From the three fundamental interactions considered in the SM, only the strong force contribution currently has an uncertainty that is relevant with respect to the tension's value [10]. Therefore, these strong contributions to $a_\mu$ are the main focus of the theoretical work towards reducing uncertainty. Hadronic contributions affect $a_\mu$ in two ways: HVP and Hadronic Light-by-Light Scattering (HLbL).

The HVP contribution is much larger than that of the HLbL, and moreover, it can be computed from experimental measurements in the well-known approach of dispersive integrals [10]. More specifically, the contribution from HVP amounts to $6845(40) \times 10^{-11}$, and from HLBL, it is $92(18) \times 10^{-11}$. HVP is essentially a hadronic correction to the photon propagator; then, because of analyticity of Green functions and the unitarity of the theory, it can be computed from the cross-section of a virtual photon decaying into hadrons. This cross-section can be extracted from $e^+e^- \longrightarrow Hadrons$ data from detectors such as BaBar [11], KLOE [12], BESIII [13], SND [14], and CMD-3 [15]. By the nature of the dispersive method, it is necessary to know the hadron production cross-section at different values of the center of mass energy. This can be achieved either by directly changing the energy of the $e^-$ and $e^+$ beams, called " direct scan" [16,17], or by fixing it and letting the (measured) initial-state radiation do the work of varying the energy of the virtual photon, which then decays into hadrons, called "radiative return" [18,19]. There are also alternative methods of measuring HVP by $\tau$ decay experiments [20] and by measuring the hadronic contribution to the running of the fine structure constant $\alpha = e^2/4\pi$ from $\mu^-e^-$ elastic cross-sections, called the MUonE project [21–23].

Before 2021, the main goals of the community for the HVP contribution were to improve the uncertainty on dispersive estimates by solving tensions between datasets, improving accountability of radiative corrections to the measured cross-sections, and considering contributions from further channels. Even though these goals still remain, the publication of the HVP contribution estimate by the BMW lattice collaboration [24] upended the priorities. This result, the first of its kind to have competitive uncertainties with respect to the dispersive estimates, reduced the tension with experiment to $1.5\,\sigma$ when considered alone. However, this prediction is in tension with dispersive estimates, and therefore, further analysis is required. Although the full HVP contribution has not been reproduced with a similar level of precision, several other lattice collaborations [25–29] have published results of a benchmark quantity, usually called intermediate Euclidean window, which aims to separate the intermediate distance part of the HVP contribution. It is less influenced both by effects due to the finite size of the lattice (large distance) and the non-zero size of lattice spacings (short distance). This quantity has allowed for more clear comparisons between dispersive and lattice estimates [30–32], which have so far resulted in confirmations of the phenomenology-lattice tension. The most recent results from the CMD-3 experiment for $\pi\pi$ production show a significant deviation from all other previous experimental results, including CMD-2 [33]. Much like recent lattice HVP results, dispersive estimates of HVP with $\pi\pi$ contributions coming only from this new set of measurements also reduce the tension between the SM prediction and $a_\mu^{\text{exp}}$ [32], which has further added to the confusion. Consistency checks from analyticity and unitarity constraints on the pion vector form factor have not shed much light on the discrepancies between experiments [32].

Now we go on to the HLbL scattering, which gives a contribution (see Figure 3) of $92(18) \times 10^{-11}$ [10]. In contrast with HVP, the theoretical side of the HLbL scattering computation had been much less understood until recently. The added complexity is due to the fact that four currents are involved, instead of only two. This introduces several difficulties. First of all, HLbL scattering cannot be as cleanly related to $e^+e^-$ annihilation or other experiments. Furthermore, the HLbL amplitude has a much more complex tensor decomposition; in four space-time dimensions, it is a linear combination of 41 tensors, even after gauge invariance constraints have been considered. Moreover, it is necessary to expand this set to a redundant one with 54 elements in order to avoid kinematic, meaning spurious or in general non-dynamical, singularities, that spoil the dispersive

approach. In the end, for the purpose of computing $a_\mu^{\mathrm{HLbL}}$, it is only necessary to know seven of these scalar coefficients, since the rest are related to them by crossing symmetry of Mandelstam's variables. Meanwhile, for HVP, one initially has two tensor structures, which are then reduced to one due to gauge invariance. In fact, this dispersion-fit tensor decomposition for HLbL was only recently found for the first time [34,35]. This multiplicity of scalar coefficients makes the dispersive approach much more complex for HLbL than it is for HVP, because each coefficient requires its own dispersive integral. In spite of this, contributions of intermediate states including pseudoscalar poles, box topologies, and rescattering diagrams [10] have been successfully computed. Particular applications with pions can be found in [35–37]. Before these breakthroughs with the dispersive method, the low energy regime of the HLbL scattering was studied mostly with hadronic models, whose uncertainty was harder to assess. In fact, the dispersive treatment of scalar and axial contributions is not yet in a satisfactory state despite recent progress made in that direction [38,39]; hence, models still play a role in current estimations [40–43]. Compared to HVP, the HLbL contribution appears at one further order of $\alpha$ than the former, and thus, its computation requires correspondingly less accuracy. Finally, a common feature for both HLbL and HVP is that they are dominated by very different degrees of freedom at low and high energies, namely, hadrons and then quarks and gluons, respectively. The fact that HVP and HLbL amplitudes enter the muon vertex as an insertion of one and two loops, respectively, makes it necessary to properly "sew" the contributions from different approaches at different kinematic regions.

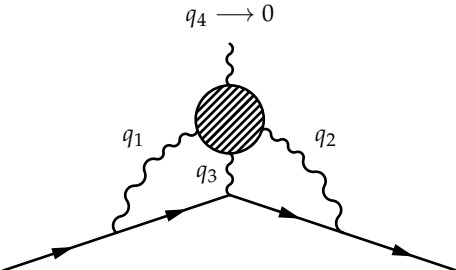

**Figure 3.** Hadronic light-by-light scattering contribution to the anomalous magnetic moment of the muon. The blob contains only strongly interacting virtual particles.

Much like with HVP, lattice estimations for HLbL have become competitive with dispersive ones in recent years [44–50]. Even though the differences between the two are within uncertainty values, lattice estimates also push the SM prediction towards the experimental value.

As mentioned previously, the HLbL amplitude at low photon virtualities (see Figure 3) has been obtained from low-energy QCD models (scalar QED for the pion, Nambu Jona-Lasinio model, and vector meson dominance, for example) [51,52] or, more recently, from dispersive integrals on hadronic production from multiple virtual photons [53]. Of course, these methods have a certain high energy limit of validity, be it conceptual or practical. For the dispersive approach, it is the latter case. Extension to heavier intermediate states has been hindered by a lack of data on the necessary subprocesses and the increasing complexity of unitarity diagrams with multiple particles. Fortunately, heavy intermediate state contributions are suppressed in dispersive integrals by a narrower phase space, and thus, one can consider states up to certain mass and still obtain a useful result. Nevertheless, to assess or reduce the uncertainty coming from the neglected heavier states, it is necessary to resort to tools that complement, replace, or evaluate the dispersive approach at high energies. These tools are called *short-distance constraints* (SDCs). For example, in the high-energy regions of dispersive integrals, data for a hadronic form factor can be replaced by the expression for its known asymptotic behavior. One can also study the asymptotic behavior of the HLbL scattering amplitude itself and use it to test how well the set of intermediate states considered in the dispersive approach resembles such behavior.

For a finite number of intermediate states, it is not possible to completely mimic such behavior [54,55], but this fact can be used to measure how well a set of intermediate states represents high energy contributions. Such studies are a key complement of dispersive computations and play a central role in uncertainty assessment [56–58]. There are two loop momenta configurations that lead to a high-energy regime in the HLbL contribution to $a_\mu$: $|q_1^2| \sim |q_2^2| \sim |q_3^2| \gg \Lambda_{QCD}^2$ and $|q_1^2| \sim |q_2^2| \gg \Lambda_{QCD}^2$, where $q_1$, $q_2$, and $q_3$ are the virtual photon momenta, $q_4$ is the real soft photon momentum representing the electromagnetic external field (see Figure 3), and $\Lambda_{QCD}$ is the QCD hadronic regime threshold. The main purpose of this research is to compute the HLbL scattering amplitude by the methods of perturbative QCD in the regime in which the absolute values of the three virtualities, $q_1^2$, $q_2^2$, $q_3^2$, are much larger than the hadronic threshold. In particular, we perform an operator product expansion (OPE) in an electromagnetic background for the HLbL scattering amplitude following [59–61], but we fully harness the background field method to provide an original and, in our view, more systematic framework, to include the hadronic contributions in the same spirit of QCD sum rules. The main result of the work is nevertheless the computation of the quark loop amplitude that constitutes the leading contribution of the HLbL scattering amplitude at high energies. Our computation can be considered an extension of the literature's result, because we obtain and study the full tensor structure of the amplitude and obtain a complete series expansion of light quark mass corrections up to arbitrary order. The computation is implemented using original *Mathematica* scripts in combination with state-of-the-art packages, *FeynCalc* [62–64] and *MBConicHulls* [65], for computations in high-energy physics.

## 2. $a_\mu$ in QFT

In this section, we review the basics of the computation of the HLbL contribution to $a_\mu$, with special focus on the dispersive approach and the Mandelstam representation on which it is based.

### 2.1. Basics

The magnetic moment of a particle is defined through its scattering amplitude on a classical magnetic field. More specifically, for a particle with spin $s$ and magnetic moment $\mu$ interacting with a classical magnetic field $\boldsymbol{B}$, the matrix element of the interaction Hamiltonian $H_{int}$ between an initial state $\psi_{\boldsymbol{p}\sigma}$ with momentum $\boldsymbol{p}$ and spin projection $\sigma$ and final state $\psi_{\boldsymbol{p}'\sigma'}$ is:

$$\langle \psi_{\boldsymbol{p}'\sigma'} | H_{int} | \psi_{\boldsymbol{p}\sigma} \rangle = -\frac{\mu}{s} (\boldsymbol{J}^{(s)})_{\sigma'\sigma} \cdot \boldsymbol{B} \, \delta^3(\boldsymbol{p}' - \boldsymbol{p}) \times 2m \, , \tag{6}$$

where $\delta^3$ represents the Dirac delta in three dimensions, $m$ stands for the particle's mass, and $\boldsymbol{J}^{(s)}$ is the little group generator associated to a massive particle of spin $s$. The factor $2m$ appears only due to the relativistic normalization of the states.

For a relativistic charged particle, the corresponding matrix element is:

$$\langle \psi_{\boldsymbol{p}'\sigma'} | H_{int} | \psi_{\boldsymbol{p}\sigma} \rangle = -e_q j_\mu A^\mu \, , \tag{7}$$

where $A_\mu$ is the classical electromagnetic potential, $j_\mu$ is the matrix element of the particle's current operator, and $e_q$ represents its electric charge. For the muon, we have $s = 1/2$, $e_q = -e$ (i.e. minus the absolute value of the electric charge of the electron) and:

$$j_\mu(x) = e^{i(p-p')x} \langle \mu_{\boldsymbol{p}'\sigma'}^- | J^\mu(0) | \mu_{\boldsymbol{p}\sigma}^- \rangle = e^{i(p-p')x} \overline{u}_{\boldsymbol{p}'\sigma'} \Gamma_\mu(p', p) u_{\boldsymbol{p}\sigma} \, , \tag{8}$$

where $J^\mu$ represents the electromagnetic current Heisenberg operator of the muon, $\Gamma_\mu$ (in QFT) is the amplitude of the full on-shell vertex diagram to the right of Figure 1, and $u_{\boldsymbol{p}s}$ and $\overline{u}_{\boldsymbol{p}'s'}$ are the spinors associated to the incoming and outgoing muon, respectively.

Considering the fact that $j_\mu$ has to behave as a Lorentz vector and it may contain Dirac matrices, then $\Gamma_\mu$ must be a linear combination of the four momenta, the Levi–Civita

symbol $\epsilon^{\mu\nu\lambda\rho}$, and Dirac bilinears, which are listed in the first column of Table 1. The complete set of independent structures that can be built, including index contractions, is written in the second column of Table 1, where the total momentum $P^{\mu} \equiv p^{\mu} + p'^{\mu}$ and the transferred momentum $q^{\mu} \equiv p'^{\mu} - p^{\mu}$ have been used. However, such a set can be greatly reduced using the on-shell character of the spinors in (8) (i.e., Dirac's equation) and gauge invariance.

**Table 1.** A priori available structures in the covariant decomposition of the muon electromagnetic on-shell vertex $\Gamma^{\mu}$ (see Equation (8) and Figure 1). We use $\sigma_{\mu\nu} = \frac{i}{2}[\gamma^{\mu}, \gamma^{\nu}]$.

| Dirac Matrices' Basis Element | Available Structures |
|:---:|:---:|
| 1 | $P^{\mu}, q^{\mu}$ |
| $\gamma^{\mu}$ | $\gamma^{\mu}, q^{\mu}\slashed{q}, q^{\mu}\slashed{P}, P^{\mu}\slashed{q}, P^{\mu}\slashed{P}, \epsilon^{\mu\nu\lambda\rho}\gamma_{\nu}P_{\lambda}q_{\rho}$ |
| $\gamma^5$ | $P^{\mu}\gamma^5, q^{\mu}\gamma^5$ |
| $\gamma^{\mu}\gamma^5$ | $\gamma^{\mu}\gamma^5, q^{\mu}\slashed{q}\gamma^5, q^{\mu}\slashed{P}\gamma^5, P^{\mu}\slashed{q}\gamma^5, P^{\mu}\slashed{P}\gamma^5, \epsilon^{\mu\nu\lambda\rho}\gamma_{\nu}\gamma^5 P_{\lambda}q_{\rho}$ |
| $\sigma_{\mu\nu}$ | $P^{\alpha}q^{\beta}\sigma_{\alpha\beta}P^{\mu}, P^{\alpha}q^{\beta}\sigma_{\alpha\beta}q^{\mu}, \sigma^{\mu\nu}P_{\nu}, \sigma^{\mu\nu}q_{\nu}, \epsilon^{\mu\nu\lambda\rho}\sigma_{\nu\lambda}P_{\rho}, \epsilon^{\mu\nu\lambda\rho}\sigma_{\nu\lambda}q_{\rho}$ |

Indeed, the most general tensor structure of $\Gamma_{\mu}(p', p)$ for on-shell spinors is:

$$\Gamma_{\mu}(p', p) = A_1(q^2)\gamma_{\mu} + P^{\mu}A_2(q^2) + \left(\gamma^{\mu} - \frac{2mq^{\mu}}{q^2}\right)\gamma^5 A_3(q^2) + P^{\mu}\gamma^5 A_4(q^2) . \quad (9)$$

By using Gordon's identity:

$$\overline{u}_{p'\sigma'}\gamma^{\mu}u_{p\sigma} = \frac{1}{2m}\overline{u}_{p'\sigma'}\{P^{\mu} + i\sigma^{\mu\nu}q_{\nu}\}u_{p\sigma} , \quad (10)$$

$$\Gamma_{\mu}(p', p) = F_1(q^2)\gamma_{\mu} + i\sigma^{\mu\nu}\frac{q_{\nu}}{2m}F_2(q^2) + (\gamma^{\mu} - \frac{2mq^{\mu}}{q^2})\gamma^5 F_3(q^2) + \sigma^{\mu\nu}\frac{q_{\nu}}{2m}\gamma^5 F_4(q^2) . \quad (11)$$

This convention is helpful because $\sigma_{\mu\nu} \equiv \frac{i}{2}[\gamma^{\mu}, \gamma^{\nu}]$ is the generator of Lorentz transformations for covariant wave functions of Dirac fermions, and therefore, $\sigma_{ij}$ generates rotations and little group transformations. Thus, the parallel with (6) becomes straightforward. In (11), $F_1$, $F_2$, $F_3$, and $F_4$ are Lorentz invariant coefficients, also called "form factors". The first two are associated to parity-conserving contributions and are also known as electric and magnetic form factor, respectively. On the other hand, $F_3$ and $F_4$ are related to parity-violating and CP-violating contributions, respectively, and are also known as anapole moment and electric dipole moment. For a pure on-shell derivation of the structure of the electromagnetic vertex, also including, after an appropriate analytic continuation of momenta, the photon being on-shell, we refer to [66,67]. Since the anomalous magnetic moment is related to a non-relativistic interaction, it is necessary to evaluate the muon vertex in the limit of zero exchanged momentum, that is, $q \to 0$. In such a limit, we have $F_1(0) = 1$ in order to define $e$ as the physical electric charge measured in the interaction with a classical Coulomb field. We also have $F_3(0) = 0$ in this limit. On the other hand, $F_2(0)$ and $F_4(0)$ are not constrained. Thus, in the limit of zero exchanged momentum and a slowly varying magnetic field, (7) and (8) become:

$$\langle\psi_{p'\sigma'}|H_{int}|\psi_{p\sigma}\rangle = 2m \times \frac{e}{m}(1 + F_2(0))(\boldsymbol{J}^{(1/2)})_{\sigma'\sigma} \cdot \boldsymbol{B} \, \delta^3(\boldsymbol{p'} - \boldsymbol{p}) , \quad (12)$$

which implies:

$$\mu = \frac{e}{2m}(1 + F_2(0)) . \quad (13)$$

At tree level, we have $\Gamma_{\mu} = \gamma_{\mu}$, and thus, $\mu = \frac{eq}{2m}$, which agrees with Pauli's equation and Dirac's equation in the non-relativistic limit. Then, quantum corrections to this classical value can be singled out by the gyromagnetic factor $g$:

$$\mu \equiv g \frac{e_q}{2m} s \implies a \equiv \frac{1}{2}(g - 2) = F_2(0) \,, \tag{14}$$

where $a$ is called the anomalous part of the magnetic moment of such a particle. For the muon, we will use the symbols $a_\mu$ and $g_\mu$.

Once it is clear which part of the on-shell vertex in (8) actually contributes to $\mu$, it is convenient to project it out. The projector needed for the magnetic form factor is [68]:

$$P_2^\mu \equiv -\frac{m^2}{q^2(q^2 - 4m^2)} \left( \gamma^\mu + \frac{q^2 + 2m^2}{m(q^2 - 4m^2)} P^\mu \right) \,, \tag{15}$$

which is to be used in the following way:

$$F_2(q^2) = \mathrm{Tr}\{(\slashed{p} + m)P_{2\mu}(\slashed{p}' + m)\Gamma^\mu\} \,, \tag{16}$$

Note that there is a divergent $1/q^2$ factor in $P_{2\mu}$. Although expanding $\Gamma^\mu$ around $q = 0$ is inevitable, we can truncate the expansion at first order:

$$\Gamma^\mu(q^2) = \Gamma^\mu(0) + q_\nu \underbrace{\partial^\nu \Gamma^\mu(q^2)|_{q=0}}_{\equiv \Gamma^{\mu\nu}} \,. \tag{17}$$

To prove that, we must first reorganize $P_2$ using a slightly different version of the Gordon identity:

$$(\slashed{p}' + m)\gamma^\mu(\slashed{p} + m) = \frac{1}{2m}(\slashed{p}' + m)\{P^\mu + i\sigma^{\mu\nu}q_\nu\}(\slashed{p} + m) \,, \tag{18}$$

$$(\slashed{p} + m)P_2^\mu(\slashed{p}' + m) = (\slashed{p} + m)\frac{-m}{2(q^2 - 4m^2)}\left( -i\sigma^{\mu\nu}\frac{q_\nu}{q^2} + \frac{3}{(q^2 - 4m^2)}P^\mu \right)(\slashed{p}' + m) \,. \tag{19}$$

We see that the divergent term in the projector is of order $1/q$, so the conclusion follows.

By inserting (17) as well as (19) into (16) with the substitutions $p' = \frac{1}{2}(P + q)$ and $p = \frac{1}{2}(P - q)$, we evaluate the corresponding expression at $q = 0$ obtaining after some algebra:

$$a_\mu = \lim_{q \to 0} \frac{im}{2(q^2 - 4m^2)} \times$$

$$\mathrm{Tr}\left\{ \left( -\frac{1}{2}\slashed{q}\sigma_{\mu\nu}\frac{q^\nu}{q^2}\left[\frac{\slashed{P}}{2} + m\right] + \left[\frac{\slashed{P}}{2} + m\right]\sigma_{\mu\nu}\frac{q^\nu}{q^2}\left[\frac{\slashed{P}}{2} + m\right] + \frac{1}{2}\left[\frac{\slashed{P}}{2} + m\right]\sigma_{\mu\nu}\frac{q^\nu}{q^2}\slashed{q} \right)\Gamma^\mu(0) \right\} \tag{20}$$

$$+ \lim_{q \to 0} \frac{im}{2(q^2 - 4m^2)} \mathrm{Tr}\left\{ \left[\frac{\slashed{P}}{2} + m\right]\sigma_{\mu\nu}\frac{q^\nu q_\beta}{q^2}\left[\frac{\slashed{P}}{2} + m\right]\Gamma^{\mu\beta} \right\} - \frac{3}{8m^2}\mathrm{Tr}\left\{ p_\mu(m + \slashed{p})\Gamma^\mu(0) \right\} \,,$$

where we have explicitly evaluated the limit when possible. We still have divergent terms together with tensor dependence on $q^\mu$. To get rid of the latter, we will take advantage of the scalar character of $F_2(q^2)$ to perform Lorentz transformations on $q^\mu$ before taking the $q \to 0$ limit.

In particular, we can perform spatial rotations; thus, we carry out an angular average over the spatial components of $q^\mu$, taking $P$ as reference. The results are:

$$\int \frac{d\Omega}{4\pi} q^\mu = 0 \,, \qquad \int \frac{d\Omega}{4\pi} q^\mu q^\nu = \frac{q^2}{3}\left( g^{\mu\nu} - \frac{P^\mu P^\nu}{P^2} \right) \,. \tag{21}$$

The first result is obvious by the oddness of the integrand. The tensor structure of the second result is evident from Lorentz covariance and the scalar factors can be obtained straightforwardly by computing the Lorentz trace on both sides.

After inserting the angular averages (21) inside (20), we obtain:

$$a_\mu = \mathrm{Tr}\left\{ \left( \frac{1}{12}\gamma_\mu - \frac{1}{4}\frac{p_\mu}{m} - \frac{1}{3}\frac{1}{m^2}p_\mu\slashed{p} \right)\Gamma^\mu(0) + \frac{1}{48m}(\slashed{p} + m)[\gamma_\mu, \gamma_\beta](\slashed{p} + m)\Gamma^{\mu\beta} \right\} \,, \tag{22}$$

which is the direct relation between $a_\mu$ and Feynman amplitudes ($\Gamma^\mu$) that we were looking for and it opens the path to specialize the result to different topologies of diagrams and, in particular, to the HLbL one.

## 2.2. Specializing $a_\mu$ to HLbL Scattering Amplitudes

In this section, we specialize the result obtained in the previous section to compute $a_\mu$ from HLbL scattering amplitudes, whose diagrams are shown in Figure 3. The term "light-by-light" makes reference to the subdiagram appearing in Figure 3, which has four external photons (three virtual and attached to the muon line and one representing and external field). The term "hadronic" is due to the fact that only strongly interacting particles (quarks and gluons) or hadrons (mesons and various resonances) are allowed to appear in the blob of Figure 3, either as virtual exchanged particles or as poles of the amplitude, for the HLbL contributions.

The first step to specialize $a_\mu$ to HLbL is to isolate the appropriate subdiagram amplitudes from the muon electromagnetic vertex ones. Making use of the Feynman rules for QED, it is possible to read the result off the Feynman diagram in Figure 3:

$$-e\bar{u}_{p'\sigma'}\Gamma^{\mu_4}u_{p\sigma} = \bar{u}_{p'\sigma'}\int \frac{d^4q_1}{(2\pi)^4}\int \frac{d^4q_2}{(2\pi)^4}\frac{-i}{q_1^2}\frac{-i}{q_2^2}\frac{-i}{q_3^2}(-ie\gamma_{\mu_1})i\frac{\not{p}'+\not{q}_1+m}{(p'+q_1)^2-m^2}(-ie\gamma_{\mu_3})$$
$$\times i\frac{\not{p}-\not{q}_2+m}{(p-q_2)^2-m^2}(-ie\gamma_{\mu_2})\times e^4\,\Pi^{\mu_1\mu_2\mu_3\mu_4}(q_1,q_2,q_3)u_{p\sigma}\,, \tag{23}$$

where $e^4\,\Pi^{\mu_1\mu_2\mu_3\mu_4}$ represents the amplitude of the hadronic blob inside Figure 3:

$$\Pi^{\mu_1\mu_2\mu_3\mu_4} = -i\int d^4x\int d^4y\int d^4z\, e^{-i(q_1x+q_2y+q_3z)}\langle\Omega|J_s^{\mu_1}(x)J_s^{\mu_2}(y)J_s^{\mu_3}(z)J_s^{\mu_4}(0)|\Omega\rangle\,. \tag{24}$$

$|\Omega\rangle$ represents the QCD vacuum and $J_s$ stands for the electromagnetic current of strongly interacting particles. In the literature, $\Pi^{\mu_1\mu_2\mu_3\mu_4}$ is referred to as "fourth rank vacuum polarization tensor" [69–71] or "HLbL tensor" [53,59]. Introducing this new convention into (23), we obtain:

$$\Gamma_{\text{HLbL}}^{\mu_4} = -e^6\int \frac{d^4q_1}{(2\pi)^4}\int \frac{d^4q_2}{(2\pi)^4}\frac{\gamma_{\mu_1}}{q_1^2}\frac{\not{p}'+\not{q}_1+m}{(p'+q_1)^2-m^2}\frac{\gamma_{\mu_3}}{q_3^2}\frac{\not{p}-\not{q}_2+m}{(p-q_2)^2-m^2}\frac{\gamma_{\mu_2}}{q_2^2}\Pi^{\mu_1\mu_2\mu_3\mu_4}\,. \tag{25}$$

Before inserting this equation into (22), let us consider the analysis presented in [72], which concludes that the cross-section of a process evaluated in the limit in which an external photon becomes soft is equal to the sum of terms proportional to the amplitude of the process without the soft photon and its derivative plus vanishing contributions proportional to the soft photon momentum. In the context of HLbL scattering, the previous statement reads:

$$\Pi^{\mu_1\mu_2\mu_3\mu_4}(q_1,q_2,q_3) \sim \Pi^{\mu_1\mu_2\mu_3}A^{\mu_4} + \partial\Pi^{\mu_1\mu_2\mu_3}B^{\mu_4} + O(q_4)\,, \tag{26}$$

where $\Pi^{\mu_1\mu_2\mu_3}$ represents the three-photon scattering amplitude, $A_{\mu_4}$ and $B_{\mu_4}$ are two vectors of order $O(1/q_4)$ and $O(q_4^0)$, respectively, and $\partial$ represents a derivative with respect to a kinematic variable of the problem. $\Pi^{\mu_1\mu_2\mu_3}$ vanishes due to Furry's theorem, and therefore, $\Gamma_{\text{HLbL}}^{\mu_4} = 0$. Moreover, since $\partial\Pi^{\mu_1\mu_2\mu_3}$ obviously vanishes too, then $\Pi^{\mu_1\mu_2\mu_3\mu_4}$ vanishes (at least) linearly in the static field limit, and thus, one concludes that the derivative of $\Pi^{\mu_1\mu_2\mu_3\mu_4}$ with respect to $q_4$ does not contain singularities.

With respect to $\Gamma_{\text{HLbL}}^{\mu\alpha}$, from (25) we find:

$$\Gamma^{\mu_4\nu_4} = e^6\int \frac{d^4q_1}{(2\pi)^4}\int \frac{d^4q_2}{(2\pi)^4}\frac{\gamma_{\mu_1}}{q_1^2}\frac{\not{p}+\not{q}_1+m}{(p'+q_1)^2-m^2}\frac{\gamma_{\mu_3}}{q_3^2}\frac{\not{p}-\not{q}_2+m}{(p-q_2)^2-m^2}\frac{\gamma_{\mu_2}}{q_2^2}\partial^{\mu_4}\Pi^{\mu_1\mu_2\mu_3\nu_4}\Big|_{q_4\to 0}\,, \tag{27}$$

where the $q_4\to 0$ limit can be safely taken according to the previous analysis. Furthermore, we have used the antisymmetry between $\mu_4$ and $\nu_4$ of $\partial^{\mu_4}\Pi^{\mu_1\mu_2\mu_3\nu_4}|_{q_4\to 0}$, which can be

deduced by differentiating the Ward identity of the current conservation twice with respect to $q_4$ and taking the soft limit:

$$\partial^{\mu_4}\partial^{\nu_4}\left(q_{4,\alpha}\Pi^{\mu_1\mu_2\mu_3\alpha}\right)\Big|_{q_4\to 0} = 0 \implies \partial^{\nu_4}\Pi^{\mu_1\mu_2\mu_3\mu_4}\Big|_{q_4\to 0} = -\partial^{\mu_4}\Pi^{\mu_1\mu_2\mu_3\nu_4}\Big|_{q_4\to 0}.$$

Finally, turning back to $a_\mu^{\text{HLbL}}$, one obtains:

$$a_\mu^{\text{HLbL}} = \frac{e^6}{48m}\int\frac{d^4q_1}{(2\pi)^4}\int\frac{d^4q_2}{(2\pi)^4}\frac{1}{q_1^2}\frac{1}{q_2^2}\frac{1}{q_3^2}\frac{1}{(p+q_1)^2-m^2}\frac{1}{(p-q_2)^2-m^2}\frac{\partial}{\partial q_{4\mu_4}}\Pi^{\mu_1\mu_2\mu_3\nu_4}\Big|_{q_4\to 0}$$
$$\times \text{Tr}\left\{(\not{p}+m)[\gamma_{\mu_4},\gamma_{\nu_4}](\not{p}+m)\gamma_{\mu_1}(\not{p}+\not{q}_1+m)\gamma_{\mu_3}(\not{p}-\not{q}_2+m)\gamma_{\mu_2}\right\}. \tag{28}$$

There are only three steps left to obtain $a_\mu^{\text{HLbL}}$: (i) compute the Dirac trace, (ii) compute $\partial^{\mu_4}\Pi^{\mu_1\mu_2\mu_3\nu_4}|_{q_4\to 0}$, and (iii) compute the two-loop integral. The trace can be performed straightforwardly. On the other hand, the computation of the HLbL amplitude is very complex, and it is therefore necessary to study it in depth before advancing further.

### 2.3. Dispersive Computation of the HLbL Amplitude

In the previous subsection, we expressed $a_\mu$ in terms of the HLbL scattering amplitude $\Pi^{\mu_1\mu_2\mu_3\mu_4}$, and in this subsection, we will present the dispersive approach for its computation.

Since $\Pi^{\mu_1\mu_2\mu_3\mu_4}$ appears inside a two-loop integral on $q_1$ and $q_2$, it is necessary to compute it at different energy regions involving perturbative and non-perturbative regimes. This is due to asymptotic freedom, which invalidates perturbation theory at energy scales below $\Lambda_{QCD}\sim 1$ GeV because the coupling $\alpha_s$ approaches 1. Non-perturbative contributions give the bulk of $a_\mu^{\text{HLbL}}$, but high-energy studies are important for error estimation, so it is convenient to perform computations in both regimes in a unified framework.

QFT in the lattice and dispersive integrals are two of very few tools that allow for the computation of amplitudes in non-perturbative regimes. The first one tries to solve the QFT equations in a finite space-time cube of side length $L$ with discrete Euclidean space-time coordinates of spacing $a$. The observables of interest are then computed for different values of large $L$ and $1/a$, and these results are then extrapolated to $L, 1/a\to\infty$ in order to recover the standard QCD results. Using a very different perspective, the dispersive approach [34–36,53] relies on the analyticity of scattering amplitudes and the unitarity of the S-matrix (probability conservation) to relate the amplitudes of a process with the cross-sections of its sub-processes, which are fitted to data. Lattice computations have high numerical complexity due to the very large number of degrees of freedom that appear when both the size $L$ of the system and its resolution $1/a$ become large, as needed to reduce the uncertainty produced by the extrapolation to $L, 1/a\to\infty$. Therefore, only very recently have its results become competitive with the dispersive ones in terms of uncertainty both for HVP [24–29] and for HLbL [44–50]. Although the dispersive approach for the computation of the HLbL contribution to $a_\mu$ also has its drawbacks, the main one relating to the Mandelstam representation of $\Pi^{\mu_1\mu_2\mu_3\mu_4}$ has been recently overcome. This has allowed for obtaining the most reliable accounts of $a_\mu^{\text{HLbL}}$ in recent years [10] in addition to the aforementioned lattice results. It is worth noting that there are alternate dispersive frameworks in addition to the one found in [34,35,53]. For example, in [73–76], a dispersive equation is applied directly to the magnetic form factor $F_2$ instead of the HLbL Feynman amplitude.

As mentioned at the beginning of this subsection, in this work, we focus on the dispersive approach. It is based on four fundamental pillars: unitarity of the *S*-matrix, the Sugawara–Kanazawa theorem for functions of a complex variable, and the Schwarz reflection theorem. We will review such pillars in order.

### 2.3.1. Unitarity of the S-Matrix

A key concept in relativistic quantum theories of fundamental interactions is conservation of probability. For transition rates, it is equivalent to the unitarity of the S-matrix. Let us explore the consequences of such a feature for the transition matrix $T$:

$$S^{\dagger} = S^{-1} \implies (1 + iT)(1 - iT^{\dagger}) = 1 \implies TT^{\dagger} = i(T^{\dagger} - T) \,. \tag{29}$$

If we evaluate a certain matrix element of $S$ and insert a complete set of momentum eigenstates in the last equation, we obtain [77]:

$$2\mathrm{Im}\mathcal{M}(i \to f) = \sum_n \left( \Pi_{i=1}^n \int \frac{d^3 q_i}{(2\pi)^3} \frac{1}{2E_i} \right) \mathcal{M}^*(f \to \{q_i\}) \mathcal{M}(i \to \{q_i\}) (2\pi)^4 \delta^4 (P_i - \sum_i q_i) \,, \tag{30}$$

where $\mathcal{M}$ stands for a Feynman amplitude, $i$ represents the initial state, $f$ represents the final state, and $q_i$ represents the on-shell intermediate states, which come from the insertion of the identity resolved in terms of the momentum eigenstates. If the initial and final states are the same, like for HVP (see Figure 2), then we obtain the optical theorem.

At this point, (30) does not seem to be of much help. First, it can only be applied in principle to the on-shell electromagnetic vertex amplitude. Secondly, we only have the imaginary part of $\Pi^{\mu_1 \mu_2 \mu_3 \mu_4}$ and we need it in full to compute observables such as $a_{\mu}^{\mathrm{HLbL}}$ in (22). Additionally, we are tasked with computing one amplitude in terms of infinitely many and arbitrarily complex different subamplitudes. The first issue can be fixed by simplifying terms on both sides of the equation such that it applies just as well to the HLbL subdiagrams. To deal with the second one, we need to make use of the Sugawara–Kanazawa theorem, which reconstructs a complex variable function based on its poles and branch cuts. It will also show us why we can consider only intermediate states up to a certain energy scale in (30) and still obtain meaningful predictions.

### 2.3.2. Sugawara–Kanazawa Theorem and Schwarz Reflection Identity

Consider a function of a complex variable $z$ with (possibly) two branch cuts along the real axis: one to the right starting at $c_1$ and extending (possibly) to positive infinity and one to the left starting at $-c_2$ and extending (possibly) to negative infinity. Based on the following three requirements:

- $f(z)$ has finite limits in the positive real infinity direction above and below the right-hand cut.
- The limit of $f(z)$ in the negative real infinity direction above and below the left-hand cut exists.
- If $f(z)$ is divergent in a certain infinite direction, such a divergence is weaker than a polynomial with finite power $N$ such that $N \geq 1$.

Then the Sugawara–Kanazawa theorem [78] claims that $f(z)$ may be represented as:

$$f(z) = \sum_i \frac{R_i}{z - x_i} + \frac{1}{\pi} \left( \int_{c_1}^{\infty} + \int_{-\infty}^{-c_2} \right) \frac{\Delta_x f(x)}{x - z} \, dx + \lim_{x \to \infty} \overline{f}(x) \,, \tag{31}$$

$$\Delta_x f(x) = \frac{1}{2i} \{ f(x + i\epsilon) - f(x - i\epsilon) \} \,, \qquad \overline{f}(x) = \frac{1}{2} \{ f(x + i\epsilon) + f(x - i\epsilon) \} \,,$$

where $R_i$ represents the residue of $f(z)$ in $x_i$ which lies on the real interval $[-c_2, c_1]$. The two integrals in (31) are performed along the real axis. This representation of $f(z)$ is usually called "dispersion relation". The last term is referred to as the "subtraction constant" and accounts for possible divergences of $f(z)$ in infinity, which enter the equation as the contribution of the circumference of a Cauchy integration path at infinity. There is another result to this theorem that essentially claims that $f(z)$ has the same limit at infinity in any direction with positive (negative) imaginary part as it has along and above (under) the right (left) cut.

The HLbL process may be regarded as a function of two of the usual Mandelstam variables $s$, $t$ and $u$ for two-two scattering. Therefore, the Sugawara–Kanazawa theorem has to be applied twice, one for each independent variable, leaving the rest constant. However, the procedure is to be applied to a function that unifies all three channels' amplitudes. The resulting double dispersive integral is known as the Mandelstam representation [79]. The first step to build it is to write a single dispersive representation for, say, $s$, which we will consider to be unsubtracted for simplicity. Since we are expecting $\mathcal{M}$ to "contain" the amplitudes for the three channels, it must be invariant under crossing. As such, given that $t$ has a fixed value, we expect to have:

$$\mathcal{M}(s,t) = \sum_i \frac{R_i^s(t)}{s - x_i^s} + \frac{1}{\pi} \int_{c_1}^{\infty} \frac{\Delta_{s'}\mathcal{M}(s',t)}{s' - s}\, ds' + \frac{1}{\pi} \int_{c_1}^{\infty} \frac{\Delta_{u'}\mathcal{M}(u',t)}{u' - u}\, du' . \tag{32}$$

Now we perform an analytic continuation on $t$ to whichever value we require. Via interaction form factors, the $t$ dependence of the residues is usually well-known, but not for $\Delta\mathcal{M}$. To deal with this, we apply (31) once again, but this time for $\Delta_s\mathcal{M}$ with fixed $s'$:

$$\Delta_{s'}\mathcal{M}(s',t) = \frac{1}{\pi} \int_{c_3(s')}^{\infty} \frac{\Delta_{t'}\Delta_{s'}\mathcal{M}(s',t')}{t' - t}\, dt' + \frac{1}{\pi} \int_{c_3(s')}^{\infty} \frac{\Delta_{u'}\Delta_{s'}\mathcal{M}(s',u')}{u' - \overline{u}}\, du' \tag{33}$$

where $\overline{s}$ is the Mandelstam variable associated to $u'$ and $t$, while $\overline{u}$ is the Mandelstam variable associated to $s'$ and $t$. A similar equation applies for $\Delta_u\mathcal{M}$. Inserting this into the single dispersion relation, we obtain:

$$\mathcal{M}(s,t) = \frac{1}{\pi^2} \int_{c_1}^{\infty} ds' \int_{c_3(s')}^{\infty} dt' \frac{\Delta_{t'}\Delta_{s'}\mathcal{M}(s',t')}{(s' - s)(t' - t)} + \frac{1}{\pi^2} \int_{c_1}^{\infty} ds' \int_{c_3(s')}^{\infty} du' \frac{\Delta_{u'}\Delta_{s'}\mathcal{M}(s',u')}{(s' - s)(u' - u)}$$
$$+ \frac{1}{\pi^2} \int_{c_1}^{\infty} du' \int_{c_3(u')}^{\infty} dt' \frac{\Delta_{t'}\Delta_{s'}\mathcal{M}(t',u')}{(u' - u)(t' - t)} + \sum_i \frac{R_i^s(t)}{s - x_i^s} . \tag{34}$$

Some terms were simplified by noting that $s' - \overline{s} = s' - s + u' - u = u' - \overline{u}$. The $\Delta_i\Delta_j\mathcal{M}$ are called double spectral functions and can be obtained from the optical theorem with help from the Schwarz reflection principle. It states that if a function $f(z)$ of a complex variable $z$ is real along certain finite segment $\Gamma$ of the real axis, then $f^*(z) = f(z^*)$ in a domain $D$ of the $z$ complex plane that contains $\Gamma$ and in which $f(z)$ is analytic. This is always the case for any amplitude because at a sufficiently low center-of-mass (CM) energy, no intermediate state is allowed, and thus, the amplitude becomes real.

On the other hand, this theorem implies the existence of a discontinuity across the physical region of the real axis for each kinematic variable because there is always at least one possible intermediate state for any process: the initial one. Let $z$ be a kinematic variable of an amplitude $\mathcal{M}$ in a physical region, then:

$$2i\text{Im}\mathcal{M}(z + i\epsilon) = \mathcal{M}(z + i\epsilon) - \mathcal{M}^*(z + i\epsilon) = \mathcal{M}(z + i\epsilon) - \mathcal{M}(z - i\epsilon) = \Delta\mathcal{M}(z) . \tag{35}$$

This result relates the Mandelstam representation with (30). We can see that the constant $c_1$ of the dispersive integral is in fact the CM-frame energy of the lightest multiparticle intermediate state. One-particle intermediate states correspond to poles. Analogously, this helps us understand the meaning of $c_3$. For a physical value of $s$ and $t$, $\Delta\mathcal{M}$ is just $\text{Im}\mathcal{M}$ and is hence real. However, if we analytically continue $\Delta\mathcal{M}$ beyond the physically permissible boundaries of $t$, it may (and does) become complex, betraying the existence of a discontinuity. $c_2(s)$ is the point where that happens. Note that this is true even if $s$ takes a physically allowed value, as is the case in the initial fixed-$t$ single dispersive integral of $\mathcal{M}$. Note that contributions from heavy intermediate states are suppressed by (1) the $1/s'$ integral kernel and (2) a reduced integration region.

In summary, from this theoretical framework, it is possible to compute scattering amplitudes starting from (31) and then obtain the spectral functions from experimental

data for subprocesses via the optical theorem and the Schwarz reflection principle. This sentence implicitly claims that the analytic properties of an amplitude can be obtained entirely from its dynamics, that is, from the intermediate states it allows. This claim is known as the "Mandelstam hypothesis" and it is related to the causality requirement of the theory.

### 2.3.3. Tensor Decomposition of $\Pi^{\mu_1\mu_2\mu_3\mu_4}$

Dispersive integrals are suitable for scalar functions and $\Pi^{\mu_1\mu_2\mu_3\mu_4}$ is clearly not one. Hence, it is necessary to decompose it, like we did for the electromagnetic vertex, into form factors, which can then be dispersively represented. Nevertheless, the Mandelstam hypothesis cannot be considered to apply in general for these scalar coefficients. The key point behind this is that the tensor structures of the decomposition may have kinematic singularities and/or zeroes, which have to be cancelled by corresponding terms in the form factors, because the amplitude does not have such terms according to the Mandelstam hypothesis. In such cases, zeroes (singularities) change the asymptotic (analytic) behavior of the coefficients, and this has an impact on the dispersion relation in the form of subtraction constants in (31) and spurious residues. Such input has to be determined experimentally, and its presence in (31) further hinders the computations In fact, it may even introduce ambiguities in the soft photon limit of $\mathrm{Im}\Pi^{\mu_1\mu_2\mu_3\mu_4}$ and its derivatives. See [53]. In order to avoid these issues, it is necessary to find a tensor decomposition of the amplitude free of kinematic singularities and/or zeroes, which is called a Bardeen–Tarrach–Tung (BTT) decomposition and was found recently [34] for $\Pi^{\mu_1\mu_2\mu_3\mu_4}$. Let us review the main steps followed in [34].

The only covariant objects which we can work with are the metric and the momenta of the four photons. There are 138 possible combinations of said objects. The most general structure is therefore:

$$
\begin{aligned}
\Pi^{\mu_1\mu_2\mu_3\mu_4} = {}& g^{\mu_1\mu_2}g^{\mu_3\mu_4}\Pi^1 + g^{\mu_1\mu_3}g^{\mu_2\mu_4}\Pi^2 + g^{\mu_1\mu_4}g^{\mu_3\mu_2}\Pi^3 \\
& + \sum_{k,l} g^{\mu_1\mu_2}q_k^{\mu_3}q_l^{\mu_4}\Pi^4_{kl} + \sum_{j,l} g^{\mu_1\mu_3}q_j^{\mu_2}q_l^{\mu_4}\Pi^5_{jl} + \sum_{j,k} g^{\mu_1\mu_4}q_k^{\mu_3}q_j^{\mu_2}\Pi^6_{jk} \\
& + \sum_{i,l} g^{\mu_3\mu_2}q_i^{\mu_1}q_l^{\mu_4}\Pi^7_{il} + \sum_{i,k} g^{\mu_4\mu_2}q_k^{\mu_3}q_i^{\mu_1}\Pi^8_{ik} + \sum_{i,j} g^{\mu_3\mu_4}q_i^{\mu_1}q_j^{\mu_2}\Pi^9_{ij} \\
& + \sum_{i,j,k,l} q_i^{\mu_1}q_j^{\mu_2}q_k^{\mu_3}q_l^{\mu_4}\Pi^{10}_{ijkl} \,,
\end{aligned}
\tag{36}
$$

where the indices are $i \in \{2,3,4\}$, $j \in \{1,3,4\}$, $k \in \{1,2,4\}$, and $l \in \{1,2,3\}$. Each index may be any three-element subset of the four photon momenta. This choice in particular is due to [80] and is useful to determine constraints from crossing symmetry. There are no kinematic singularities in the scalar coefficients at this point. However, there are kinematic zeroes coming from two constraints that we have not yet explicitly accounted for: gauge invariance and crossing symmetry.

Gauge invariance may be explicitly imposed by projecting each one of the Lorentz indices of the amplitude onto the orthogonal space of the associated virtual photon momentum. To this end, the following projectors may be used [81]:

$$
I_{12}^{\mu\nu} = g^{\mu\nu} - \frac{q_1^\mu q_2^\nu}{q_1 \cdot q_2}, \qquad I_{34}^{\mu\nu} = g^{\mu\nu} - \frac{q_3^\mu q_4^\nu}{q_3 \cdot q_4} \,,
\tag{37}
$$

$$
\Pi^{\mu_1\mu_2\mu_3\mu_4} = I_{12}^{\mu_1'\mu_1} I_{12}^{\mu_2\mu_2'} I_{34}^{\mu_3'\mu_3} I_{34}^{\mu_4\mu_4'} \Pi_{\mu_1'\mu_2'\mu_3'\mu_4'} \,,
\tag{38}
$$

and only 43 tensor structures remain, which means that we have taken into account 95 constraints coming from gauge invariance. Furthermore, in four space-time dimensions, there are two additional linear relations among the 138 tensors of (36) [82] which reduce the number of independent tensors to 41. This reduction can be motivated by the number of

independent off-shell helicity amplitudes for HLbL scattering. In four dimensions, virtual photons have three helicity states $(+1, 0, -1)$; hence, the a priori number of such amplitudes is $3^4 = 81$. However, parity conservation in strong interactions constrains amplitudes related by parity transformations to be equal. This implies that 80 of these helicity amplitudes have a "copy", except the one for which all photons have helicity equal to zero, which transforms into itself. This analysis yields 41 independent helicity amplitudes.

Of course, there are other projectors suitable for the job. We could have given each momentum its own projector, for example:

$$g^{\mu_1 \mu_2} - \frac{q_1^{\mu_1} q_1^{\mu_2}}{q_1^2} \qquad \text{or} \qquad g^{\mu_1 \mu_2} - \frac{q_2^{\mu_1} q_2^{\mu_2}}{q_2^2} \,.$$

However, by applying any transverse projectors to $\Pi^{\mu_1 \mu_2 \mu_3 \mu_4}$, we are introducing kinematic singularities of the type $1/q_i \cdot q_j$. Thus, the less projectors we introduce and the simpler they are, the less types of kinematic singularities will be introduced.

In this case, two appearances of $I_{12}$ and two of $I_{34}$ in the tensor structures lead to poles associated to all the possible combinations of $q_1 \cdot q_2$ and $q_3 \cdot q_4$ with up to two repetitions of each. This can be solved by building linear combinations of these singular tensor structures such that the poles cancel. The precise procedure proposed by [81] deals with the poles by decreasing singularity order. In the first step, poles of the form $(q_1 \cdot q_2)^2 (q_3 \cdot q_4)^2$ are eliminated, first by linear combinations and, once this is not possible, by multiplying them by $q_1 \cdot q_2$ or $q_3 \cdot q_4$. The next step is to deal with the single-double poles in the same way, that is, $(q_1 \cdot q_2)(q_3 \cdot q_4)^2$ and $(q_1 \cdot q_2)^2 (q_3 \cdot q_4)$. Single-single poles then come, and so forth, until there are no kinematic singularities left. The decomposition obtained after this procedure is completed is not yet suitable for a dispersive representation because the tensor "basis" found is actually not linearly independent in $q_1 \cdot q_2 = 0$ or $q_3 \cdot q_4 = 0$, nor does it actually span the complete space of possible gauge-invariant tensors for the HLbL amplitude in those cases. This is due to the existence of 11 linear combinations of the basis elements that are proportional to $q_1 \cdot q_2$ and/or $q_3 \cdot q_4$ and some new tensor structures, thus introducing the linear dependence and span issues just mentioned. This phenomenon was first described by Tarrach [83] for the BTT decomposition of the $\gamma\gamma \to \pi\pi$ process. These 11 new structures have to be added to the group found previously with projectors to build a set that spans all relevant gauge-invariant tensors even at $q_1 \cdot q_2 = 0$ and $q_3 \cdot q_4 = 0$ [34]:

$$\Pi^{\mu_1 \mu_2 \mu_3 \mu_4} = \sum_i^{54} \hat{T}_i^{\mu_1 \mu_2 \mu_3 \mu_4} \hat{\Pi}_i \,. \tag{39}$$

In fact, the linear relations that are valid in four space-time dimensions mentioned in [82] have to be dropped to obtain a tensor decomposition suitable for dispersion relations [34], which means that for any number of space-time dimensions (four or otherwise), the minimal tensor "basis" suitable for dispersion relations contains 54 elements. Since the set is linearly dependent, the definition of the scalars $\hat{\Pi}_i$ is obviously redundant.

It is not surprising that the 41-element basis does not work as expected in the singular points of the projectors (37), but where do the new tensors come from? To shed light on that issue, let us consider, for example, $\Pi^7_{21}$, which is one of the form factors that disappears after using the gauge-invariant projectors. That means that it has been replaced by a sum of other form factors using a constraint equation. It can be explicitly obtained by studying the coefficient of $g^{\mu_3 \mu_2} q_1^{\mu_4}$ in the equation $q_{1\mu'_1} \Pi^{\mu'_1 \mu_2 \mu_3 \mu_4} = 0$, which yields the constraint:

$$0 = \Pi^3 + (q_1 \cdot q_2) \Pi^7_{21} + (q_1 \cdot q_3) \Pi^7_{31} + (q_1 \cdot q_4) \Pi^7_{41} \,. \tag{40}$$

It is clear that $\Pi^7_{21}$ cannot be simplified using this constraint when $q_1 \cdot q_2 = 0$; thus, the simplified tensor structures obtained using this constraint no longer span the amplitude fully. Furthermore, the remaining three form factors in (40) cease to be linearly independent when $q_1 \cdot q_2 = 0$, thus hinting at the other issue we found. One might think that this is just

an unfortunate result due to a poor solution of the constraint system, i.e., bad choice of projectors, and that we should look for one that solves for $\Pi^3$ instead of $\Pi^7_{21}$, for example. Unfortunately, this is not possible; it is not hard to see that for all four gauge invariance constraint equations, only $\Pi^1$, $\Pi^2$, and $\Pi^3$ can be replaced without compromising the kinematics of the problem, and there are 95 different constraint equations, so there will always be at least 92 replacements that will have problematic limits.

Finally, it is worth noting that, thanks to the choice of the $i, j, k, l$ in (36), the 54-tensor "basis" (It is a basis in the sense that it spans the tensor structures of the amplitude, but it is not linearly independent) is closed under crossing. In fact, only seven of the fifty-four tensors are actually independent in terms of crossing transformations There are some elements that are actually crossing antisymmetric, but the corresponding kinematic zero does not affect the computation of $a_\mu$ [34].

*2.4. Master Formula for the HLbL Contribution to $a_\mu$*

In this subsection, we obtain $a_\mu^{\text{HLbL}}$ from the scalar coefficients of (39), which is key to connecting it to experimental data in the low-energy regime via dispersion relations.

Since the tensors $\hat{T}_i^{\mu_1\mu_2\mu_3\mu_4}$ from (39) have all the kinematic constraints and symmetries incorporated, they must vanish in the soft photon limit like $\Pi^{\mu_1\mu_2\mu_3\mu_4}$ does, as argued previously. Furthermore, it is possible to find $\hat{T}_i^{\mu_1\mu_2\mu_3\mu_4}$ such that in the limit $q_4 \to 0$, the derivatives of 35 of these tensors vanish, which leads us from (28) to:

$$
\begin{aligned}
a_\mu^{\text{HLbL}} &= e^6 \int \frac{d^4 q_1}{(2\pi)^4} \int \frac{d^4 q_2}{(2\pi)^4} \frac{1}{q_1^2} \frac{1}{q_2^2} \frac{1}{(q_1+q_2)^2} \frac{1}{(p+q_1)^2 - m^2} \frac{1}{(p-q_2)^2 - m^2} \sum_i^{19} \hat{T}_i \hat{\Pi}_i \ , \\
\hat{T}_i &\equiv \frac{1}{48m} \text{Tr} \left\{ (\slashed{p}+m)[\gamma_{\nu_4}, \gamma_{\mu_4}](\slashed{p}+m)\gamma_{\mu_1}(\slashed{p}+\slashed{q}_1+m)\gamma_{\mu_2}(\slashed{p}-\slashed{q}_2+m)\gamma_{\mu_2} \right\} \\
&\quad \times \left( \frac{\partial}{\partial q_{4\nu_4}} \hat{T}_i^{\mu_1\mu_2\mu_3\mu_4} \Big|_{q_4 \to 0} \right) .
\end{aligned}
\tag{41}
$$

The objects $\hat{T}_i$ act as kernels for the two-loop integral. Their number can be further reduced to 12 by harnessing the symmetry of the integral and some of the kernels under the $q_1 \leftrightarrow -q_2$ exchange, which implies that some pairs of kernels actually give the same result and can be absorbed into one.

At this point, $a_\mu^{\text{HLbL}}$ seems to depend on $p$, but we know from momentum conservation that $F_2 = F_2(q_4^2)$, and hence, $a_\mu^{\text{HLbL}}$ is not a function of any momentum. We can remove the spurious dependence on $p$ through angular averages. Let us start by performing a Wick rotation, in which we essentially render all four vectors' time components imaginary. This causes all scalar products to acquire a minus sign and renders the corresponding metric Euclidean, transforming the two loops accordingly. This has non-trivial consequences, because it amounts to a rotation of the real axis of time-component integration into the imaginary one. In the case that the region swept by such a rotation contains singularities, the resulting contributions have to be taken into account accordingly. However, for $a_\mu$, there are no such issues [34], and the Wick rotation may be performed without problems. The Wick-rotated version of the momenta is represented by $Q_1$, $Q_2$, and $P$ $\hat{P}$ and $\hat{Q}_i$ represent their unit vectors, while $|P|$ and $|Q_i|$ represent their norm. Then it is possible to remove the spurious dependence on $P$ by averaging $a_\mu^{\text{HLbL}}$ over all possible orientations of $P$:

$$
a_\mu^{\text{HLbL}} = \int \frac{d^4\Omega(P)}{2\pi^2} a_\mu^{\text{HLbL}}
\tag{42}
$$

Wick-rotated propagators strongly resemble the generating function of Gegenbauer polynomials [84], which allows us to represent each one of the former as a linear combination of the latter. Finally, the integrals are found using the polynomials' orthogonality [85]. After performing the corresponding average, it is possible to perform five of the six four-dimensional angular integrals on $Q_1$ and $Q_2$, and the final result is:

$$a_\mu^{\text{HLbL}} = \frac{2\alpha^3}{3\pi^2} \int_0^\infty dQ_1 \int_0^\infty dQ_2 \int_{-1}^1 d\tau \sqrt{1-\tau^2} |Q_1|^3 |Q_2|^3 \times \sum_i^{12} T_i \overline{\Pi}_i \,. \tag{43}$$

There are three aspects of this last step that are worth noting:

- The integral over $Q_2$ in spherical coordinates is considered in the first place. It is possible to take any four momenta as a reference for the angular integral; it does not matter because the integrals will go over all the possible values anyway. We take $Q_1$ as a reference.
- The integrand is only dependent on one angle (in $\tau = \hat{Q}_1 \cdot \hat{Q}_2$), and it is therefore convenient to assign $\tau$ as one of the three Euclidean angles over which the angular integrals of the four-momentum $Q_2$ is performed. It is relevant which of these three angles we are referring to, because it will determine what the Jacobian will look like. In the master formula, there is a term $1 - \tau^2$, a sine squared, which means $\tau$ does not represent either the polar or azimuthal angle of the three-dimensional sphere embedded in the four-dimensional space. Thus, the angular integral on $Q_2$ yields:

$$\int d\tau \sqrt{1-\tau^2} \int d\theta d\phi \sin\theta = 4\pi \int d\tau \sqrt{1-\tau^2} \,, \tag{44}$$

  where $\theta$ and $\phi$ represent the three-dimensional polar and azimuthal angles of the four-dimensional $Q_2$ space.
- Once the angular integrals on $Q_2$ have been performed, there is no dependence on $\tau$ or another angle left on the integrand. This means that we can perform the four-dimensional solid angle integral on $Q_1$, which yields $2\pi^2$.

In summary, we have presented the basics of the computation of $a_\mu$ and in particular $a_\mu^{\text{HLbL}}$, which led to the (43) master formula, which is based on a BTT tensor decomposition of the HLbL amplitude with 54 elements such that its scalar coefficients are free of kinematic singularities and zeroes. The tensor "basis" is actually not linearly independent in the whole phase space due to 11 structures that need to be added by hand in order to span the amplitude at the kinematic points $q_1 \cdot q_2 = 0$ or $q_3 \cdot q_4 = 0$. Furthermore, in four space-time dimensions, there are two additional linear relations among the 138 tensors of (36) [82] which the 54 BTT structures inherit. Of course, the linear dependence of these tensor structures introduces redundancies in the definition of the associated scalar coefficients. However, in [37] it was shown that these redundancies do not affect the observables, as expected, due to a series of dispersive sum rules obeyed by the form factors.

As we stated in Section 2.3, the dispersive calculation of an amplitude offers a way to establish a hierarchy of contributions for the intermediate states that enter the computation via the unitarity relation of (30). In the context of low-energy HLbL scattering, the most relevant intermediate states are expected to be the lightest (up to 1–2 GeV) one- or two-hadron intermediate states, and a thorough summary of their contributions can be found in [10] along with preliminary results for higher multiplicity contributions. The biggest among these by far are the pseudo-scalar poles ($\pi$, $\eta$, and $\eta'$). Subleading contributions like axial mesons are still a significant source of uncertainty and require further study. It is also necessary to better understand the behavior of the HLbL tensor beyond the 1–2 GeV threshold. The high-energy regime of the HLbL contribution has been already reached from the light intermediate states when the high-energy part of the integral in (43) has been carried out for high virtuality values of the transition and vector form factors. Although these penetrations into the high-energy regime are expected to take into account most of the contribution, in this regime, heavier states with more complex topologies that are not taken into account dispersively can also contribute. It is this high-energy regime of HLbL scattering in the context of $a_\mu$ with which the next sections deal.

## 3. Operator Product Expansion of $\Pi^{\mu_1\mu_2\mu_3\mu_4}$

Computations in the high-energy regime of QCD play several important roles in the determination of $a_\mu$. First, for data-driven computations, they provide the asymptotic behavior of transition and electromagnetic form factors for the high-energy "tail" of the integral in the master formula (43) and dispersive integrals. Examples of these short-distance constraints (SDCs) can be found in [86–91] and in [36,92] for the specific case of the pion. Second, SDCs can be found for the HLbL amplitude itself, which are used to evaluate how much of this asymptotic behavior is recovered by dispersive computations and thus assess the uncertainty produced by the missing high-energy contributions. These come from the heavy intermediate states with topologically more complex unitarity diagrams that are ignored in the dispersive approach.

In this work, we are interested in the SDC for the HLbL tensor. There are two high-energy regimes for the HLbL process with one real soft photon: one where all three Euclidean virtualities are similarly high ($Q_1^2 \sim Q_2^2 \sim Q_3^2 \gg \Lambda_{QCD}^2$) (we refer to the Euclidean virtualities $q_i^2 = -Q_i^2$), and one where two are similarly high and much greater than the third Euclidean virtuality ($Q_1^2 \sim Q_2^2 \gg Q_3^2 \sim \Lambda_{QCD}^2$ and crossed versions). It is worth remembering that we refer to the Euclidean virtualities, because the considered photons are far off-shell, since when the space-time separation goes to zero, the corresponding momenta $q_i$ become space-like. Each of these regimes imposes asymptotic behavior constraints on different subsets of BTT scalar coefficients, and therefore, they allow for the independent evaluation of the different sets of intermediate states. In this section, we will focus on the $Q_1^2 \sim Q_2^2 \sim Q_3^2 \gg \Lambda_{QCD}^2$ regime of the HLbL tensor, which we will study by performing an Operator Product Expansion (OPE) where the soft photon is introduced as a background electromagnetic field, as was performed in [60].

### 3.1. OPE of $\Pi^{\mu_1\mu_2\mu_3\mu_4}$ in an Electromagnetic Background Field: A First Look

A product of operators carrying very high momenta can be expressed in terms of a linear combination of local operators carrying zero momentum with singular coefficients that carry the momentum dependence of the original operator.

From this definition, it is evident that the OPE constitutes a very well-suited framework for the evaluation of the HLbL tensor in the $Q_1^2 \sim Q_2^2 \sim Q_3^2 \gg \Lambda_{QCD}^2$ regime. However, the limit $q_4 \to 0$ does not allow us to include the fourth current of (24) in the OPE. One could in principle start the construction of such an OPE for the four currents of $\Pi_{HLbL}^{\mu_1\mu_2\mu_3\mu_4}$, but it does not work for our particular problem. For example, the Wilson coefficient associated to the identity operator is just the HLbL tensor in perturbative QCD, which involves an expansion in terms of the strong coupling constant and the usual large logarithms for increasing powers: $\alpha_s^n(\mu) \ln^n \left\{ Q_4^2/\mu^2 \right\}$, where $\mu$ represents the renormalization subtraction point in the $\overline{MS}$ scheme. In order for the logarithms not to blow up, it is necessary to have $\mu \sim Q_4$, but in such a case, $\alpha_s$ would enter the non-perturbative domain of QCD and the expansion would be spoiled anyway. Wilson coefficients for higher-dimensional operators also suffer from infrared singularities since they depend upon the $1/q_4^2$ propagator. Including the fourth current $J^{\mu_4}$ in the OPE, even though its momentum is not large, is not the only problem. Even if the OPE were performed only for the three currents with high momenta, there would still be matrix elements of the type $\langle 0|O_n J^{\mu_4}(q_4)|0\rangle$, which cannot be perturbatively computed in QCD. In general, these issues are different consequences of the fact that perturbative QCD is not the correct framework to describe the soft interaction that is required by $a_\mu$. It is therefore necessary to perform the OPE of the three high-momentum currents only and also take the fourth one into account properly. This can be achieved by letting the soft photon be introduced by an external electromagnetic field instead of a quark current. This approach was first used in [93] in the context of the computation of the magnetic moment of nucleons, then it was used in [94] for the hadronic corrections to the electroweak contribution to $a_\mu$, and finally it was again picked up in [59–61] for the HLbL tensor. A pedagogical review of the framework is presented in [95].

A new object suitable for such an OPE is:

$$\Pi^{\mu_1\mu_2\mu_3} = \frac{1}{e}\int d^4x \int d^4y\, e^{-i(q_1x+q_2y)} \langle 0|T\,J^{\mu_1}(x)J^{\mu_2}(y)J^{\mu_3}(0)|\gamma(q_4)\rangle = -\epsilon_{\mu_4}(q_4)\Pi^{\mu_1\mu_2\mu_3\mu_4}\,,$$

where the soft photon $q_4 \to 0$ is included implicitly in the initial state. In addition, this time, $J^\mu$ makes reference to the electromagnetic current of the three lightest quarks, namely, up, down, and strange or $u$, $d$, and $s$, and thus:

$$J^\mu = \overline{\Psi}\hat{Q}\gamma^\mu\Psi \qquad\qquad \hat{Q} = \mathrm{diag}\left(\frac{2}{3}, -\frac{1}{3}, -\frac{1}{3}\right)\,, \tag{45}$$

where $\hat{Q}$ is the charge matrix and now $\Psi$ is a vector of bispinors with quark flavor and color indices, which are summed upon and suppressed in the current.

Since $\langle 0|...|\gamma(q_4)\rangle$ is an on-shell matrix element, only gauge-invariant operators contribute to the OPE. From these, only $F_{\mu\nu}$, the field strength tensor, contributes to first order in $q_4$, and hence, only operators that have the same quantum numbers and symmetries of $F_{\mu\nu}$ are relevant. In summary, this means that at the first order in the external electromagnetic field, we have for the regime of high virtualities at hand:

$$\Pi^{\mu_1\mu_2\mu_3} \equiv i\Pi_F^{\mu_1\mu_2\mu_3\mu_4\mu_5}(q_1,q_2)\langle 0|F_{\mu_4\mu_5}(0)|\gamma(q_4)\rangle = q_{4\mu_4}\epsilon_{\mu_5}(q_4)\Pi_F^{\mu_1\mu_2\mu_3[\mu_4\mu_5]}\,.$$

$$\left.\frac{\partial\Pi^{\mu_1\mu_2\mu_3\mu_4}}{\partial q_{4\mu_5}}\right|_{q_4\to 0} = \Pi_F^{\mu_1\mu_2\mu_3[\mu_4\mu_5]}\,. \tag{46}$$

We see here a confirmation of the antisymmetric nature of $\partial\Pi^{\mu_1\mu_2\mu_3\mu_4}$ arising from gauge invariance. We can also see that the real object of interest for $a_\mu^{\mathrm{HLbL}}$ is actually $\Pi_F^{\mu_1\mu_2\mu_3[\mu_4\mu_5]}$, which is made from the Wilson coefficients of the OPE. Note that it is explicitly free of any $q_4$ dependence and therefore does not suffer from singularities at $q_4^2 \to 0$.

We already discussed that the OPE elements for $\Pi^{\mu_1\mu_2\mu_3}$ have the same Lorentz structure and symmetries of $F_{\mu_1\mu_2}$, that is:

- second rank antisymmetric tensor;
- odd charge conjugation parity (remember, in this regard, the famous Furry's theorem).

In [60], operators with these features and mass dimension up to 6 are taken into account, and the rest are neglected. This choice is ultimately supported by the fact the contribution of higher-dimensional operators turns out to be at least two orders of magnitude smaller than the leading order. It is, however, also true that the non-perturbative matrix elements of the dimension seven operators are less known. We focus here, for simplicity of reading at this step of our analysis, on the case of only one flavor and therefore on the following list of operators:

$$
\begin{aligned}
S_{1,\mu\nu} &\equiv ee_f F_{\mu\nu}\,, & S_{2,\mu\nu} &\equiv \overline{\Psi}\sigma_{\mu\nu}\Psi\,, \\
S_{3,\mu\nu} &\equiv ig_S\overline{\Psi}G_{\mu\nu}\Psi\,, & S_{4,\mu\nu} &\equiv ig_S\overline{\Psi}\,\overline{G}_{\mu\nu}\gamma_5\Psi\,, \\
S_{5,\mu\nu} &\equiv \overline{\Psi}\Psi\,ee_f F_{\mu\nu}\,, & S_{6,\mu\nu} &\equiv \frac{\alpha_s}{\pi}G_a^{\alpha\beta}G_{\alpha\beta}^a\,ee_f F_{\mu\nu}\,, \\
S_{7,\mu\nu} &\equiv g_S\overline{\Psi}(G_{\mu\lambda}D_\nu + D_\nu G_{\mu\lambda})\gamma^\lambda\Psi + g_S\overline{\Psi}(G_{\nu\lambda}D_\mu + D_\mu G_{\nu\lambda})\gamma^\lambda\Psi\,, \\
S_{\{8\},\mu\nu} &\equiv \alpha_s(\overline{\Psi}\Gamma\Psi\overline{\Psi}\Gamma\Psi)_{\mu\nu}\,,
\end{aligned}
\tag{47}
$$

where $\Psi$ represents again a quark field in a given flavor of electric charge $ee_f$, the color indices are implicitly summed upon, $\Gamma$ represents a combination of Dirac gamma matrices, $D_\nu$ represents the gauge-covariant derivative, $G_{\mu\nu}^a$ represents the gluon field strength tensor, $G_{\mu\nu} \equiv it^a G_{\mu\nu}^a$ and $\overline{G}_{\mu\nu} \equiv \frac{i}{2}\epsilon^{\mu\nu\alpha\beta}G_{\alpha\beta}$. Since the largest non-perturbative QCD energy scale is the perturbative threshold $\Lambda_{QCD}$, then the contributions from operators with mass dimension $d$ are expected to be suppressed like $\left(\frac{\Lambda_{QCD}}{Q_i}\right)^{d-1}$ (Note that the matrix element

$\langle 0|...|\gamma \rangle$ of an operator with mass dimension $d$ has mass dimension $d - 1$), which means that an $O\left(\frac{\Lambda_{QCD}^d}{Q_i^d}\right)$ error is introduced when a cut-off dimension $d$ is imposed. When included in the high-energy integration region of the master formula (43), the integration domain should be used coherently in agreement with the mass dimension cut-off of the OPE.

Due to the quantum numbers of the $S_{i,\mu\nu}$ operators, we will be able to factorize the plane wave of the external electromagnetic field times the non-zero expectation value in the true QCD vacuum of a Lorentz, gauge-invariant and charge conjugation even operator, a so-called condensate $X_i^S$:

$$\langle 0|S_{i,\mu\nu}|\gamma \rangle \equiv X_i^S \langle 0|F_{\mu\nu}|\gamma \rangle \,, \tag{48}$$

which render the connection between the $S_{i\mu\nu}$ operators and $\Pi_F$ more evident.

Now that the operator basis is known up to dimension six, the next step is to obtain the Wilson coefficients of its elements. Since these are local operators, evaluated at $x = 0$, the obvious step is to perform a Taylor expansion of the field variables. However, such an expansion hinders the computation of the Wilson coefficients because it involves terms that are not even gauge-covariant, while the $S_{i,\mu\nu}$ are gauge-invariant. Hence, the computation of the Wilson coefficients becomes very complex, particularly for the gluon operators, as can be seen in [96,97].

Instead, if one chooses the Fock–Schwinger gauge [98] for the photon and gluon fields, defined by the constraint $A_\mu(x - x_0)^\mu = 0$ for some constant $x_0$, the following expansions hold [99]:

$$\Psi(x) = \Psi(0) + x^{\mu_1} D_{\mu_1} \Psi(0) + \frac{1}{2!} x^{\mu_1} x^{\mu_2} D_{\mu_1} D_{\mu_2} \Psi(0) + ... \,,$$

$$A_\alpha^a(x) = \frac{1}{2 \times 0!} x^{\mu_1} G_{\mu_1\alpha}^a(0) + \frac{1}{3 \times 1!} x^{\mu_1} x^{\mu_2} D_{\mu_1} G_{\mu_2\alpha}^a(0) + ... \,, \tag{49}$$

$$A_\alpha(x) = \frac{1}{2 \times 0!} x^{\mu_1} F_{\mu_1\alpha}(0) + \frac{1}{3 \times 1!} x^{\mu_1} x^{\mu_2} D_{\mu_1} F_{\mu_2\alpha}(0) + ... \,,$$

where $A^a$ represents the gluon field and we have chosen $x_0 = 0$ for simplicity. The gauge-covariant form of these expansions offers a clear advantage for the computation of Wilson coefficients associated to gauge-invariant operators. On the other hand, the gauge-fixing constraint evidently breaks traslational invariance and the propagator, being gauge-dependent, inherits this feature, which results in a rather complex expression [100]. As a result, the gain in simplicity due to explicit gauge covariance may be lost in perturbative corrections due to the complexity of the propagators.

A way around this issue is to split the gauge field into a background part that acts as a classical field that parameterizes the effects of the non-perturbative QCD vacuum on perturbative dynamics dictated by the asymptotic freedom and a fluctuation part that represents the perturbative oscillations around the vacuum solution. The former leads to the non-zero values $\langle S_{i,\mu\nu} \rangle$ and can be fixed by the radial gauge constraint, but it is not a dynamical field variable and hence does not have an associated propagator, and vice versa for the fluctuation part, which is fixed by a different gauge constraint that we will explore in the next section. Such a split can be represented as:

$$A_\mu(x) = a_\mu(x) + A_\mu'(x) \,, \qquad A_\mu^a(x) = a_\mu^a(x) + A_\mu^{'a}(x) \,, \tag{50}$$

where the unprimed variables are the classical fields and the primed ones represent the fluctuations around this classical value. It is not needed to include the specific form of the QCD vacuum fields but just to parameterize them as external fields, as is, for instance, carried out in the Coleman–Weinberg approach. In the next subsection, we present in detail the theoretical framework that supports the background field method for these split fields.

### 3.2. OPE of $\Pi^{\mu_1\mu_2\mu_3\mu_4}$ in an Electromagnetic Background Field: Theoretical Framework

The basic rules in background field theory are as follows. (1) Diagrams contain no external lines of quantum fluctuations; (2) matrix elements in the original theory (i.e., without field splitting) can be computed by functionally differentiating vacuum-to-vacuum diagrams in the background theory with respect to the background fields, which justifies the fact that a product of background fields, say $\overline{\psi}\sigma_{\mu\nu}\psi$, is related to the composite operator $\overline{\Psi}\sigma_{\mu\nu}\Psi$; and (3) the gauge-fixing term in the Lagrangian is constrained to be $-\frac{1}{2\xi}(\overline{D}^{\mu}A_{\mu}^{'a})^2$, where $\overline{D}^{\mu}A_{\mu}^{'a} \equiv \partial^{\mu}A_{\mu}^{'a} + g_S f^{abc}a_{\mu}^{b}A^{'c\mu}$ acts as a background covariant derivative and $\xi$ is the arbitrary gauge-fixing parameter, in order to keep the Lagrangian's invariance under gauge transformations of the background fields.

Now that we have presented the theoretical framework of the separation of fields that we will use, we are ready to obtain the computational tools that we will need to build the OPE. Let us read in detail the Feynman rules from the Lagrangian $\mathcal{L}^{\text{HLbL}}$ after the separation of the fields in classical background and quantum parts:

$$
\begin{aligned}
\mathcal{L}^{\text{HLbL}} =\ & -\frac{g_S}{2} f^{a\mu\nu} f^{abc} A_{\mu}^{'b} A_{\nu}^{'c} - \frac{1}{2}\left(\overline{D}^{\mu}A^{'a\nu}\overline{D}_{\mu}A_{\nu}^{'a} + \overline{D}^{\mu}A_{\mu}^{'a}\overline{D}^{\nu}A_{\nu}^{'a} - \overline{D}^{\mu}A^{'a\nu}\overline{D}_{\nu}A_{\mu}^{'a}\right) \\
& - g_S f^{a\overline{b}\overline{c}} A^{'\overline{b}\mu} A^{'\overline{c}\nu}\overline{D}_{\mu}A_{\nu}^{'a} - \frac{1}{4}g_S^2 f^{abc} f^{a\overline{b}\overline{c}} A_{\mu}^{'b} A_{\nu}^{'c} A^{'\overline{b}\mu} A^{'\overline{c}\nu} \\
& + \overline{\psi}_l'(\{i\slashed{\partial} - m\}\delta_{lk} + e\hat{Q}\delta_{lk}\slashed{a} + g_S t_{lk}^a \slashed{a}^a)\psi_k' + \overline{\psi}_l(e\hat{Q}\delta_{lk}\slashed{A}' + g_S t_{lk}^a \slashed{A}^{'a})\psi_k' \\
& + \overline{\psi}_l'(e\hat{Q}\delta_{lk}\slashed{A}' + g_S t_{lk}^a \slashed{A}^{'a})\psi_k + \overline{\psi}_l'(e\hat{Q}\delta_{lk}\slashed{A}' + g_S t_{lk}^a \slashed{A}^{'a})\psi_k' \\
& - w_a^*\overline{D}_{\mu}\{\overline{D}^{\mu}w_a + f_{abc}w_c A_{b}^{'\mu}\}\,,
\end{aligned}
\tag{51}
$$

where we have split the quark fields following the same (un)primed convention and we have included color indices $l$ and $k$. Furthermore, $f_{\mu\nu}^a$ and $f_{\mu\nu}$ are background counterparts of the usual field strength tensors and ghost fields are represented by $w$ and $w^*$. This expression has, of course, been heavily simplified by ignoring terms that are either constant or linear in fluctuation fields. The classical background fields actually minimize the *quantum* effective action, which in the absence of external sources and at leading order, is equivalent to the same statement on the classical action. For quark and gluon fields, the vacuum expectation values of course do not receive perturbative contributions, since tadpole perturbative diagrams are zero at all orders. Photon fluctuations have also been ignored due to our exclusive interest in strong interactions.

As usual, the kernel of quadratic terms in a specific path-integral variable is the inverse of the corresponding free propagator, while the rest are interaction vertices. These are very similar to the ones in a theory with no background fields, the only difference being that any one line can now be created or annihilated by the vacuum. Feynman rules for quark-gluon vertices are summarized in Figure 4. The vertices for gluon self-interactions can be found straightforwardly, but we do not quote them here because they are not relevant at the perturbative level we are working at.

The quark-free propagator is modified by the background fields in the usual way:

$$
(\{i\slashed{\partial} - m\}\delta_{l'l}\delta^{f'f} + e\hat{Q}^{f'f}\delta_{l'l}\slashed{a} + g_S\delta^{f'f}t_{l'l}^a\slashed{a}^a)S_{lk}^{fs}(x,y) = i\delta^4(x-y)\delta^{f's}\delta_{l'k}\,.
\tag{52}
$$

Therefore, it can be computed recursively by supposing that the strength of the background gauge fields is much smaller than the characteristic momentum of the process of interest, which is a reasonable hypothesis in our context. Note that the new indices $f'f$ represent quark flavor and had been suppressed previously. In the approximation of a weak external electromagnetic field, an expansion to $O(e)$ is enough. For the background gluon field, it is necessary to go to $O(g_S^2)$ in agreement with the highest mass dimension of the local operators that were chosen for the OPE.

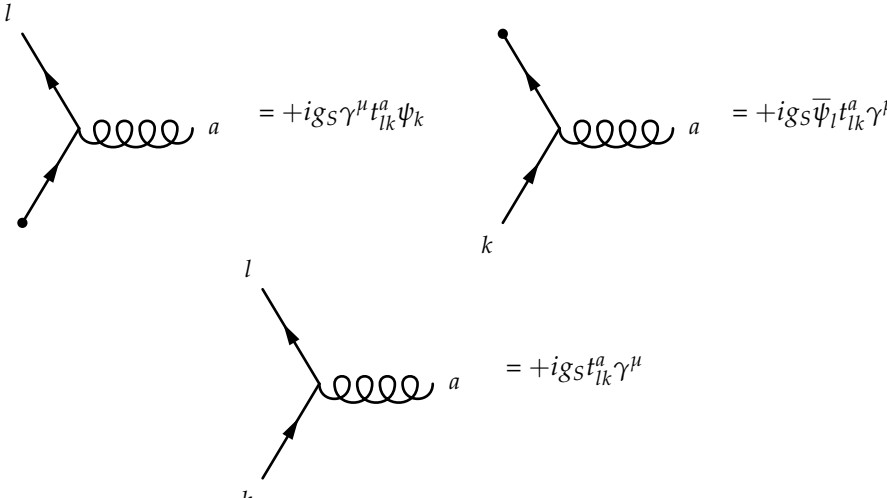

**Figure 4.** These figures show the three different types of quark interactions with a quantum gluon. In the two diagrams at the top, one quark line is non-pertubatively annihilated by the vacuum. $l$ and $k$ represent the color of the quarks, $a$ represents the color of the gluon, and trivial quark flavor indices are suppressed.

In addition, we will see that due to the presence of a background, the propagator is not translationally invariant. This of course also breaks momentum conservation along the propagator in its Fourier-transformed version. Therefore, there are in general three different types of momentum-space quark propagators:

$$S_{lk}^{fs}(p_1, p_2) \equiv \int d^4x \int d^4y \, e^{ip_1 x} e^{-ip_2 y} S_{lk}^{fs}(x,y) \,, \tag{53}$$

$$S_{lk}^{fs}(p_1) \equiv \int d^4x \, e^{ip_1 x} S_{lk}^{fs}(x,0) = \int \frac{d^4 p_2}{(2\pi)^4} S_{lk}^{fs}(p_1, p_2) \,, \tag{54}$$

$$\tilde{S}_{lk}^{fs}(p_2) \equiv \int d^4x \, e^{-ip_2 x} S_{lk}^{fs}(0,x) = \int \frac{d^4 p_1}{(2\pi)^4} S_{lk}^{fs}(p_1, p_2) \,, \tag{55}$$

By Taylor-expanding the gauge fields, we have intrinsically assumed that they are soft, but not even that restores translation invariance of the propagator. In summary, the free quark propagator in the momentum space (and in the presence of background gluon and photon fields) at $O(eg_S^2)$ is:

$$
\begin{aligned}
S_{lk}^{fs}(p_1, p_2) ={}& (2\pi)^4 \delta^4(p_1 - p_2) S_{p_1}^0 \delta^{fs} \delta_{lk} \\
&+ i S_{p_1}^0 \int_{q_1} \{ e\hat{Q}^{fs} \delta_{lk} \slashed{A}_{q_1} + g_S t_{lk}^a \delta^{fs} \slashed{A}_{q_1}^a \} S_{p_1+q_1}^0 (2\pi)^4 \delta^4(p_1 - p_2 + q_1) \\
&- e\hat{Q}^{fs} g_S t_{lk}^a S_{p_1}^0 \int_{q_1} \slashed{A}_{q_1} S_{p_1+q_1}^0 \int_{q_2} \slashed{A}_{q_2}^a S_{p_1+q_1+q_2}^0 (2\pi)^4 \delta^4(p_1 - p_2 + q_1 + q_2) \\
&- e\hat{Q}^{fs} g_S t_{lk}^a S_{p_1}^0 \int_{q_1} \slashed{A}_{q_1}^a S_{p_1+q_1}^0 \int_{q_2} \slashed{A}_{q_2} S_{p_1+q_1+q_2}^0 (2\pi)^4 \delta^4(p_1 - p_2 + q_1 + q_2) \\
&- g_S^2 t_{ll'}^a t_{l'k}^b \delta^{fs} S_{p_1}^0 \int_{q_1} \slashed{A}_{q_1}^a S_{p_1+q_1}^0 \int_{q_2} \slashed{A}_{q_2}^b S_{p_1+q_1+q_2}^0 (2\pi)^4 \delta^4(p_1 - p_2 + q_1 + q_2) \\
&- ie\hat{Q}^{fs} g_S^2 t_{ll'}^a t_{l'k}^b \Big( S_{p_1}^0 \int_{q_1} \slashed{A}_{q_1} S_{p_1+q_1}^0 \int_{q_2} \slashed{A}_{q_2}^a S_{p_1+q_1+q_2}^0 \int_{q_3} \slashed{A}_{q_3}^b S_{p_1+q_1+q_2+q_3}^0 \\
&\qquad + S_{p_1}^0 \int_{q_1} \slashed{A}_{q_1}^a S_{p_1+q_1}^0 \int_{q_2} \slashed{A}_{q_2} S_{p_1+q_1+q_2}^0 \int_{q_3} \slashed{A}_{q_3}^b S_{p_1+q_1+q_2+q_3}^0 \\
&\qquad + S_{p_1}^0 \int_{q_1} \slashed{A}_{q_1}^a S_{p_1+q_1}^0 \int_{q_2} \slashed{A}_{q_2}^b S_{p_1+q_1+q_2}^0 \int_{q_3} \slashed{A}_{q_3} S_{p_1+q_1+q_2+q_3}^0 \Big) \\
&\qquad \times (2\pi)^4 \delta^4\Big(p_1 - p_2 + \sum_{i=1,2,3} q_i\Big) \,.
\end{aligned}
\tag{56}
$$

Note that the Dirac delta is always under the effect of the integrals to its left. Additionally, we use the convention:

$$S^0(p) = i\frac{\not{p} + m}{p^2 - m^2 + i\epsilon} \equiv S^0_p \,. \tag{57}$$

We have kept the momentum conservation delta of the $y$ vertex explicitly for $S(p_1, p_2)$ in order to make the relation with $S(p)$ and $\tilde{S}(p)$ more evident. At this point, we can use the expansions of the gauge fields in (49). Since the local operators considered for the OPE contain no derivatives of the photon fields and contain only up to one derivative of the gluon field, it is enough to retain the first term of the expansion for the photon field and the first two terms for the gluon. Therefore, in the momentum representation, we can use the replacement:

$$a^\mu(q) = \frac{i}{2}(2\pi)^4 f^{\nu\mu}(0)\frac{\partial}{\partial q^\nu}\delta^4(q) = -\frac{i}{2}(2\pi)^4 \delta^4(q) f^{\nu\mu}(0)\frac{\partial}{\partial q^\nu}$$

$$a^{a\mu}(q) = (2\pi)^4 \delta^4(q)\left(-\frac{i}{2}f^{a\nu\mu}(0)\frac{\partial}{\partial q^\nu} - \frac{1}{3}D^\tau f^{a\nu\mu}(0)\frac{\partial}{\partial q^\nu}\frac{\partial}{\partial q^\tau}\right)\,. \tag{58}$$

These expressions were derived for soft insertions where momentum is leaving the diagram. From the distributional point of view, the derivatives are supposed to act on test functions [101]. This yields the following result:

$$\begin{aligned}
S^{fs}_{lk}(p_1, p_2) = (2\pi)^4 \Bigg(& S^0_{p_1}\delta^{fs}\delta_{lk}\delta^4(p_1 - p_2) \\
&+ \frac{1}{2}\{e\hat{Q}^{fs}\delta_{lk}f^{\mu_1\nu_1}(0) + g_s t^a_{lk}\delta^{fs}f^{a\mu_1\nu_1}(0)\}\frac{\partial}{\partial q_1^{\mu_1}}S^0_{p_1}\gamma_{\nu_1}S^0_{p_1+q_1}\delta^4(p_1 - p_2 + q_1) \\
&- \frac{i}{3}g_s t^a_{lk}\delta^{fs}\overline{D}^\tau f^{a\mu_1\nu_1}\frac{\partial}{\partial q_1^\tau}\frac{\partial}{\partial q_1^{\mu_1}}S^0_{p_1+q_1}S^0_{p_1}\gamma_{\nu_1}\delta^4(p_1 - p_2 + q_1) \\
&+ \frac{1}{4}e\hat{Q}^{fs}g_s t^a_{lk}\left(f^{\mu_1\nu_1}f^{a\mu_2\nu_2} + f^{a\mu_1\nu_1}f^{\mu_2\nu_2}\right) \\
&\quad\times \frac{\partial}{\partial q_1^{\mu_1}}\frac{\partial}{\partial q_2^{\mu_2}}\left(S^0_{p_1}\gamma_{\nu_1}S^0_{p_1+q_1}\gamma_{\nu_2}S^0_{p_1+q_1+q_2}\delta^4(p_1 - p_2 + q_1 + q_2)\right)\Bigg|_{q_{1,2}=0} \\
&+ \frac{1}{4}g_s^2 t^a_{ll'}t^b_{l'k}\delta^{fs}\left(f^{a\mu_1\nu_1}f^{b\mu_2\nu_2}\right) \\
&\quad\times \frac{\partial}{\partial q_1^{\mu_1}}\frac{\partial}{\partial q_2^{\mu_2}}\left(S^0_{p_1}\gamma_{\nu_1}S^0_{p_1+q_1}\gamma_{\nu_2}S^0_{p_1+q_1+q_2}\delta^4(p_1 - p_2 + q_1 + q_2)\right)\Bigg|_{q_{1,2}=0} \\
&+ \frac{1}{8}e\hat{Q}^{fs}g_s^2 t^a_{lk'}t^b_{k'k}\left(f^{\mu_1\nu_1}f^{a\mu_2\nu_2}f^{b\mu_3\nu_3} + f^{a\mu_1\nu_1}f^{\mu_2\nu_2}f^{b\mu_3\nu_3} + f^{a\mu_1\nu_1}f^{b\mu_2\nu_2}f^{\mu_3\nu_3}\right) \\
&\quad\times \frac{\partial}{\partial q_1^{\mu_1}}\frac{\partial}{\partial q_2^{\mu_2}}\frac{\partial}{\partial q_3^{\mu_3}}\left(S^0_{p_1}\gamma_{\nu_1}S^0_{p_1+q_1}\gamma_{\nu_2}S^0_{p_1+q_1+q_2}\gamma_{\nu_3}S^0_{p_1+q_1+q_2+q_3}\right. \\
&\quad\quad\times \left.\delta^4(p_1 - p_2 + \sum_{i=1,2,3}q_i)\right)\Bigg|_{q_{1,2,3}=0}\Bigg)\,. \tag{59}
\end{aligned}$$

This expression for the free quark propagator can be understood as an expansion in terms of diagrams (see Figure 5) with increasing number of (background) gauge bosons.

The "free" gluon propagator contains interactions with the background gluon fields in a similar but more involved way than the quark one. However, at the order that we are interested in, there appear no such propagators, so we refer the interested reader to see the details of the computation in Section 2 of [102].

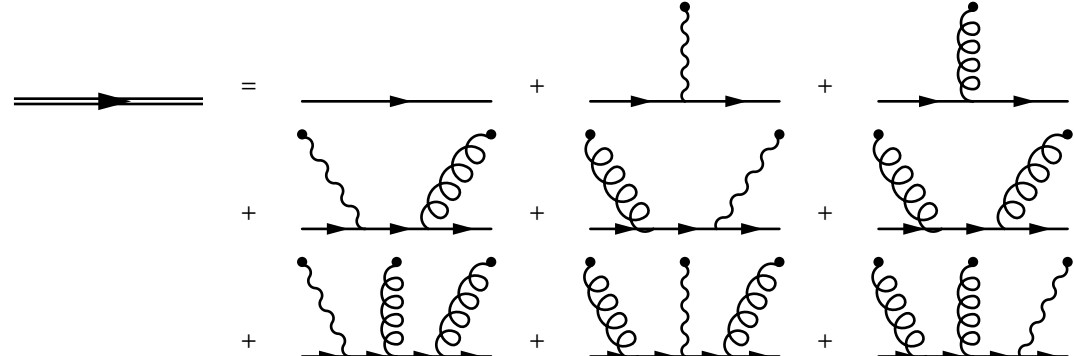

**Figure 5.** This figure shows the expansion of the free quark propagator in a background of gauge fields in terms of diagrams with interactions with gluons and photons that are created/annihilated in the vacuum. The order in which diagrams appear in the sum corresponds to the order of terms in Equation (59).

## 4. Computation of Un-Renormalized Wilson Coefficients

In the previous section, we obtained expressions for the quark and gluon fluctuation propagators which contained background insertions of vacuum expectation values (VEVs) such as $f^{a\mu\nu}$ and $f^{\mu\nu}$. Furthermore, we saw that vertices from the Dyson series also introduce VEVs of quark operators, thus giving us all the tools required to build the OPE of $\Pi^{\mu_1\mu_2\mu_3}$ with background fields and find the Wilson coefficients that require the computation of $\Pi_F^{\mu_1\mu_2\mu_3\mu_4\mu_5}$. Concerning the actual computation of Wilson coefficients, let us start by considering the one related to $S_{1,\mu\nu} = ee_f F_{\mu\nu}$. This term represents the configuration in which hard momenta travel through all internal lines of the diagrams; thus, there are no cut lines. The leading order contribution for this configuration is given by the quark loop (see Figure 6), where different contributions are obtained by inserting the soft photon in different sides of the triangle and/or inverting the orientation of the loop. Since $S_{1,\mu\nu}$ is the operator with the lowest dimension in the OPE, its Wilson coefficient is expected to give the most relevant contribution to $\Pi_F^{\mu_1\mu_2\mu_3\mu_4\mu_5}$ and therefore to $a_\mu$. When splitting the quark fields in the currents of $\Pi^{\mu_1\mu_2\mu_3}$, this diagram comes from the term that contains only quantum fluctuations. In the end, the contribution from the Wilson coefficient of $S_{1,\mu\nu}$ to $\Pi_F^{\mu_1\mu_2\mu_3\mu_4\mu_5}$ is:

$$\Pi_{F(S_1)}^{\mu_1\mu_2\mu_3\mu_4\nu_4} = i\frac{N_c}{2}\int\frac{d^4p}{(2\pi)^4}\sum_f e_f^4\frac{\partial}{\partial q_{4\nu_4}}\sum_{\sigma(1,2,4)}\text{Tr}\left\{\gamma^{\mu_3}S^0(p+q_1+q_2+q_4)\gamma^{\mu_4}\right.$$
$$\left.\times S^0(p+q_1+q_2)\gamma^{\mu_1}S^0(p+q_2)\gamma^{\mu_2}S^0(p)\right\}\Bigg|_{q_4=0}, \tag{60}$$

where $N_c$ is the number of quark colors, $f$ represents quark flavor, and $\sigma(1,2,4)$ represents a permutation over the set $\{(q_i,\mu_i)|\,i\in\{1,2,4\}\}$. The derivative with respect to the soft photon momentum can be traced back to (59). Note that the propagator depends implicitly on the quark flavor through the masses, assumed to be all equal for the light flavors considered. Furthermore, the effect of the derivative on the propagators is to duplicate them:

$$\lim_{q_4\to0}\frac{\partial}{\partial q_{\nu_4}}S(p+q_4) = i\lim_{q_4\to0}S^0(p+q_4)\gamma^{\nu_4}S(p+q_4) = iS^0(p)\gamma^{\nu_4}S^0(p). \tag{61}$$

The focus of our work is on the contribution from $S_{1,\mu\nu}$, which gives a much larger contribution to $a_\mu$ than any other operator, so we will not discuss the contributions from other operators in detail. Instead, we give a brief overview of the computation. We refer the interested reader to [60].



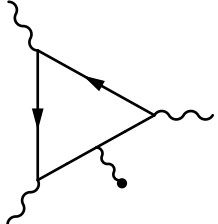

**Figure 6.** Representative diagram of the leading order contribution to the Wilson coefficient of $S_{1,\mu\nu}$ in the OPE of $\Pi^{\mu_1\mu_2\mu_3}$. The black dot represents creation/annihilation of a line by the background fields in the vacuum.

Contributions with one cut quark line and at most one soft gauge boson insertion ($S_{2-5,\mu\nu}$ and $S_{7,\mu\nu}$) are obtained at leading order from the diagrams in Figure 7. Their corresponding amplitudes are computed from terms in $\Pi^{\mu_1\mu_2\mu_3}$ that contain two soft quark fields and require no vertices from the Dyson series expansion. Soft gluon or photon insertions on quark hard lines, if necessary, come from propagators of quark fluctuations, as seen in the previous section.

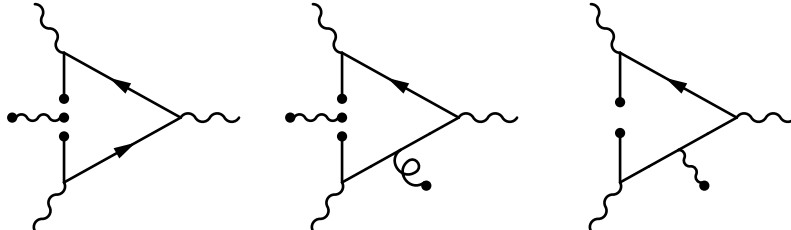

**Figure 7.** Representative diagrams of the leading order contribution to the Wilson coefficient of $S_{2,\mu\nu}$ (first diagram), $S_{3,4,7,\mu\nu}$ (second diagram), and $S_{5,\mu\nu}$ (third diagram) in the OPE of $\Pi^{\mu_1\mu_2\mu_3}$. The black dot represents creation/annihilation of a line by the background fields in the vacuum.

Let us now consider the operator with two cut gluon lines, that is, $S_{6,\mu\nu}$. Diagrams contributing to this operator are very similar to the quark loop of $S_{1,\mu\nu}$, but they have two soft gluon insertions (see Figure 8). As with the first quark loop, these insertions must be permuted in all possible ways to obtain the full contribution.

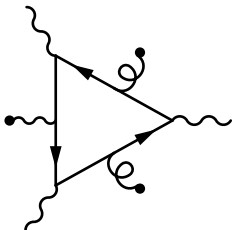

**Figure 8.** Representative diagram of the leading order contribution to the Wilson coefficient of $S_{6,\mu\nu}$ in the OPE of $\Pi^{\mu_1\mu_2\mu_3}$. The black dot represents creation/annihilation of a line by the background fields in the vacuum.

With regards to operators with four quark background insertions ($S_{8,\mu\nu}$), diagrams that contribute to the Wilson coefficients of this operator correspond to the quark loop with two cut quark lines; therefore, the diagram is divided into two parts, which have to be connected by a gluon (see Figure 9). There are six different ways in which the virtual gluon line can connect the two parts of the diagram and all have to be accounted for. The corresponding two gluon-quark vertices are responsible for the $\alpha_S$ coefficient of $S_{8,\mu\nu}$.

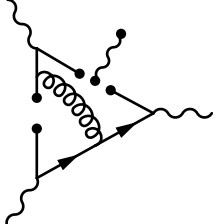

**Figure 9.** Representative diagram of the leading order contribution to the Wilson coefficient of $S_{8,\mu\nu}$ in the OPE of $\Pi^{\mu_1\mu_2\mu_3}$. The black dot represents creation/annihilation of a line by the background fields in the vacuum.

Except for $S_{1,\mu\nu}$, contributions from all operators to $\Pi_F^{\mu_1\mu_2\mu_3\mu_4\mu_5}$ depend on the susceptibilities $X_i^S$. By definition, these are non-perturbative quantities which are usually computed either by lattice, models, and/or educated guesses. The most well-known one is $X_5$ because it is related to the quark condensate, which is a common subject of study in lattice computations. We have suppressed the $S$ index because the elements of the OPE need renormalization and therefore a new set of susceptibilities $X_i$ is defined in terms of the renormalized operators, as we will see in the next section. A more recent version of the review cited in [60] can be found in [103], where Figure 14, Table 22 and references therein represent a thorough compilation of results for the quark condensate. The rest of the susceptibilities are not so well-known and their numerical values are estimated in [60] by a combination of models and educated guesses.

Up to this point, we have presented all un-renormalized Wilson coefficients associated with operators in (47) and, more importantly, their contribution to $\partial^{\mu_5}\Pi^{\mu_1\mu_2\mu_3\mu_4}$. Computation of the Wilson coefficients is, however, not yet complete, for renormalization of the OPE elements has not been taken into account. In contrast to the usual situation in perturbative computations, we have not encountered ultraviolet divergences in the Wilson coefficients of this section. In fact, except for $S_{1,\mu\nu}$ and $S_{6,\mu\nu}$, all of their leading order contributions are at the tree level. As we will see in the next sections for the quark loop, the Wilson coefficients of these two operators, although finite, have infrared contributions that are renormalized by the quark masses. Such singularities scale as logarithms and negative powers of $m_f$. These singular terms are problematic in a twofold way. From a computational perspective, these singular factors may spoil convergence of the perturbative computation when the momenta of the process, namely $Q_i$, get much bigger that the mass scale of the quarks, which is actually our situation. From a conceptual point of view, it is also questionable to have Wilson coefficients with infrared contributions; in the OPE framework, they are meant to represent the contribution from the parts of the diagram through which the very high external momenta travel. In the next section, we will present how renormalization of the product of background fields "cures" these infrared divergences and thus completes the separation of low- and high-energy contributions of the OPE.

## 5. OPE of $\Pi^{\mu_1\mu_2\mu_3\mu_4}$ in an Electromagnetic Background Field: Renormalization

In this section, we will present the renormalization program for the operators that form the OPE for $\Pi^{\mu_1\mu_2\mu_3}$ in the $\overline{MS}$ scheme.

The Wilson coefficients that were presented in the previous section are UV finite at the computed order, but they do have infrared-divergent terms such as $1/m_f^2$ and $\ln\{Q_i^2/m_f^2\}$ that are regularized by the quark masses. These terms may spoil the convergence of the perturbative expansion. Moreover, the Wilson coefficients should not have infrared contributions in the first place; therefore, it should be possible to safely compute them in the massless quark limit. There is an additional kind of low-energy effect that may affect Wilson coefficients; the ones arising from diagrams where soft quark and gluon lines receive self-energy corrections. This does not apply for photon soft lines, since we do not consider photon fluctuations. For example, Figure 10 shows how such divergences can arise in diagrams that contribute to the Wilson coefficient of $S_{2,\mu\nu}$. The gray blob of

Figure 10 involves a perturbative series in $\alpha_S$ at zero momentum, which of course does not converge since the processes it is trying to describe belong to the non-perturbative domain. These diagrams, however, do not appear at the order we are considering, and therefore, we will not discuss them in detail.

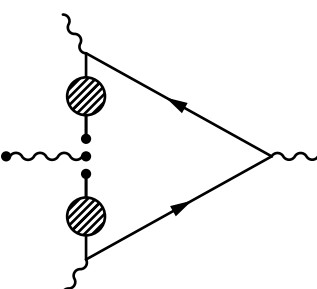

**Figure 10.** Diagram with infrared divergences affecting the Wilson coefficient of $S_{2,\mu\nu}$ in the OPE of $\Pi^{\mu_1\mu_2\mu_3}$. The black dot represents creation/annihilation of a line by the background fields in the vacuum. The shaded blob represents self-energy corrections to the soft quark line.

The prescription of a renormalization program in this context is not very surprising considering that the OPE is built from composite operators which are known to require counterterms of their own to be renormalized. Therefore, one could expect that after carrying out the renormalization of these composite operators, the infrared divergences of the Wilson coefficients are cancelled. In [60], renormalization was performed in the full-field framework by dressing operators $S_{i,\mu\nu}$, that is, by inserting them into the Dyson series expansion.

However, from the point of view of the background field method that we have followed in the previous sections, these operators are simply products of classical background fields, so at first, it may seem rather odd to assert that they require renormalization. Nevertheless, this is not at all surprising if we trace the step back to the third section. There it was argued that Green functions in the "original" theory could be obtained from the background theory by functionally differentiating vacuum-to-vacuum amplitudes with respect to background fields. Thus, products of background fields were converted into Green functions. Insertions of composite operators, however, work differently, because they have their own source terms in the generating functional of the background theory. In contrast to the fields, whose source terms only involved the fluctuation part of the field, namely $J_n\phi'_n$, the insertion of a composite operator $O_i$ involves the full fields, that is, $L_i O_i(\{\phi + \phi'\})$, where $L_i$ is the source. This is required in order for the relation between the effective action and its background counterpart to remain valid. For example, the insertion of $S_{2,\mu\nu}$ in the background framework actually involves:

$$(\overline{\psi} + \overline{\psi}')\sigma_{\mu\nu}(\psi + \psi') \tag{62}$$

instead of just:

$$\overline{\psi}\sigma_{\mu\nu}\psi . \tag{63}$$

This does not mean that computation of Wilson coefficients of the third and fourth sections is wrong, for the operator in (63) is the one that is related to the matrix element we are interested in. We will later see that they can be related to the renormalized composite operator in a straightforward manner. Instead, this means that operator mixing is naturally ingrained in the background field formalism. This also means that it is the operator in (62) that needs renormalization, regardless of whether it ends up curing divergences in the Wilson coefficients of the operator in (63) as well.

After justifying the need for renormalization in the background theory, now we can proceed to apply it to the operators in (47). In our renormalization program, composite operators are inserted in Green's functions, which then are computed using dimensional regularization to preserve gauge invariance. Finally, the relation between singular terms

and counterterms is defined by modified minimal subtraction $\overline{MS}$. As was mentioned earlier, counterterms required for renormalization of composite operators are a linear combination that includes other composite operators with singular coefficients. This is referred to as "operator mixing". For simplicity, we will compute Green's functions with an insertion of each composite operator and no other fields involved, for otherwise, additional singularities renormalized by the Lagrangian's counterterms would appear. However, not just *any* operators can mix under renormalization. Only operators with the same quantum numbers can. Furthermore, since the background field method does not break background gauge invariance, then this means that mixing also respects gauge invariance (as long as the Green's function in which it is inserted has only background quark and gauge fields) [104–106]. In the end, this means that the operators that form the OPE of $\Pi^{\mu_1\mu_2\mu_3}$ mix among themselves under renormalization.

In our context, this means that the renormalization of the elements of our OPE will have the following shape:

$$Q^0_{\mu\nu} = \hat{Z} Q_{\mu\nu} \qquad Q^0_{i,\mu\nu} = Q^0_{i,\mu\nu}(\psi + \psi', a^{a\mu} + A^{'a\mu}, a^\mu) \,, \tag{64}$$

where $Q^0_{\mu\nu}$ represents the vector whose components are the bare elements of the OPE of (47) and it is a function of the full fields, that is, the sum of the background and fluctuation parts. $Q_{\mu\nu}$ contains its renormalized versions. Consequently, $\hat{Z}$ is an $8 \times 8$ matrix containing constants with regularized ultraviolet divergences. As we will see in the following, the vector of operators $Q_{\mu\nu}$ does not coincide with the $S_{\mu\nu}$ that we defined earlier, but they are related by a constant matrix whose elements contain regularized infrared divergences. Consequently, one can define:

$$Q_{\mu\nu} = \hat{U} S_{\mu\nu} \implies Q^0_{\mu\nu} = \hat{Z}\hat{U} S_{\mu\nu} \,. \tag{65}$$

Renormalization is used to separate contributions coming from different energy scales, and in this case, such an objective is achieved, since the elements in $\hat{U}$ are just the required ones to cancel the infrared contributions of the Wilson coefficients. Furthermore, it is important to note that we could not have avoided singular terms in the Wilson coefficients by using $Q^0_{\mu\nu}$ instead of $S_{\mu\nu}$, since in such a case, we would have traded infrared for ultraviolet contributions. Instead, it is necessary to use renormalization to successfully separate low- and high-energy contributions and find $Q_{\mu\nu}$. The renormalized Wilson coefficients $C$ are free of infrared contributions and are defined in terms of the bare ones $C_S$ as:

$$\Pi^{\mu_1\mu_2\mu_3} = C_S^{\mu_1\mu_2\mu_3\mu_4\mu_5} \cdot \langle 0|S_{\mu\nu}|\gamma\rangle \equiv C^{\mu_1\mu_2\mu_3\mu_4\mu_5} \cdot \langle 0|Q_{\mu\nu}|\gamma\rangle \tag{66}$$

$$\implies C^{\mu_1\mu_2\mu_3\mu_4\mu_5} = (\hat{U}^{-1})^T C_S^{\mu_1\mu_2\mu_3\mu_4\mu_5} \,. \tag{67}$$

Note that renormalized susceptibilities $X$ can also be defined for $Q_{\mu\nu}$ and can be related to the un-renormalized ones $X_i^S$ in a straightforward way:

$$Q_{\mu\nu} \equiv X F_{\mu\nu} \implies X = \hat{U} X^S \,. \tag{68}$$

As always, it is of course necessary to specify an order at which renormalization constants will be truncated. The appearance of non-perturbative matrix elements in the OPE introduces non-perturbative expansion parameters ($\Lambda_{QCD}/Q$) in addition to the perturbative ones ($g_S$ and $e$). In terms of the latter, the cut-off is placed at $O(e^{-1}g_S^2)$. With respect to the former, we have $O(\Lambda_{QCD}^6/Q^6)$, which, in addition to gauge invariance conservation of the background field theory, essentially means that operators $S_{i,\mu\nu}$ only mix among themselves. Since we are considering the three lightest quarks, its masses' effects can be regarded as perturbations as well, thus introducing another expansion parameter

$m_f / \Lambda_{QCD}$. Nevertheless, we can obtain the full dependence of the mixing coefficients on the quarks' masses.

It is important to note that perturbative and non-perturbative parameters must not be regarded independently. The mixing matrix $\hat{U}$ is meant to modify the Wilson coefficients, as shown in the previous equation; therefore, each element must be expanded up to the order of the Wilson coefficients which it modifies. This introduces an interplay between the dimension of the operators that are mixing and the order of their Wilson coefficients. The precise implications of this assertion should become more clear throughout the rest of this section.

Now we are ready to put the renormalization program we just described to use. For $S_{1,\mu\nu}$, renormalization is at its simplest. Since the photon field does not have quantum fluctuations, then $Q^0_{1,\mu\nu}$ is just equal to $S_{1,\mu\nu}$, and hence, it cannot mix with any other operator.

### 5.1. Mixing of the $Q^0_{2,\mu\nu}$ Operator

The first and most non-trivial case is $Q_{2,\mu\nu}$. A Green's function with a full-field insertion of this composite operator is given by:

$$\langle 0|Q^0_{2,\mu\nu}|\gamma\rangle = \overline{\psi}\sigma_{\mu\nu}\psi + \langle 0|\overline{\psi}\sigma_{\mu\nu}\psi'|\gamma\rangle + \langle 0|\overline{\psi}'\sigma_{\mu\nu}\psi|\gamma\rangle + \langle 0|\overline{\psi}'\sigma_{\mu\nu}\psi'|\gamma\rangle \,, \tag{69}$$

where we are evaluating the matrix element of a Heisenberg operator, and therefore, the Dyson series of interaction vertices has to be inserted. Mixing with $S_{1,\mu\nu}$ can only come from the fourth term and it requires the contraction of both quark fluctuations and a soft insertion of the photon field in the resulting propagator. As we will see later in this section, further soft insertions lead to mixing with other operators. Since the Wilson coefficient of $S_{2,\mu\nu}$ is $O(e^{-1}g_S^0)$ and the mixing coefficient is $O(e^0 g_S^0)$, then the net mixing contribution is of order $O(e^{-1}g_S^0)$, already the same as the Wilson coefficient of $S_{1,\mu\nu}$. Therefore, we can cut off the mixing coefficient at this point. The result is:

$$\langle 0|\overline{\psi}'\sigma_{\mu\nu}\psi'|\gamma\rangle = -\operatorname{Tr}\{S^{ff}_{ll}(0,0)\sigma_{\mu\nu}\} = \frac{e\mu^{2\epsilon}}{2}f^{\mu_1\nu_1}\int\frac{d^d p_1}{(2\pi)^d}\frac{\partial}{\partial q_1^{\mu_1}}\operatorname{Tr}\{S^0_{p_1}\gamma_{\nu_1}S^0_{p_1+q_1}\sigma_{\mu\nu}\}\Big|_{q_1=0}$$

$$= 4iN_c e\mu^{2\epsilon}e_f m_f f_{\mu\nu}\int\frac{d^d p_1}{(2\pi)^d}\frac{1}{[p_1^2 - m_f^2]^2}$$

$$= -\frac{N_c e e_f}{4\pi^2}m_f f_{\mu\nu}\Gamma(\epsilon)\left(\frac{4\pi\mu^2}{m_f^2}\right)^\epsilon \,, \tag{70}$$

where $d \equiv 4 - 2\epsilon$ is the shifted dimension, $\mu$ is the mass parameter that carries the mass dimension of $e$ in the regularized theory, and we have used the well-known formula:

$$\int\frac{d^d p}{(2\pi)^d}\frac{1}{[p^2 - \Delta]^n} = \frac{(-1)^n}{(4\pi)^{d/2}}i\frac{\Gamma(n-\frac{d}{2})}{\Gamma(n)}\left(\frac{1}{\Delta}\right)^{n-\frac{d}{2}} \,. \tag{71}$$

Note that the derivation of this formula involves a Wick rotation of the integration variable; therefore, it is necessary to ensure that $\Delta$ is positive, as it of course is in (70). Otherwise, the integrand acquires a discontinuity when the norm of the spatial momentum $p^2$ becomes smaller than the absolute value $|\Delta|$. This can be accounted for by giving the pole a vanishing imaginary part, $\Delta \longrightarrow \Delta - i0^+$, which plays a role analogous to the Feynman prescription for free propagators.

Turning back to (70), it is necessary to expand the result around $\epsilon = 0$ to expose the singular terms. The result when one discards terms that vanish when $\epsilon \to 0$ is:

$$\langle 0|\overline{\psi}'\sigma_{\mu\nu}\psi'|\gamma\rangle = -\frac{N_c}{4\pi^2}m_f\left(\frac{1}{\hat{\epsilon}} + \ln\left\{\frac{\mu^2}{m_f^2}\right\}\right)S_{1,\mu\nu} \,. \tag{72}$$

where $\gamma_E \approx 0.5772$ is the Euler–Mascheroni constant and we have used the expansion of the gamma function around $\epsilon \to 0$:

$$\Gamma(\epsilon) = \frac{1}{\epsilon} - \gamma_E + O(\epsilon) . \tag{73}$$

As was mentioned previously, we define the singular term to be subtracted in the $\overline{MS}$ scheme:

$$\frac{1}{\hat{\epsilon}} \equiv \frac{1}{\epsilon} + \ln\{4\pi\} - \gamma_E. \tag{74}$$

This mixing coefficient states two facts about the insertions of the operator $Q_{2,\mu\nu}$. (1) A part of their singular ultraviolet behavior can be effectively renormalized by mixing with $S_{1,\mu\nu}$, and (2) they involve a low-energy contribution from $S_{1,\mu\nu}$, which is represented by the mass logarithm. This is an explicit example of the appearance of the $\hat{Z}$ and $\hat{U}$ matrices that were defined in (65).

The leading order contribution to the mixing of $Q_{2,\mu\nu}$ with $S_{3,4,5,\mu\nu}$ and $S_{7,\mu\nu}$ is given by the second and third terms on the right-hand side of (69). They require the introduction two quark-gluon vertices. However, Wilson coefficients of these operators are all $O(e^{-1})$, so the mixing coefficient cannot receive perturbative corrections from interaction vertices at the relevant order. The same analysis of course applies the other way around; thus, $S_{2-5,\mu\nu}$ and $S_{7,\mu\nu}$ do not mix with each other at the order of our computation.

The mixing of $Q_{2,\mu\nu}$ with $S_{6,\mu\nu}$ can of course only come from the fourth term in (69). Just as with the mixing with $S_{1,\mu\nu}$, we can contract both fluctuations without introducing interaction vertices and then three soft insertions can be performed on the propagator. The corresponding result is:

$$\langle 0|\overline{\psi}'\sigma_{\mu\nu}\psi'|\gamma\rangle = -\frac{ee_f g_S^2}{8} \mathrm{Tr}\{t^a t^b\} \left( f^{\mu_1\nu_1} f^{a\mu_2\nu_2} f^{b\mu_3\nu_3} + f^{a\mu_1\nu_1} f^{\mu_2\nu_2} f^{b\mu_3\nu_3} + f^{a\mu_1\nu_1} f^{b\mu_2\nu_2} f^{\mu_3\nu_3} \right)$$

$$\times \int \frac{d^4p}{(2\pi)^4} \frac{\partial}{\partial q_1^{\mu_1}} \frac{\partial}{\partial q_2^{\mu_2}} \frac{\partial}{\partial q_3^{\mu_3}} \mathrm{Tr}\{S_p^0 \gamma_{\nu_1} S_{p+q_1}^0 \gamma_{\nu_2} S_{p+q_1+q_2}^0 \gamma_{\nu_3} S_{p+q_1+q_2+q_3}^0 \sigma_{\mu\nu}\}\Big|_{q_{1,2,3}=0} \Big)$$

$$= -\frac{1}{72 m_f^3} S_{6,\mu\nu} . \tag{75}$$

Finally, the mixing coefficient of $Q_{2,\mu\nu}$ with $S_{8,\mu\nu}$ is evidently beyond the perturbative order that we are interested in.

### 5.2. Mixing of the $Q_{3,\mu\nu}^0$ Operator

For this operator, we have:

$$\langle 0|Q_{3,\mu\nu}^0|\gamma\rangle = -g_S\langle 0|\overline{\psi}t^a G_{\mu\nu}^a(a+A')\psi|\gamma\rangle - g_S\langle 0|\overline{\psi}t^a G_{\mu\nu}^a(a+A')\psi'|\gamma\rangle$$
$$- g_S\langle 0|\overline{\psi}'t^a G_{\mu\nu}^a(a+A')\psi|\gamma\rangle - g_S\langle 0|\overline{\psi}'t^a G_{\mu\nu}^a(a+A')\psi'|\gamma\rangle , \tag{76}$$

where $G_{\mu\nu}^a(a+A')$ is the gluon field strength tensor separated between fluctuation and background parts, namely:

$$G_{\mu\nu}^a(a+A') = f_{\mu\nu}^a + \overline{D}_\mu A_\nu'^a - \overline{D}_\nu A_\mu'^a + g_S f^{abc} A_\mu'^b A_\nu'^c , \tag{77}$$

as was introduced in the previous sections. All the Wilson coefficients of the previous section were computed up to $O(e^{-1}g_S^0)$; therefore, no terms with gluon quantum fluctuations give relevant contributions to the mixing and we can replace $G_{\mu\nu}^a \longrightarrow f_{\mu\nu}^a$ in $\langle Q_{3,\mu\nu}^0\rangle$. This means that at the order that is relevant for us, $Q_{3,\mu\nu}^0$ can only mix with operators that have at least one soft insertion of $f^{a\mu\nu}$. From (47), the only compatible one is $S_{6,\mu\nu}$. In principle, $S_{7,\mu\nu}$ is compatible as well, but to obtain terms with covariant derivatives, it is necessary

to introduce additional $g_S^2$ factors that take the mixing beyond the established cutoff. The leading order contribution to that mixing coefficient is given by the term in (76) with two quark fluctuations when they are contracted with each other and one soft gluon and one soft photon insertion are performed on the resulting quark propagator:

$$-\langle 0|\overline{\psi}' t^a f_{\mu\nu}^a \psi'|\gamma\rangle = \frac{1}{4} ee_f f_{\mu\nu}^a g_S^2 \operatorname{Tr}\{t^a t^b\} \left( f^{\mu_1\nu_1} f^{b\mu_2\nu_2} + f^{b\mu_1\nu_1} f^{\mu_2\nu_2} \right)$$

$$\times \int \frac{d^4 p}{(2\pi)^4} \frac{\partial}{\partial q_1^{\mu_1}} \frac{\partial}{\partial q_2^{\mu_2}} \operatorname{Tr}\left\{ S_p^0 \gamma_{\nu_1} S_{p+q_1}^0 \gamma_{\nu_2} S_{p+q_1+q_2}^0 \right\}\bigg|_{q_{1,2}=0} \tag{78}$$

$$= \frac{1}{36 m_f} S_{6,\mu\nu} \ .$$

*5.3. Mixing of the $Q_{4,\mu\nu}^0$ Operator*

For $Q_{4,\mu\nu}^0$ we have:

$$\langle 0|Q_{4,\mu\nu}^0|\gamma\rangle = -g_S \langle 0|\overline{\psi} t^a \overline{G}_{\mu\nu}^a (a+A')\gamma_5 \psi|\gamma\rangle - g_S \langle 0|\overline{\psi} t^a \overline{G}_{\mu\nu}^a (a+A')\gamma_5 \psi'|\gamma\rangle$$
$$- g_S \langle 0|\overline{\psi}' t^a \overline{G}_{\mu\nu}^a (a+A')\gamma_5 \psi|\gamma\rangle - g_S \langle 0|\overline{\psi}' t^a \overline{G}_{\mu\nu}^a (a+A')\gamma_5 \psi'|\gamma\rangle \ , \tag{79}$$

where $\overline{G}^{a\mu\nu}(a+A')$ is the dual of $G^{a\mu\nu}(a+A')$, that is, $\overline{G}^{a\mu\nu} = \frac{i}{2}\epsilon^{\mu\nu\alpha\beta}G_{\alpha\beta}^a$. The analysis of the relevant mixing coefficients at the order of interest is essentially the same as for $Q_{3,\mu\nu}^0$; therefore, there is mixing only with $S_{6,\mu\nu}^0$ and the corresponding coefficient is:

$$-\langle 0|\overline{\psi}' t^a \overline{f}_{\mu\nu}^a \gamma_5 \psi'|\gamma\rangle = \frac{1}{4} ee_f \overline{f}_{\mu\nu}^a g_S^2 \operatorname{Tr}\{t^a t^b\} \left( f^{\mu_1\nu_1} f^{b\mu_2\nu_2} + f^{b\mu_1\nu_1} f^{\mu_2\nu_2} \right)$$

$$\times \int \frac{d^4 p}{(2\pi)^4} \frac{\partial}{\partial q_1^{\mu_1}} \frac{\partial}{\partial q_2^{\mu_2}} \operatorname{Tr}\left\{ S_p^0 \gamma_{\nu_1} S_{p+q_1}^0 \gamma_{\nu_2} S_{p+q_1+q_2}^0 \gamma_5 \right\}\bigg|_{q_{1,2}=0}$$

$$= \frac{1}{24 m_f} S_{6,\mu\nu} \ . \tag{80}$$

*5.4. Mixing of the $Q_{5,\mu\nu}^0$ Operator*

For $Q_{5,\mu\nu}^0$ we have:

$$\frac{1}{ee_f}\langle 0|Q_{5,\mu\nu}^0|\gamma\rangle = \langle 0|\overline{\psi}\psi F_{\mu\nu}|\gamma\rangle + \langle 0|\overline{\psi}\psi' F_{\mu\nu}|\gamma\rangle + \langle 0|\overline{\psi}' F_{\mu\nu}\psi|\gamma\rangle + \langle 0|\overline{\psi}'\psi' F_{\mu\nu}|\gamma\rangle \ . \tag{81}$$

In a similar fashion as with $Q_{3,\mu\nu}$ and $Q_{4,\mu\nu}$, the soft photon insertion allows only for mixing with $S_{1,\mu\nu}$ and $S_{6,\mu\nu}$. The leading order contribution to both mixing coefficients is once again given by the term in (81) with two quark fluctuations:

$$\langle 0|\overline{\psi}'\psi' ee_f F_{\mu\nu}|\gamma\rangle = -\operatorname{Tr}\{S_{ll}^{ff}(0,0)\} ee_f f_{\mu\nu} \ . \tag{82}$$

Mixing with $S_{1,\mu\nu}$ is obtained by performing no soft insertions in the quark propagator:

$$\langle 0|\overline{\psi}'\psi' ee_f F_{\mu\nu}|\gamma\rangle = -\frac{m_f^3}{4\pi^2} \left( \frac{1}{\hat{\epsilon}} + \ln\left\{ \frac{\mu^2}{m_f^2} \right\} + 1 \right) S_{1,\mu\nu} \ , \tag{83}$$

As mentioned in [60], this mixing coefficient can be used to subtract the low-energy contributions to the $O(m_f^4)$ correction to the massless part of the quark loop. On the other hand, mixing with $S_{6,\mu\nu}$ is obtained by inserting two soft gluons on the quark propagator:

$$\langle 0|\overline{\psi}'\psi ee_f F_{\mu\nu}|\gamma\rangle = -\frac{1}{4}ee_f f_{\mu\nu}g_S^2 \operatorname{Tr}\{t^a t^b\}f^{a\mu_1\nu_1}f^{b\mu_2\nu_2}$$

$$\times \int \frac{d^4p}{(2\pi)^4}\frac{\partial}{\partial q_1^{\mu_1}}\frac{\partial}{\partial q_2^{\mu_2}}\operatorname{Tr}\left\{S_p^0\gamma_{\nu_1}S_{p+q_1}^0\gamma_{\nu_2}S_{p+q_1+q_2}^0\right\}\Big|_{q_{1,2}=0} \quad (84)$$

$$= -\frac{1}{12m_f}S_{6,\mu\nu}\ .$$

### 5.5. Mixing of the $Q_{6,\mu\nu}^0$ Operator

For this operator, the same separation of the gluon field strength tensor can be performed that was described for $Q_{3,\mu\nu}$ and $Q_{4,\mu\nu}$. However, since this operator is $O(eg_S^2)$ and all other Wilson coefficients are $O(e^{-1}g_S^0)$, then its mixing coefficients are beyond the perturbative order of the computation.

### 5.6. Mixing of the $Q_{7,\mu\nu}^0$ Operator

With respect to $Q_{7,\mu\nu}^0$ we have:

$$\langle 0|Q_{7,\mu\nu}^0|\gamma\rangle = ig_S\langle 0|\overline{\psi}(t^a G_{\mu\lambda}^a D_\nu + D_\nu t^a G_{\mu\lambda}^a)\gamma^\lambda\psi|\gamma\rangle - (\mu \longleftrightarrow \nu)$$
$$+ ig_S\langle 0|\overline{\psi}(t^a G_{\mu\lambda}^a D_\nu + D_\nu t^a G_{\mu\lambda}^a)\gamma^\lambda\psi'|\gamma\rangle - (\mu \longleftrightarrow \nu)$$
$$+ ig_S\langle 0|\overline{\psi}'(t^a G_{\mu\lambda}^a D_\nu + D_\nu t^a G_{\mu\lambda}^a)\gamma^\lambda\psi'|\gamma\rangle - (\mu \longleftrightarrow \nu)$$
$$+ ig_S\langle 0|\overline{\psi}'(t^a G_{\mu\lambda}^a D_\nu + D_\nu t^a G_{\mu\lambda}^a)\gamma^\lambda\psi'|\gamma\rangle - (\mu \longleftrightarrow \nu)\ , \quad (85)$$

where this time, both the field strength tensor and the covariant derivative are implicitly divided into background and fluctuation parts. Since this operator is already $O(e^0 g_S)$, it can only mix with $S_{3,\mu\nu}$, $S_{4,\mu\nu}$, and $S_{6,\mu\nu}$. Mixing with the first two is relevant only up to terms that do not introduce higher orders of $g_S$; therefore, only the first term in (85) may contribute. However, this would give just the background version $S_{7,\mu\nu}$ of $Q_{7,\mu\nu}$. With respect to the mixing with $S_{6,\mu\nu}$, the leading contribution comes from the last term in (85) when both quark fluctuations are contracted and their corresponding propagator has a soft gluon insertion. For this mixing, only terms that introduce at most another order of $g_S$ are relevant; therefore, only the fully background part of the $t^a G_{\mu\lambda}^a D_\nu$ terms is relevant. In the end, the mixing coefficient between $Q_{7,\mu\nu}$ and $S_{6,\mu\nu}$ is:

$$ig_S\langle 0|\overline{\psi}'(t^a G_{\mu\lambda}^a D_\nu + D_\nu t^a G_{\mu\lambda}^a)\gamma^\lambda\psi'|\gamma\rangle = \frac{i}{2}\mu^{2\epsilon}ee_f g_S^2 \operatorname{Tr}\{t^a t^b\}f_{\mu\lambda}^a\left(f^{\mu_1\nu_1}f^{b\mu_2\nu_2} + f^{b\mu_1\nu_1}f^{\mu_2\nu_2}\right)$$

$$\times \int \frac{d^d p}{(2\pi)^d}ip_\nu\frac{\partial}{\partial q_1^{\mu_1}}\frac{\partial}{\partial q_2^{\mu_2}}\operatorname{Tr}\left\{S_p^0\gamma_{\nu_1}S_{p+q_1}^0\gamma_{\nu_2}S_{p+q_1+q_2}^0\gamma^\lambda\right\}\Big|_{q_{1,2}=0}$$

$$= -\frac{1}{6}\left(\frac{1}{\hat{\epsilon}} + \ln\left\{\frac{\mu^2}{m_f^2}\right\} + \frac{7}{6}\right)S_{6,\mu\nu}\ , \quad (86)$$

where we have implicitly included the $(\mu \longleftrightarrow \nu)$ permutation.

### 5.7. Mixing of the $Q_{8,\mu\nu}^0$ Operator

Finally, it is worth mentioning that operator $Q_{8,\mu\nu}^0$ is already $O(e^0 g_S^2)$, and therefore, its mixing coefficients are not relevant for the Wilson coefficients at the computed order. Note that the order $O(e^0 g_S^2)$ of the operator $Q_{8,\mu\nu}^0$ is not arbitrary, but rather, it is due to the fact that we need to introduce two interaction vertices from the Dyson series to complete the two cut quark lines (see Figure 9).

Up to this point, we have followed [60] to present the computation of the HLbL tensor $\Pi^{\mu_1\mu_2\mu_3\mu_4}$ in the high-energy regime via an OPE in the presence of an electromagnetic background field. We have generalized such an approach to include gluon and quark

background fields as well. In the OPE, there is a separation of perturbative contributions (which are bigger) and non-perturbative ones coming from matrix elements of strongly interacting operators. All contributions are computed at leading order up to dimension six operators. Infrared contributions to the Wilson coefficients of the OPE represented both conceptual and computational problems, but they were dealt with by performing renormalization of the composite operators of the OPE. The need for a renormalization program in the background field method context, when composite operators are represented by products of background classical quark, gluon, and photon fields, is not evident and it must be justified. We presented the rationale behind the renormalization scheme and performed all necessary computations within the background field method framework.

## 6. Computation of the Quark Loop by the Method of Bijnens

The main contribution to $\Pi^{\mu_1\mu_2\mu_3\mu_4}$ (and, thus, to $a_\mu$) comes from the Wilson coefficient of the electromagnetic field strength tensor $F_{\mu\nu}$, which is the quark loop (see Figure 6) and its expression is given in (60). After renormalization there are contributions from non-perturbative operators. However, such mixing contributions only affect the mass corrections of the quark loop. Consequently, in the remaining sections, we focus on the computation of the quark loop contribution to $a_\mu$ from the high-energy integration regions of the master formula (43). From (28), we know exactly how $\partial_{\nu_4}\Pi^{\mu_1\mu_2\mu_3\mu_4}_{HLbL}$, and thus $\Pi^{\mu_1\mu_2\mu_3\mu_4\nu_4}_F$, contributes to $a_\mu$ without recurring to a specific tensor basis. However, it is convenient to express the result in the tensor basis used for the master formula in order to benefit from the Gegenbauer polynomial framework that allowed us to simplify a full two-loop integral, containing eight integrals, into a threefold one.

In [60], the computation was performed by applying projectors which extract the relevant contributions to the $\overline{\Pi}_i$ of the master formula (43) out of the amplitude

$$\hat{\Pi}_i = P_i^{\mu_1'\mu_2'\mu_3'\mu_4'\nu_4'}\Pi_{F\,\mu_1'\mu_2'\mu_3'\mu_4'\nu_4'}\,. \tag{87}$$

Some denominator cancellations can be performed on the resulting scalar loop integrals such that they are written in terms of scalar tadpole, self-energy, and triangle integrals. Note that no scalar box integrals arise due to the $q_4 \to 0$ limit, which guarantees that, after applying the soft derivative, only three different propagators appear in the quark loop. These three scalar master integrals are then expanded as a function of the square of the infinitesimal (in the considered regime) quark mass $m_f^2$. Finally, the infrared divergences that appear as $\ln(Q_3^2/m_f^2)$ in the mass-suppressed corrections are cancelled via mixing with $S_{2,\mu\nu}$, as discussed in previous sections. The final result can be written as:

$$\hat{\Pi}_m^{\overline{MS}} = \hat{\Pi}_m^0 + m_f^2\hat{\Pi}_{\overline{MS},m}^{m_f^2} + O(m_f^4)\,, \tag{88}$$

$$\hat{\Pi}_m^0 = \frac{N_c e_q^4}{\pi^2}\sum_{i,j,k,n}\left[c_{ijk}^{(m,n)} + f_{ijk}^{(m,n)}F + g_{ijk}^{(m,n)}\ln\left(\frac{Q_2^2}{Q_3^2}\right) + h_{ijk}^{(m,n)}\ln\left(\frac{Q_1^2}{Q_2^2}\right)\right]\lambda^{-n}Q_1^{2i}Q_2^{2j}Q_3^{2k}\,,$$

$$\hat{\Pi}_{\overline{MS},m}^{m_f^2} = \frac{N_c e_q^4}{\pi^2}\sum_{i,j,k,n}\lambda^{-n}Q_1^{2i}Q_2^{2j}Q_3^{2k} \tag{89}$$

$$\times\left[d_{ijk}^{(m,n)} + p_{ijk}^{(m,n)}F + q_{ijk}^{(m,n)}\ln\left(\frac{Q_2^2}{Q_3^2}\right) + r_{ijk}^{(m,n)}\ln\left(\frac{Q_1^2}{Q_2^2}\right) + s_{ijk}^{(m,n)}\ln\left(\frac{Q_3^2}{\mu^2}\right)\right]\,,$$

where $c_{ijk}^{(m,n)}$, $f_{ijk}^{(m,n)}$, $g_{ijk}^{(m,n)}$, $h_{ijk}^{(m,n)}$, $d_{ijk}^{(m,n)}$, $p_{ijk}^{(m,n)}$, $q_{ijk}^{(m,n)}$, $r_{ijk}^{(m,n)}$, and $s_{ijk}^{(m,n)}$ are constant coefficients and their values are given in appendix C.1 of [60]. $\lambda$ is the Källen function of the three virtual photon momenta:

$$\lambda(q_1^2,q_2^2,q_3^2) \equiv q_1^4 + q_2^4 + q_3^4 - 2q_1^2q_2^2 - 2q_1^2q_3^2 - 2q_2^2q_3^2\,, \tag{90}$$

where we have used the standard notation $q^{2n} \equiv (q^2)^n$. In addition, $\mu$ represents the subtraction point of the $\overline{MS}$ renormalization scheme, which we introduced in the previous section. Finally, $F = F(Q_1^2, Q_2^2, Q_3^2)$ is the massless triangle integral:

$$F(Q_1^2, Q_2^2, Q_3^2) \equiv (4\pi)^2 i \int \frac{d^4 p}{(2\pi)^4} \frac{1}{p^2} \frac{1}{(p - q_1)^2} \frac{1}{(p - q_1 - q_2)^2} \; . \tag{91}$$

Note that the expressions of the form factors $\hat{\Pi}_i$ have several terms with negative powers of $\lambda$, which constitute spurious kinematic singularities in the $\lambda \to 0$ limit. These were introduced by the projectors that were used to extract the form factors from the quark loop amplitude, but they are explicitly cancelled in contributions from all other Wilson coefficients. In the case of the quark loop, however, there is implicit dependence on $\lambda$ coming from the massless triangle integral $F(Q_1^2, Q_2^2, Q_3^2)$, which thus obscures the cancellation of these singularities. When $F$ is Taylor-expanded around $\lambda = 0$, it is possible to see that all negative powers of $\lambda$ cancel explicitly. Such expansion is necessary in the integration regions of the master formula in which two virtual photon momenta have a similar size and are much bigger than the third one, namely $Q_1 \sim Q_2 \gg Q_3 \gg \Lambda_{QCD}$ and crossed versions. This regime is not quite the same as the $Q_1 \sim Q_2 \sim Q_3 \gg \Lambda_{QCD}$ symmetric regime, but the OPE remains valid anyway as long as we remain in the perturbative QCD domain.

## 7. Computation of the Quark Loop Amplitude in Our Work

We follow an alternative approach with respect to [60]. Instead of projecting the quark loop amplitude onto the form factors of the master formula as a first step, we compute the amplitude in its tensor form. At intermediate stages of the computation, we have to deal with tensor loop integrals, which we are able to write in terms of scalar ones by means of a kinematic singularity-free tensor decomposition method first presented in [107]. Once the tensor decomposition is performed, we finally project out the $\hat{\Pi}_i$ form factors of the master formula. In this way, we are able to verify that there are no quark loop contributions neglected by the projection procedure, which is an implicit check of the generality of the tensor structures of the HLbL tensor found in [34,35] that we discussed in Section 2.3.3. Finally, we compute the scalar integrals found in the tensor decomposition by means of their Mellin–Barnes representation [108]. The series representation of Mellin–Barnes integrals provides a full systematic expansion of the chiral corrections to the massless part of the quark loop. Finally, we perform a numeric evaluation of the master Formula (43) considering the quark loop contribution to the form factors $\hat{\Pi}_i$ and we discuss the results.

Our whole computation of the quark loop amplitude was carried out using version 12.3 of the software *Mathematica* and we also made extensive use of version 9.3.1 of *FeynCalc* package [62–64] to compute Dirac traces and for intermediate steps involving tensors.

### 7.1. First Stages of the Quark Loop Computation

The first step in our computation was to perform the differentiation and take the limit with respect to $q_4^{\nu_4}$, whose effect is to duplicate the propagator that they act upon:

$$\lim_{q_4 \to 0} \frac{\partial}{\partial q_{\nu_4}} S(p + q_4) = i \lim_{q_4 \to 0} S(p + q_4) \gamma^{\nu_4} S(p + q_4) = i S(p) \gamma^{\nu_4} S(p) \; . \tag{92}$$

This formula is different to the one cited in [60] due to different quark propagator conventions. It is convenient to perform this differentiation and limit before computing the trace and the loop integral, because by doing so, we reduce the number of different propagators and external momenta from four to three.

After the Dirac trace was computed, several denominator simplifications were performed to reduce the complexity of the structure of the remaining tensor loop integrals. This led to the appearance of integrals with only two different types of propagators in addition to the obvious ones with three. From these, the one with the most complex tensor

structure was a fifth-rank tensor with five propagators (but only three of them different from each other).

### 7.1.1. Tensor Loop Integral Decomposition

In general, the computation of tensor loop integrals involves decomposing them into a linear combination of their external momenta and the metric tensor in which coefficients are given in terms of scalar loop integrals. A standard procedure to achieve this is the Passarino–Veltman decomposition [109,110]. Scalar coefficients of this decomposition are obtained by contracting the tensor integral with each element of the tensor basis in which it is being decomposed. This yields a system of equations involving scalar integrals and form factor scalar products of the external momenta of the integral. One downside is that the form factors of the Passarino–Veltman decomposition always contain negative powers of the determinant of the Gram matrix of tensors used as a basis. These spurious kinematic singularities may be difficult to handle when the integrals of the (43) master formula are performed. Moreover, there are already unavoidable $\lambda^{-n}$ factors from the BTT projectors, so it is very inconvenient to introduce more singularities.

Since the Passarino–Veltman decomposition is technically inconvenient for our computation, we preferred to use an approach proposed by Davydychev in [107] for tensor decomposition into scalar integrals which does not introduce kinematic singularities in the coefficients, at the cost of shifting the (space-time) dimension of the scalar integrals. Let us describe this decomposition procedure before continuing with the discussion of the quark loop computation. First, we need to introduce suitable notation. Tensor and scalar loop integrals are represented as:

$$I^{(N)}_{\mu_1...\mu_M}(d;\nu_1,...,\nu_N) \equiv \int \frac{d^d p}{(2\pi)^d} \frac{p_{\mu_1}...p_{\mu_M}}{D_1^{\nu_1}...D_N^{\nu_N}} , \qquad I^{(N)}(d;\nu_1,...,\nu_N) \equiv \int \frac{d^d p}{(2\pi)^d} \frac{1}{D_1^{\nu_1}...D_N^{\nu_N}} . \tag{93}$$

where $D_i = (q_i + p) - m_i^2 + i\epsilon$ represents the usual scalar (possibly massive) propagator, $\nu_i$ is the power of propagator $D_i$ in the integral, $q_i$ is an arbitrary external momentum, and the Feynman prescription is implemented by $\epsilon \to 0^+$.

With this convention, the decomposition formula of tensor loop integrals in terms of scalar ones with shifted dimensions can be written as [107]:

$$\begin{aligned}
I^{(N)}_{\mu_1...\mu_M}(d;\nu_1,...,\nu_N) = \sum_{\substack{\lambda,\kappa_1,...,\kappa_N \\ 2\lambda+\sum_i \kappa_i = M}} & \left(-\frac{1}{2}\right)^\lambda \{[g]^\lambda [q_1]^{\kappa_1}...[q_N]^{\kappa_N}\}_{\mu_1...\mu_M} \\
& \times (\nu_1)_{\kappa_1}...(\nu_N)_{\kappa_N} (4\pi)^{M-\lambda} I^{(N)}(d+2(M-\lambda);\nu_1+\kappa_1,...,\nu_N+\kappa_N) ,
\end{aligned} \tag{94}$$

where $(\nu)_\kappa \equiv \Gamma(\nu+\kappa)/\Gamma(\nu)$ is the Pochhammer symbol. Note that there is a difference in the equation we cite here and the one written in [107] with respect to the factor of $4\pi$ due to the difference in the normalization convention for loop integrals. The structure between brackets represents the symmetrized tensor structure in which $g^{\mu_1\mu_2}$ appears $\lambda$ times, and each $q_i^{\mu_j}$ appears $\kappa_i$ times. Consequently, the restriction $2\lambda + \sum_i \kappa_i = M$ ensures that the tensor rank of the integral is conserved. The sum extends to all non-negative values of $\lambda, \kappa_1, ..., \kappa_N$. The proof of this formula rests mainly on the Schwinger representation of scalar loop integrals and recurrence formulas obtained by differentiation of such integrals with respect to each external momentum $q_i$. Finally, the result is generalized by induction. The proof of (94) is described in great detail in [107] and we will not repeat it here. Nevertheless, there are some features of the formula which are worth motivating. First, note that the number of times that a tensor element $q_i^{\mu_j}$ appears in the decomposition is related to the power with which its associated denominator $D_i$ appears. This is in fact reminiscent of the external momentum derivatives which where used to obtain the formula. For example, the starting point of the proof of the formula for the vector integral $I^{(N)}_\mu(d;\nu_1,...,\nu_N)$ is the following differential identity:

$$\frac{1}{2\nu_1}\frac{\partial}{\partial q_1^{\mu}}I^{(N)}(d;\nu_1,...,\nu_N) = -I_{\mu}^{(N)}(d;\nu_1+1,...,\nu_N) - q_{1\mu}I^{(N)}(d;\nu_1+1,...,\nu_N).\tag{95}$$

The difference in the powers of the $\nu_1$ in the derivative term and the two terms to the right is solved by using the Schwinger representation for scalar integrals, namely:

$$I^{(N)}(d;\nu_1,...,\nu_N) = \pi^{d/2}\,i^{1-d}\,\Gamma\left(\sum_i\nu_i - \frac{d}{2}\right)\left[\prod_i\Gamma(\nu_i)\right]^{-1}\tag{96}$$

$$\times\int_0^1\cdots\int_0^1\prod\beta_i^{\nu_i-1}d\beta_i\,\delta\left(\sum_i\beta_i - 1\right)\left(\sum_{j<l}\sum\beta_j\beta_l(p_j-p_l)^2 - \sum_i\beta_im_i^2\right)^{d/2-\sum_i\nu_i},$$

which is valid for $Re\{\nu_i\} > 0$. There, one can see how a shift in the sum of powers of denominators $\sum\nu_i$ may be offset by a twofold shift in the scalar integral's dimension. In the case of the metric tensor, its appearance is related to a reduction in the shift in the dimension of the scalar integral. This is due to the fact that metric tensors enter this decomposition from terms in which an external momentum derivative acts on its corresponding momentum, not on the scalar integral that is multiplying it; therefore, it requires no additional offset and its dimensional shift is not increased. An explicit example of this situation can bee seen when taking a second derivative of the vector integral $I_{\mu}^{(N)}$ in (95).

We applied (94) to the tensor integrals appearing in our computation of the quark loop amplitude; thus, its tensor structure was explicitly written in terms of the external momenta and the metric. As such, it was then possible to compare this structure to the $\partial_{q_4}^{\nu_4}\hat{T}_i^{\mu_1\mu_2\mu_3\mu_4}$ tensor basis that is used for the (43) master formula. To achieve this, we extracted the quark loop contributions to the form factors $\hat{\Pi}_i$ with the help of the projectors of [60]. We found that all form factors received non-zero contributions from the quark loop. Furthermore, when we subtracted such contributions from the amplitude itself, the result was equal to zero, which means that the quark loop amplitude contains no spurious parts that do not contribute to $a_{\mu}$. This implies that the first-principles arguments presented in Section 2.3.3 to justify the decomposition of the HLbL tensor completely characterize the tensor structure of the quark amplitude, at least with respect to the soft derivative part of the decomposition.

It is worth noting that the tensor basis used in [60] is the one proposed in [35], not the one from [34]. The choice of any of these two sets is of course irrelevant for $a_{\mu}$, and in this work, we used the latter, because we were interested in using the projectors of [60].

### 7.1.2. Computation of Scalar Integrals with Shifted Dimensions

After tensor-decomposing loop integrals and applying projectors on the quark loop amplitude, the form factors $\hat{\Pi}_i$ are given in terms of scalar integrals with shifted dimensions coming from (94). It is necessary to compute them in order to perform the $|Q_1|$, $|Q_2|$ and $\tau$ integrals of the master formula. These scalar loops appear with two and three different propagators in the quark loop amplitude (see Figure 11) and we compute them using a general Mellin–Barnes representation for $N$-point scalar integrals with equal masses, which was first published in [108]. The self-energy and triangle expressions are, respectively:

$$I^{(2)}(d;\nu_1,\nu_2) = \frac{i^{1-d}}{(4\pi)^{d/2}}\frac{(-m^2)^{\frac{d}{2}-\nu_1-\nu_2}}{\Gamma(\nu_1)\Gamma(\nu_2)}\int_{s_1}\frac{\Gamma(s_1+\nu_1)\Gamma(s_1+\nu_2)}{\Gamma(2s_1+\nu_1+\nu_2)}\left(-\frac{q_1^2}{m^2}\right)^{s_1}$$

$$\times\Gamma(-s_1)\Gamma\left(s_1+\nu_1+\nu_2-\frac{d}{2}\right),\tag{97}$$

and

$$
\begin{aligned}
I^{(3)}(d; \nu_1, \nu_2, \nu_3) = {} & \frac{i^{1-d}}{(4\pi)^{d/2}} \frac{(-m^2)^{\frac{d}{2} - \sum_i \nu_i}}{\Gamma(\nu_1)\Gamma(\nu_2)\Gamma(\nu_3)} \int_{\substack{s_{12} \\ s_{13} \\ s_{23}}} \left( -\frac{q_{12}^2}{m^2} \right)^{s_{12}} \left( -\frac{q_{13}^2}{m^2} \right)^{s_{13}} \left( -\frac{q_{23}^2}{m^2} \right)^{s_{23}} \\
& \times \Gamma(-s_{12})\Gamma(-s_{13})\Gamma(-s_{23}) \\
& \times \Gamma\left( \nu_1 + s_{12} + s_{13} \right)\Gamma\left( \nu_2 + s_{12} + s_{23} \right)\Gamma\left( \nu_3 + s_{13} + s_{23} \right) \\
& \times \Gamma\left( \sum_i \nu_i - \frac{d}{2} + s_{12} + s_{13} + s_{23} \right)\left[ \Gamma\left( \sum_i \nu_i + 2s_{12} + 2s_{13} + 2s_{23} \right) \right]^{-1}.
\end{aligned}
\tag{98}
$$

.

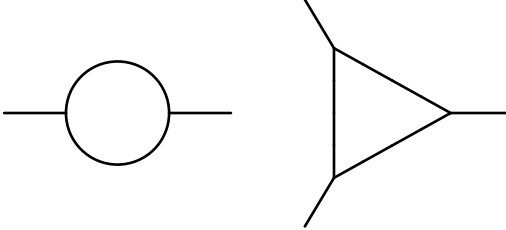

**Figure 11.** Self-energy and triangle topologies appearing in the quark loop computation.

The computation of these integrals is a complex task even for the self-energy case for both practical and conceptual reasons. From a practical perspective, the appearance of multiple Gamma functions, each with its own set of poles and zeros, renders the pole structure of the integrand unusually complex. Furthermore, the triangle integral has a triple-nested integral and the poles of the Gamma functions are intertwined, which introduces a conceptual difficulty; the standard complex variable residue framework that is enough for the self-energy case cannot be naively expanded in general by iteration to consider multiple complex variable integrals. The mathematical tools necessary to face this issue can be found in Appendix A.

### 7.2. Final Stages of the Quark Loop Computation and Analysis

At this point, we have all the necessary tools to compute self-energy and tadpole integrals ((97) and (98)). Nevertheless, in the quark loop expression, there appear more than one hundred different scalar integrals of these two types; hence, automation is required. For this, we have used a *Mathematica* package called *MBConicHulls* (This package requires *Mathematica* 12 or a more recent version) [65], which calls upon functions of another package called *MultivariateResidues* [111] that has to be installed as a dependency. In [65,112], the authors describe how the computation of Mellin–Barnes integrals with multivariate residues, reviewed in Appendix A, can be organized in a very compact algorithm that uses very intuitive geometric concepts and allows for understanding the practical implications of the rather abstract results of multivariate complex calculus.

A typical Mellin–Barnes integral representing a scalar triangle loop has sixteen different series representations, and each of them contains up to six different subseries. Consequently, the assessment of the convergence regions of the series representations found by the *MBConicHulls* package requires automation as well. We have developed a program that evaluates the asymptotic behavior of a given triple series and finds its region of convergence by comparing it with the behavior of other series whose convergence conditions are already known. The concept of the program is based on Horn's theorem for the convergence of hypergeometric series of up to three variables [113], which is a rather natural extension of D'Alembert's ratio test to the multivariate case. Let us consider the triple series

$$
\sum_{n_1, n_2, n_3}^{\infty} C(n_1, n_2, n_3) x^{n_1} y^{n_2} z^{n_3}.
\tag{99}
$$

It is considered hypergeometric as long as the coefficients

$$f(n_1, n_2, n_3) = \frac{C(n_1 + 1, n_2, n_3)}{C(n_1, n_2, n_3)}, \quad g(n_1, n_2, n_3) = \frac{C(n_1, n_2 + 1, n_3)}{C(n_1, n_2, n_3)},$$

$$h(n_1, n_2, n_3) = \frac{C(n_1, n_2, n_3 + 1)}{C(n_1, n_2, n_3)}$$

(100)

are rational functions of $n_1$, $n_2$, and $n_3$. If so, then the convergence region of the integral is given by the intersection of the following five sets:

$$E = \left\{ (|x|, |y|, |z|) \;\middle|\; \forall (n_1, n_2, n_3) \in \mathbb{R}_+^3 \;:\; |x| < \rho(n_1, n_2, n_3) \vee |y| < \sigma(n_1, n_2, n_3) \vee |z| < \tau(n_1, n_2, n_3) \right\}$$

$$X = \left\{ (|x|, |y|, |z|) \;\middle|\; \forall (n_2, n_3) \in \mathbb{R}_+^2 \;:\; |x| < \rho(0, n_2, n_3) \vee |y| < \sigma(0, n_2, n_3) \vee |z| < \tau(0, n_2, n_3) \right\}$$

$$Y = \left\{ (|x|, |y|, |z|) \;\middle|\; \forall (n_1, n_3) \in \mathbb{R}_+^2 \;:\; |x| < \rho(n_1, 0, n_3) \vee |y| < \sigma(n_1, 0, n_3) \vee |z| < \tau(n_1, 0, n_3) \right\}$$

$$Z = \left\{ (|x|, |y|, |z|) \;\middle|\; \forall (n_1, n_2) \in \mathbb{R}_+^2 \;:\; |x| < \rho(n_1, n_2, 0) \vee |y| < \sigma(n_1, n_2, 0) \vee |z| < \tau(n_1, n_2, 0) \right\}$$

$$C = \left\{ (|x|, |y|, |z|) \;\middle|\; |x| < \rho(1, 0, 0) \wedge |y| < \sigma(1, 0, 0) \wedge |z| < \tau(1, 0, 0) \right\},$$

(101)

where $\mathbb{R}_+$ represents the set of positive reals and $\rho$, $\sigma$, and $\tau$ capture the asymptotic behavior of $f$, $g$, and $h$:

$$\rho(n_1, n_2, n_3) = \left| \lim_{u \to \infty} f(un_1, un_2, un_3) \right|^{-1}, \quad \sigma(n_1, n_2, n_3) = \left| \lim_{u \to \infty} g(un_1, un_2, un_3) \right|^{-1},$$

$$\tau(n_1, n_2, n_3) = \left| \lim_{u \to \infty} h(un_1, un_2, un_3) \right|^{-1}.$$

(102)

The program that we have developed computes $\rho$, $\sigma$, and $\tau$ for each subseries that form a series representation of a Mellin–Barnes integral and identifies its region of convergence by comparing them to the $\rho$, $\sigma$, and $\tau$ of series whose convergence conditions are known. Care had to be taken for the program not be misled by redefinitions of the arguments of the series or the presence of logarithms. We have also taken into account the result found in [112] which extends the use of Horn's theorem to series that are not hypergeometric by the definition given previously, because they include polygamma functions. Finally, the program chooses the appropriate series representation according to the kinematic regime indicated beforehand.

The convergence region of some triple series representations of triangle loops in shifted dimensions could not be found in the mathematical literature due to them being quite nonstandard. In such cases, the approach presented in [114], alternate to [108], was followed. That paper refers to scalar triangle loop integrals in arbitrary space-time dimension with three different masses, but unit propagator powers:

$$J_3^{(d)} = \int \frac{d^d p}{(2\pi)^d} \frac{1}{(p + p_1 + p_2)^2 - m_3^2} \frac{1}{(p + p_1)^2 - m_2^2} \frac{1}{p^2 - m_1^2} .$$

(103)

The first step of the computation is to use Feynman parameters in the standard way, as we described for the self-energy loop when introducing formula (97). An appropriate change of variables renders one of the two Feynman parameter integrals straightforward to perform. After using a Mellin–Barnes representation of the integrand, the remaining Feynman parameter integral has the one-variable Gaussian hypergeometric function $_2F_1$ as its solution. Finally, the Mellin–Barnes integral of $_2F_1$ yields the double Appell hypergeometric function $F_1$. The key point of this result is that $F_1$ belongs to the well-known family of Gaussian hypergeometric functions, and as such, its convergence and analytical continuation properties are well-known [115,116]. We quote here the result for arbitrary space-time dimension $d$ valid in the high-energy regime:

$$J_3^{(d)} = \frac{i\Gamma\left(\frac{4-d}{2}\right)}{(4\pi)^{\frac{d}{2}}\lambda^{1/2}(p_1^2, p_2^2, p_3^2)} \left\{ J_{123}^{(d)} - (M_3 - i\epsilon)^{\frac{d-4}{2}} J_{123}^{(d=4)} + (1,2,3) \leftrightarrow (2,3,1) \right.$$

$$\left. + (1,2,3) \leftrightarrow (3,1,2) \right\}, \tag{104}$$

where $p_3 = -p_1 - p_2$ and:

$$J_{ijk}^{(d)} = \frac{x_{ij}}{(x_k - x_{ij})}(M_{ij} - i\epsilon)^{\frac{d-4}{2}} F_1\left(\frac{1}{2}; 1, \frac{4-d}{2}; \frac{3}{2}; \frac{x_{ij}^2}{(x_k - x_{ij})^2}, -\frac{p_i^2 x_{ij}^2}{M_{ij} - i\epsilon}\right)$$

$$- \frac{x_{ij}^2}{2(x_k - x_{ij})^2}(M_{ij} - i\epsilon)^{\frac{d-4}{2}} F_1\left(1; 1, \frac{4-d}{2}; 2; \frac{x_{ij}^2}{(x_k - x_{ij})^2}, -\frac{p_i^2 x_{ij}^2}{M_{ij} - i\epsilon}\right) \tag{105}$$

$$- \left\{ x_{ij} \to 1 - x_{ij} \; ; \; x_k \to 1 - x_k \right\}$$

$$x_{ij} = \frac{p_i^2 + m_i^2 - m_j^2}{2p_i^2} . \tag{106}$$

$M_3$ and $M_{ij}$ for $i, j = 1, 2, 3$ are defined in terms of Cayley and Gramm determinants for the triangle loop. Their definition and properties are given in Appendix B. The definition of $x_k$ is rather lengthy and not very relevant, so it is written in the appendix as well. The Appell function $F_1$ has the convergent series representation:

$$F_1(a; b, b'; c; x, y) = \sum_{n_1, n_2 = 0} \frac{(a)_{n_1 + n_2}(b)_{n_1}(b')_{n_2}}{(c)_{n_1 + n_2}} \frac{x^{n_1}}{n_1!} \frac{y^{n_2}}{n_2!} \tag{107}$$

for $|x| < 1$ and $|y| < 1$. In the high-energy regime, we have $\left|\frac{x_{ij}^2}{(x_k - x_{ij})^2}\right| = \left|\frac{m_i^2 - M_{ij}}{M_3 - M_{ij}}\right| < 1$ and $\left|\frac{p_i^2 x_{ij}^2}{M_{ij}}\right| = \left|1 - \frac{m_i^2}{M_{ij}}\right| < 1$; therefore, this representation is valid for our quark loop computation. From this result, triangle loops with arbitrary propagator powers can be computed from $J_3^{(d)}$ via derivatives with respect to the masses:

$$I^{(N)}(d; \nu_1, ..., \nu_N) = \prod_i \left(\frac{1}{\Gamma(\nu_i)}\left(\frac{\partial}{\partial m_i^2}\right)^{\nu_i - 1}\right) J_3^{(d)}\bigg|_{m_i = m} . \tag{108}$$

We are interested in integrals with $d \in \{4, 6, 8, 10, 12\}$. It is not difficult to note that the gamma function pole at $d = 4$ in $J_3^{(d)}$ is of course a spurious singularity. On the other hand, for $d \geq 6$, there are actual ultraviolet singularities, but it is possible to check in a lengthy but straightforward way that singular terms vanish and dependence on the renormalization scale disappears when propagator powers grow high enough in $I^{(N)}$. This is relevant for us, because the loop integrals we find are ultraviolet finite.

The script that performs the steps that we have described throughout this section can be found in this repository: https://github.com/DanielMelo2000/QuarkLoopCode (accessed on 1 April 2024). After the Mellin–Barnes representations of the scalar integrals with shifted dimensions are computed by the methods described previously, the last step is the computation of the integrals over $|Q_1|$, $|Q_2|$, and $\tau$ that remain in the master formula.

First, let us remember that even though we followed a kinematic singularity-free tensor loop decomposition, there are spurious kinematic singularities which have been introduced by the negative powers of $\lambda$ that are present in the projectors with which one extracts the HLbL form factors from the quark loop. As discussed at the beginning of this section, these singularities cancel explicitly for contributions of all Wilson coefficients, except the quark loop. This is expected, since the spurious nature of the singularities implies that they must

disappear in tree-level contributions. For self-energy and triangle loop integrals in shifted dimensions, we do not arrive in general at a closed analytical expression, but rather a series representation. Therefore, spurious kinematic zeros inside these terms do not necessarily show explicitly to cancel singularities. This introduces numerical instability in the region of the master formula's angular integral when $\tau \equiv \hat{Q}_1 \cdot \hat{Q}_2 \to \pm 1$, because that is when $\lambda$ is equal to or approaches zero:

$$\lambda(q_1^2, q_2^2, q_3^2) = (q_3^2 - q_1^2 - q_2^2)^2 - 4q_1^2 q_2^2 = 4Q_1^2 Q_2^2 \left( \tau^2 - 1 \right), \tag{109}$$

where we have switched back to the Euclidean versions of the virtual photon momenta $q_i \cdot q_j \to -Q_i \cdot Q_j$. When computing the contribution to the master formula's integral from these regions, it is convenient to expand the integral's series representations around $\tau = \pm 1$ to avoid numerical instability. The fact that we traced the $\lambda = 0$ singularity to a value in $\tau$ has useful practical implications. Indeed, self-energy and triangle scalar integrals are computed in a single and triple series representation, respectively, where the expansion variables are the differences between external momenta. The external momenta that can appear in quark loop scalar integrals are $Q_1$ and $Q_3$ or $Q_2$ and $Q_3$, depending on the permutation one is considering. However, only $Q_3$ depends on $\tau$. Therefore any integral that does not depend on $Q_3$ does not require special treatments.

To perform the integrals on the Euclidean norm of the virtual photons' momenta, it is important to keep in mind that the quark loop was obtained from an OPE in perturbative QCD. Therefore, its range of validity starts above $\Lambda_{\text{QCD}}$, the perturbative threshold. $\Lambda_{\text{QCD}}$ is usually taken to be close to the proton's mass, which is about $\sim 940$ MeV. In principle, this means that one can compute the quark loop contribution to $a_\mu$ starting from $|Q_1| = |Q_2| = 1$ GeV $\equiv Q_{min}$; however, taking into account that the OPE framework discussed in previous sections introduces an implicit counting parameter $\Lambda_{\text{QCD}}/|Q|$, one would expect the error coming from neglected higher non-perturbative effects to be large right above $\Lambda_{\text{QCD}}$. The relation between the size of such an error and the values of $Q_{min}$ was studied in [60]. To that end, they computed the quark loop contribution as a function of $Q_{min}$ in the interval $[1 \text{ GeV}, 4 \text{ GeV}]$ and the contributions from the non-perturbative condensates of the previous sections were considered as well. Their results showed that massless quark loop contributions fall like $1/Q_{min}^2$ and, in general, contributions from elements of the OPE with dimension $d$ behave like $1/Q_{min}^d$. This is expected from the asymptotic behavior of the integral kernels $T_i$ of the master formula (43), which for $|Q_i| \to \infty$ behave like $m_\mu^2/Q_i^2$, except for $T_1$, which falls like $m_\mu^4/Q_i^4$. Since mass effects become small for large momenta $|Q_i|$ and the massless quark loop contribution does not introduce an energy scale, then it must fall like $1/Q_{min}$. Mass corrections to the quark loop are suppressed by $m_f^2/Q_i^2$ with respect to the massless part. In addition, contributions from other OPE elements $S_{i,\mu\nu}$ of dimension $d$ are comparatively suppressed as well by a factor $(\Lambda_{QCD}/|Q_i|)^{d-2}$; thus, the asymptotic behavior of their contributions is explained. Asymptotic freedom also plays a role in this result. As we mentioned in the OPE sections, the correction to the naive dimensional counting of the OPE is given by the anomalous dimension of each OPE element, but QCD's asymptotic freedom ensures that, at high enough energy, corrections are small.

For the value of the quarks' masses $m_f$ and the renormalization scale $\mu$, we followed the simplified choice of [60], which was:

$$m_u = m_d = 5 \text{ MeV} \qquad m_s = 100 \text{ MeV} \qquad \mu = Q_{min}. \tag{110}$$

In [10], constituent masses are used, because they are more appropriate when comparing with low-energy results. Note that no running of the masses is performed. This is justified by the very small size of mass corrections to the quark loop. With these values, we computed the quark loop contribution for $Q_{min} = 1$ or 2 GeV. As discussed previously in this section, we obtained a systematic expansion of the quark loop in terms of the quark masses. This

allowed us to study the mass corrections to the massless quark loop contribution, and qw found them to be very small, even at the $m_f^2$ order. Furthermore, we found the result for $Q_{min} = 1$ GeV to be about four times bigger than the $Q_{min} = 2$ GeV case.

In [60], the massless quark loop was found to be the largest contribution to $a_\mu$ by two orders of magnitude and the leading mass corrections were even smaller than the non-perturbative di-quark magnetic susceptibility ($S_{2,\mu\nu}$ in the OPE) by two further orders of magnitude. The complete results of [60] are summarized in Table 2, where one can see the dominance of quark loop contributions with respect to the other. Note, however, that those results do not show the complete picture, because the quark loop contribution, which is the leading perturbative contribution to $a_\mu^{\mathrm{HLbL}}$, does not really involve strong interactions, and hence, it does not depend on $\alpha_s$. Nevertheless, in [61] the next-to-leading-order (NLO) gluonic correction to the massless part of the quark loop was computed (see Figure 12), taking into account the running of $\alpha_s(\mu)$, and its contribution to $a_\mu^{\mathrm{HLbL}}$ was found to be about 10 % of the leading order and negative.

**Table 2.** Results published in [60] about the contribution of the quark loop and the rest of the OPE elements $S_{i,\mu\nu}$ to $a_\mu$ as a function of the cut-off $Q_{min}$ from which the master formula integral is performed.

| OPE Element | Magnetic Susceptibility | Mass Order | Contribution to $a_\mu$ from $Q_{min}$ 1 GeV | 2 GeV |
|---|---|---|---|---|
| $S_{1,\mu\nu}$ | 1 | $m^0$ | $1.73 \times 10^{-10}$ | $4.35 \times 10^{-11}$ |
| | | $m^2$ | $-5.7 \times 10^{-14}$ | $-3.6 \times 10^{-15}$ |
| $S_{2,\mu\nu}$ | $-4 \times 10^{-2}$ GeV | $m^1$ | $-1.2 \times 10^{-12}$ | $-7.3 \times 10^{-14}$ |
| | | $m^3$ | $6.4 \times 10^{-15}$ | $1.0 \times 10^{-16}$ |
| $S_{3,\mu\nu}$ | $3.5 \times 10^{-3}$ GeV$^3$ | | $-3.0 \times 10^{-14}$ | $-4.7 \times 10^{-16}$ |
| $S_{4,\mu\nu}$ | $3.5 \times 10^{-3}$ GeV$^3$ | | $3.3 \times 10^{-14}$ | $5.3 \times 10^{-16}$ |
| $S_{5,\mu\nu}$ | $-1.6 \times 10^{-2}$ GeV$^3$ | | $-1.8 \times 10^{-13}$ | $-2.8 \times 10^{-15}$ |
| $S_{6,\mu\nu}$ | $2 \times 10^{-2}$ GeV$^4$ | $m^0$ | $1.3 \times 10^{-13}$ | $2.0 \times 10^{-15}$ |
| $S_{7,\mu\nu}$ | $3.3 \times 10^{-3}$ GeV$^4$ | | $9.2 \times 10^{-13}$ | $1.5 \times 10^{-14}$ |
| $S_{8,1,\mu\nu}$ | $-1.4 \times 10^{-4}$ GeV$^4$ | | $3.0 \times 10^{-13}$ | $4.7 \times 10^{-15}$ |
| $S_{8,2,\mu\nu}$ | $-1.4 \times 10^{-4}$ GeV$^4$ | | $-1.3 \times 10^{-13}$ | $-2.0 \times 10^{-15}$ |

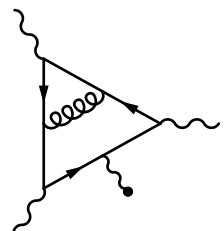

**Figure 12.** Representative diagram of the NLO contribution to the Wilson coefficient of $S_{1,\mu\nu}$ in the OPE of $\Pi^{\mu_1\mu_2\mu_3}$. The black dot represents creation/annihilation of a line by the background fields in the vacuum. This diagram represents the first QCD correction to the quark loop.

At the beginning of Section 3, we mentioned two different roles that SDCs play in the computation of $a_\mu$. First, one can obtain the high-energy asymptotic behavior of a Green function to learn the asymptotic behavior of an hadronic form factor, in order to fill the gap of missing or scarce experimental data in the high-energy parts of dispersive integrals or the master formula. Secondly, one can similarly use this approach directly to the HLbL tensor in the high-energy regions of the master integral, which is the purpose of the quark loop computation that we carried out. However, in the latter case, the interplay between low- and high-energy contributions is not clear-cut, because low-energy contributions are computed for the full $|Q_1|$ and $|Q_2|$ intervals of the master integral, not only up to 1 or 2 GeV. Thus, one sees that some overlapping of contributions is present and there is a risk of double counting. Therefore, for the high-energy contribution to be successfully accounted

for, it is necessary to understand how much of it has already been taken into account by low-energy computations.

One way to answer that question is to determine how much the low-energy contributions' asymptotic behavior resembles the results of the high-energy framework. It was argued in [54,55] that it is impossible to fulfill all QCD SDCs with a finite number of resonances. To obtain an estimate of the missing high-energy contributions caused by such a mismatch, one can use a top-down approach: to constrain hadronic contributions to fulfill SDC and study how much the result differs from when they are constrained by experimental data, that is, by their low-energy behavior. For example, the mixed-virtuality regime $Q_1 \sim Q_2 \equiv Q \gg Q_3 \gg \Lambda_{QCD}$ of the HLbL tensor, first studied in [117], imposes the following constraint:

$$\lim_{Q,Q_3 \to \infty} Q^2 Q_3^2 \overline{\Pi}_1 = -\frac{2}{3\pi^2} \tag{111}$$

and a similar one for crossed condition for $\overline{\Pi}_2$. In addition, as we already argued, the symmetric regime $Q_1 \sim Q_2 \sim Q_3 \equiv Q \gg \Lambda_{QCD}$, via the massless quark loop, imposes the following asymptotic behavior:

$$\lim_{Q \to \infty} Q^4 \overline{\Pi}_1 = -\frac{4}{9\pi^2} . \tag{112}$$

The proposals to ensure that the transition form factors match the mixed-virtuality behavior have ranged from ignoring their momentum dependence [117] to summing an infinite tower of axial and vector resonances in holographic QCD [40,41]. In [56,57], a hybrid approach is followed; pseudo-scalar pole contributions are computed in a large-$N_c$ Regge model such that they satisfy SDC, but those results are only used in the low-energy region of integration of the master formula. The integral over the remaining part is computed with the quark loop expression, taking advantage of the asymptotic behavior of the massless quark loop contribution to $\overline{\Pi}_1$, which fulfills the mixed-virtuality SDC as well. This reduces model dependence with respect to the first two approaches mentioned and allows for the clear separation of the effect of SDC on low- and high-energy contributions to lower double counting risks. Nevertheless, such risks still remain with respect to axial vector contributions, in a transition region between the perturbative and non-perturbative domains of QCD, and are still a significant source of uncertainty for $a_\mu^{\text{HLbL}}$. Compared to the data-driven computation, there is an increase in the contribution from pseudo-scalar poles:

$$\Delta a_\mu^{LSDC} = \left[ 8.7(5.5)_{\text{PS-poles}} + 4.6(9)_{\text{pQCD}} \right] \times 10^{-11} = 13(6) \times 10^{-11} , \tag{113}$$

where the superindex *LSDC* illustrates the fact that we are only considering the constraints regarding the asymptotic behavior of the "longitudinal" part of the HLbL tensor in the mixed-virtuality regime. The "transversal" form factors are $\overline{\Pi}_{3-12}$. They are related to the contribution from axial vectors and obey a different SDC in the mixed-virtuality regime. This result is in very good agreement with the holographic QCD one [40]. In contrast, it hints at an overestimation from the approach proposed in [117]. When $\Delta a_\mu^{LSDC}$ is computed fully with the large-$N_c$ Regge model, the result is very close to (113). In the end, the net increase in the HLbL contribution due to SDC is estimated to be $\Delta a_\mu^{SDC} = 15(10) \cdot 10^{-11}$ [10]. A part of the uncertainty of (113) is estimated by varying the matching scale between the Regge model and the quark loop, and is then added to each element's model or theoretical uncertainty. Therefore, higher-order corrections to the quark loop can decrease the uncertainty of $\Delta a_\mu^{LSDC}$. In [58], the SDC contribution was reassessed, taking into consideration the perturbative corrections to the quark loop, and the result was:

$$\Delta a_\mu^{LSDC} = \left[ 8.7(5.3)_{\text{PS-poles}} + 4.2(1)_{\text{pQCD}} \right] \times 10^{-11} = 13(5) \times 10^{-11} , \tag{114}$$

which reduces the uncertainty of the previous result. It is worth mentioning that the negative $O(\alpha_s)$ correction to the massless quark loop improves the agreement between the Regge sum of pseudoscalars and the perturbative result. However, further study regarding the matching procedure is still needed.

Recently, two works [118,119] were published regarding the extension of the background OPE framework to the mixed-virtuality regimes of the HLbL tensor. To illustrate the broader range of use of the background field framework that we used in this work, we briefly review their results. The analogue of $\Pi^{\mu_1\mu_2\mu_3}$ is now:

$$
\begin{aligned}
\Pi^{\mu_1\mu_2} &= \frac{i}{e^2} \int \frac{d^4 q_4}{(2\pi)^4} \int d^4 x \int d^4 y \, e^{-i(q_1 x + q_2 y)} \langle 0 | T J^{\mu_1}(x) J^{\mu_2}(y) | \gamma^*(q_3) \gamma(q_4) \rangle \\
&= -\epsilon_{\mu_3}(q_3) \epsilon_{\mu_4}(q_4) \Pi^{\mu_1\mu_2\mu_3\mu_4}(q_1, q_2, q_3) \, .
\end{aligned}
\tag{115}
$$

When the OPE is performed up to operators with mass dimension $D = 4$, the quoted result is:

$$
\begin{aligned}
\Pi^{\mu_1\mu_2} = &-\frac{1}{4} \langle F_{\nu_3\mu_3} F_{\nu_4\mu_4} \rangle \frac{\partial}{\partial q_{3\nu_3}} \frac{\partial}{\partial q_{4\nu_4}} \Pi^{\mu_1\mu_2\mu_3\mu_4}_{\text{quark loop}} \Big|_{q_3 = q_4 = 0} \\
&- \frac{e_f^2}{e^2} \langle \overline{\psi}(0) \left( \gamma^{\mu_1} S^0(-\hat{q}) \gamma^{\mu_2} - \gamma^{\mu_2} S^0(-\hat{q}) \gamma^{\mu_1} \right) \psi(0) \rangle \\
&- \frac{i e_f^2}{e^2 \hat{q}^2} \left( g^{\mu_1 \delta} g^{\mu_2}_\beta + g^{\mu_2 \delta} g^{\mu_1}_\beta - g^{\mu_1\mu_2} g^\delta_\beta \right) \left( g_{\alpha\delta} - 2 \frac{\hat{q}_\alpha \hat{q}_\delta}{\hat{q}^2} \right) \langle \overline{\psi}(0) \left[ \overrightarrow{D}^\alpha - \overleftarrow{D}^\alpha \right] \gamma^\beta \psi(0) \rangle \, ,
\end{aligned}
\tag{116}
$$

where $\hat{q} \equiv (q_1 - q_2)/2$ and the matrix element $\langle ... \rangle$ now includes the virtual photon $\gamma^*(q_3)$ and the real soft one $\gamma(q_4)$. The term $\Pi_{\text{quark loop}}$ is proportional to the quark loop amplitude discussed in this work. The origin of the first term is quite clear; it comes from matrix elements with four contracted quark fluctuations in which the resulting two fermion propagators have a total of two soft photon insertions between the two (see Figure 13), hence the two derivatives and field strength tensors. The second and third terms come instead from terms with two background quark fields and two fluctuations (see Figure 14), where the background fields are Taylor-expanded as usual up to order $O(x_1 - x_2)$. The appearance of $\hat{q}$ comes from the fact that only $x_1 - x_2$ is close to zero in the mixed-virtuality regime, in contrast to the symmetric regime, where the three currents' coordinates are close. It is worth mentioning that in this case, quark operators start at dimension $D = 3$, and therefore, they are, in principle, the leading term of the OPE instead of the perturbative quark loop.

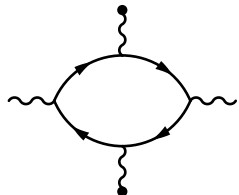

**Figure 13.** Representative diagram of the fully perturbative contribution to the OPE of $\Pi^{\mu_1\mu_2}$ in the mixed-virtuality regime. A black dot represents creation/annihilation of a line by the background fields in the vacuum. Depending on the value of $q_3$, one of these photons may interact perturbatively with the vacuum.

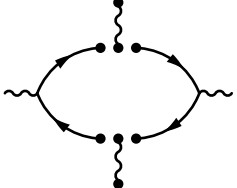

**Figure 14.** Representative diagram of the one-cut quark contributions to the OPE of $\Pi^{\mu_1\mu_2}$ in the mixed-virtuality regime. A black dot represents creation/annihilation of a line by the background fields in the vacuum. Depending on the value of $q_3$, one of these photons may interact perturbatively with the vacuum.

## 8. Conclusions

We have reviewed the basic framework for the dispersive computation of the HLbL contribution to the anomalous magnetic moment of the muon, $a_\mu^{\text{HLbL}}$, focusing on the kinematic singularity free tensor decomposition that allowed it. Unlike hadronic models used previously, dispersive estimates have allowed for the computation of unambiguous contributions, at least for pseudoscalar poles, which has improved the uncertainty estimation. We have also discussed the corresponding role of SDC as a means of uncertainty assessment and high-energy contribution computation.

The main focus of this work is on the HLbL scattering amplitude in the symmetric high-energy regime, in which it can be represented via an OPE in which the soft photon is regarded as a background field to avoid infrared-divergent Wilson coefficients. To present a thorough discussion, we introduced the background field method and stressed how renormalization and operator mixing are included in a very natural way within that framework. The same applies for the derivation of the Wilson coefficients, in which perturbative and non-perturbative contributions are systematically separated and do not require much decision making from the user. From the OPE of $\Pi^{\mu_1\mu_2\mu_3\mu_4}$, we found that the quark loop is the leading contribution after infrared-divergent logarithms have been subtracted by renormalization, in agreement with the literature.

Finally, we presented the computation of the quark loop with its full tensor structure, that is, without projecting the form factors of the HLbL tensor out of it, in contrast with previous computations. This allowed us to check the generality of the basis elements of the kinematic singularity-free tensor decomposition of the HLbL tensor. We concluded that these elements do span the tensor structures of the quark amplitude, thus obtaining an explicit check that we have not found in the literature. To compute the full quark loop amplitude, it was necessary to use a decomposition algorithm for tensor loop integrals and we used one that did not introduce further spurious kinematic singularities, at the cost of shifting the dimensions of the integrals. Using a Mellin–Barnes representation for the resulting scalar integrals allowed us to keep full mass dependence and obtain a complete series representation of the required integrals that contains all quark mass effects at any order in the high-energy regime. We also presented the fundamentals of single and multiple complex variable residue computation necessary to provide a reasonably thorough mathematical foundation for the procedure. The aforementioned computations were implemented by a *Mathematica* script that used *FeynCalc*. To highlight the importance of the quark loop computation result, we described the use of the quark loop computation as an SDC to the low-energy contributions to $a_\mu^{\text{HLbL}}$ and its effects in the critical task of lowering the uncertainty of the anomalous magnetic moment of the muon value in the Standard Model.

As we mentioned in the previous section, the $O(\alpha_s)$ (two-loop) correction to the massless quark loop has been performed and it has yielded a $\sim$10% correction. This suggests that a three-loop correction would probably have a size comparable to that of some non-perturbative contributions, whose magnetic susceptibilities $X_i$ have not been

rigorously computed yet. Consequently, a more detailed study of these from lattice groups is necessary.

The extension of the background OPE formalism to the mixed-virtuality high-energy regime of the HLbL amplitude has recently been performed for the first time, which provides an alternative to systematically expand upon the knowledge that we had on this regime [117]. The case when the lowest virtual momentum is within the perturbative regime of QCD has already been computed at the leading order, but results for perturbative corrections have not been published yet. The complementary case in which the lowest virtual momentum is below the perturbative threshold presents added complexities due to the lack of knowledge on the resulting non-perturbative matrix elements. This creates an opportunity for numerical studies from a lattice. Ultimately, improvements on SDC coming from the mixed-virtuality regime can help to lower the uncertainty from the lower-energy contributions to $a_\mu^{\text{HLbL}}$ or contribute to a better estimation of axial vector contributions, which continue to be a rather large source of uncertainty.

**Author Contributions:** Conceptualization, D.M., E.R. and R.F.; methodology, D.M., E.R. and R.F.; software, D.M., E.R. and R.F.; validation, D.M., E.R. and R.F.; formal analysis, D.M., E.R. and R.F.; resources, D.M., E.R. and R.F.; writing—original draft preparation, D.M., E.R. and R.F.; writing— review and editing, D.M., E.R. and R.F.; visualization, D.M., E.R. and R.F. All authors have read and agreed to the published version of the manuscript.

**Funding:** This research received no external funding.

**Data Availability Statement:** https://github.com/DanielMelo2000/QuarkLoopCode (accessed on 1 April 2024).

**Conflicts of Interest:** The authors declare no conflicts of interest.

## Appendix A. Mellin–Barnes Integrals, Multivariate Residues, and Hypergeometric Functions

The Davydychev tensor decomposition, which has the benefit of not introducing additional kinematic singularities, comes at the cost of introducing scalar integrals in shifted dimensions. In (97) and (98), we have arrived at a representation for the emerging scalar integrals in terms of Mellin–Barnes representation. Analytical expressions to these are often given in terms of hypergeometric-like series (The presence of polygamma functions means they are not hypergeometric, although their convergence analysis is the same) in one or more variables; therefore, they can give us a systematic expansion of the quark mass effects on the loop. In this appendix, we present a general framework of computation for nested Mellin–Barnes integrals.

### Appendix A.1. General Properties of Mellin–Barnes Integrals

In particular, we have to deal with $P$-fold Mellin–Barnes integrals of the form:

$$J(\{e_j\}, f; \{g_j\}, h; u_1, ..., u_P) = \int_{-i\infty}^{+i\infty} ... \int_{-i\infty}^{+i\infty} \prod_i^P \left\{ \frac{ds_i}{2\pi i} (-u_i)^{s_i} \right\} \frac{\prod_{j=1}^k \Gamma(e_j \cdot s + g_j)}{\Gamma(f \cdot s + h)} , \quad (A1)$$

where $s$ is a $P$-dimensional complex vector containing the integration variables, $e_j$ and $f$ are $P$-dimensional real vectors, $g_j$ and $h$ are real numbers, and $u_i$ is a complex number. Looking at (97) and (98), we have $f = (2, ..., 2)^T$ and $h = \sum_i v_i$, and that is a general feature of scalar loops. The vectors $e_j$ and the numbers $g_j$ do not have a general form, but can be easily read from the integrand in each case. The integral paths are shifted from the origin by a finite real quantity $\gamma_i$ to prevent them from splitting the poles of a Gamma function in the numerator into subsets or passing through one of them. If one is computing the integral in dimensional regularization, the former purpose might not be compatible with the limit $\epsilon \to 0$. We do not consider this situation here, as it is not relevant for this work. Instead, we refer the reader to the comprehensive study carried out in [120]. In general, the Gamma

functions in both the numerator and denominator of the integrand may also appear with powers higher than one and there may be multiple gamma functions in the denominator, but we will not consider such cases, as they do not happen in scalar loops.

There are two quantities upon which some important features of the integral in (A1) depend:

$$\Delta \equiv \sum_i e_i - f \qquad \alpha \equiv \text{Min}_{||\hat{y}||=1}\left\{ \sum_i |e_i \cdot \hat{y}| - |f \cdot \hat{y}| \right\},$$
(A2)

where $|.|$ symbolizes complex norm, and $||.||$ represents Euclidean vector norm. In particular, for all integrals of the type (94), we have $\Delta = 0$. The asymptotic behavior of the integrand is of course key for Mellin–Barnes integrals, and these two quantities characterize it. First, let us see the meaning of $\alpha$. For this, let us consider the asymptotic behavior of the integrand in (A1) when the imaginary part of $s_i$ grows big:

$$\Gamma(r + i\tau) \longrightarrow \sqrt{2\pi}|\tau|^{r-1/2}e^{-\pi|\tau|/2} \qquad \text{for} \quad |\tau| \to \infty .$$
(A3)

Then evaluating the complex norm of the integrand of (A1) in the asymptotic regime $s_i = \lim_{|R_i| \to \infty} \gamma_i - x_i + iR_i$, where $x_i$ and $R_i$ are real numbers, and $\gamma_i$ represents the real shift to the integration paths, we obtain:

$$\left| \prod_i^P \left\{ (-u_i)^{s_i} \right\} \frac{\prod_{j=1}^k \Gamma(e_j \cdot s + g_j)}{\Gamma(f \cdot s + h)} \right| \longrightarrow \prod_i^P \left\{ |u_i|^{\gamma_i} \right\} \frac{\prod_{j=1}^k |e_j \cdot R|^{\sum_i e_i \cdot (\gamma - x) + g_j - 1/2}}{|f \cdot R|^{f \cdot (\gamma - x) + h - 1/2}}$$
$$\times \exp\left\{ -\left( arg\{u_i\} + \pi \right) R_i \right.$$
$$\left. -\left( \sum_j |e_j \cdot R| - |f \cdot R| \right) \frac{\pi}{2} \right\} .$$
(A4)

The first line on the right-hand side is a polynomial in $R_i$, while the other two are exponential. Thus, we see that the integral in (A1) is absolutely convergent for

$$-arg\{-u\} \cdot R < \left( \sum_j |e_j \cdot R| - |f \cdot R| \right) \frac{\pi}{2} ,$$
(A5)

where $arg\{u_i\}$ is the argument of the complex variable $u_i$ and the components of $arg\{-u\}$ are equal to $arg\{u_i\} + \pi$. Since the inequality (A5) is homogeneous in $R$, then it can be simplified as

$$\text{Max}_{||\hat{y}||=1}|arg\{-u\} \cdot \hat{y}| < \alpha \frac{\pi}{2} .$$
(A6)

Finally, using the well-known Cauchy–Schwartz (also known as Cauchy–Bunyakovsky or Cauchy–Bunyakovsky–Schwartz inequality) inequality, one concludes that

$$\text{Max}_{||\hat{y}||=1}|arg\{-u\} \cdot \hat{y}| = ||arg\{-u\}|| \implies ||arg\{-u\}|| < \alpha \frac{\pi}{2} .$$
(A7)

Therefore, one sees that $\alpha$ characterizes the convergence regions of the Mellin–Barnes integral in (A1). For scalar loops, one has $\alpha > \sum_j |(\hat{y})_j| - |\sum_j (\hat{y})_j| > 0$; hence, there is always a non-trivial region of convergence.

While $\alpha$ is related to the convergence of the integral as a function of $u$, that is, the asymptotic behavior of the integrand in imaginary directions, $\Delta$ does the same with respect to the real part of the integration variables $s$. This is key for knowing the direction to which the contours of integration can be closed. To justify this interpretation, we follow a procedure analogous to that of $\alpha$, although this time, the Stirling formula is specialized to the case of a big real part:

$$|\Gamma(r + i\tau)| \longrightarrow \sqrt{2\pi}|r|^{r-1/2}e^{-r} .$$
(A8)

With such a formula, we study the integrand in the limit $s_i = \lim_{|x_i| \to \infty} \gamma_i - x_i + iR_i$:

$$
\left| \prod_i^P \left\{ (-u_i)^{s_i} \right\} \frac{\prod_{j=1}^k \Gamma(\boldsymbol{e}_j \cdot \boldsymbol{s} + g_j)}{\Gamma(\boldsymbol{f} \cdot \boldsymbol{s} + h)} \right| \longrightarrow \exp \left\{ - \left( \sum_j \boldsymbol{e}_j - \boldsymbol{f} \right) \cdot (\boldsymbol{\gamma} - \boldsymbol{x}) \right\}
$$
$$
\times \left| \prod_i^P \left\{ |u_i|^{-x_i} \right\} \frac{\prod_{j=1}^k |\boldsymbol{e}_j \cdot \boldsymbol{x}|^{\boldsymbol{e}_j \cdot \boldsymbol{x} - 1/2}}{|\boldsymbol{f} \cdot \boldsymbol{x}|^{\boldsymbol{f} \cdot \boldsymbol{x} - 1/2}} \right| ,
$$
(A9)

where $\boldsymbol{x}$ characterizes the direction to which the contour of integration closes, and thus, we see that for $\boldsymbol{\Delta} \neq 0$, there are preferred directions in the complex plane. Instead, when $\boldsymbol{\Delta} = 0$, there are many (infinitely many, as we will discuss later) regions where the integrand decreases depending on the values of $|u_i|$, and as such, there are multiple series representations which, if $\alpha > 0$, are analytic continuations of one another [121].

    Let us introduce useful definitions to shed more light on the meaning of $\Delta$, which is crucial for the computation of Mellin–Barnes integrals. We have found that the exponential increase or decrease in the Mellin–Barnes integrand in infinite real directions of the complex space $\mathbb{C}^P$ depends on a scalar product with $\boldsymbol{\Delta}$. More specifically, we conclude that the integrand increases exponentially for any $\boldsymbol{s} \in \mathbb{C}^P$ with a large real part such that $\boldsymbol{\Delta} \cdot Re\{\boldsymbol{s}\} > \boldsymbol{\Delta} \cdot \boldsymbol{\gamma}$ and the converse statement is valid if $\boldsymbol{\Delta} \cdot Re\{\boldsymbol{s}\} < \boldsymbol{\Delta} \cdot \boldsymbol{\gamma}$. We will later see that one can compute Mellin–Barnes integrals by closing a multivariable infinite "contour" in the region in which the integrand vanishes asymptotically, as one can expect from a naive multivariate generalization of Jordan's lemma. Consequently, we now introduce a definition that will come in handy below. Let $l_{\boldsymbol{\Delta}}$ be a hyperplane in the subspace $\mathbb{R}^P$ with normal vector $\boldsymbol{\Delta}$ whose points are defined by the condition $\boldsymbol{\Delta} \cdot Re\{\boldsymbol{s}\} = \boldsymbol{\Delta} \cdot \boldsymbol{\gamma}$. Note that $l_{\boldsymbol{\Delta}}$ constitutes a critical region of the asymptotic behavior of the Mellin–Barnes integrand. Let $\pi_{\boldsymbol{\Delta}}$ represent the "half" of $\mathbb{R}^P$ for which $\boldsymbol{\Delta} \cdot Re\{\boldsymbol{s}\} < \boldsymbol{\Delta} \cdot \boldsymbol{\gamma}$, which is the region of exponential decrease in the integrand. $\pi_{\boldsymbol{\Delta}}$ can be regarded as the real projection of a section $\Pi_{\boldsymbol{\Delta}}$ of $\mathbb{C}^P$. Since $\boldsymbol{\Delta}$ is a real vector, then such a section can be defined as a direct product: $\Pi_{\boldsymbol{\Delta}} \equiv \pi_{\boldsymbol{\Delta}} + i\mathbb{R}^P$. For $P = 1$, $l_{\boldsymbol{\Delta}}$ and $\pi_{\boldsymbol{\Delta}}$ are a point and a line, for $P = 2$, they are a line and a plane, and for $P = 3$, they are a plane and a 3D cube, respectively. The points of $\Pi_{\boldsymbol{\Delta}}$ are characterized by the condition $Re\{\boldsymbol{\Delta} \cdot \boldsymbol{s}\} < \boldsymbol{\Delta} \cdot \boldsymbol{\gamma}$; therefore, as we just discussed, it should be expected for the integrand poles that belong to $\Pi_{\boldsymbol{\Delta}}$ to play a major role in the computation of Mellin–Barnes integrals.

*Appendix A.2. Multivariate Generalization of Jordan's Lemma for Mellin–Barnes Integrals*

    Let us now consider the actual computation of Mellin–Barnes integrals. In univariate residues, we have the well-known Jordan's lemma:

$$
\frac{1}{2\pi i} \int_{-\infty}^{+\infty} dx \, f(x) e^{i\lambda x} = \sum_{a \in S} Res_a f(z) ,
$$
(A10)

where $\lambda > 0$ and $S$ is the set of poles of $f(z)$ in the upper half of the complex plane. This formula is valid if $\lim_{|z| \to \infty} |f(z)| = 0$ for $z$ in that region. Note that Jordan's lemma is usually taken to be the result regarding the vanishing of the integral of a complex variable function along an infinite semicircle, of which (A10) is a famous application, but here we adhere to the convention of [122]. If $\lambda < 0$, then the upper and lower halves of the complex plane change roles. This formula is only valid for one-dimensional Mellin–Barnes integrals, that is, the self-energy ones. It can in principle be applied also for multiple integrals as long as the location of the poles remains univariate. An example of such a situation would be a twofold Mellin–Barnes integral, such that in the numerator there are two gamma functions $\Gamma(z_1)\Gamma(z_2)$, and a counterexample would be $\Gamma(z_1)\Gamma(z_2 + z_1)$. In the latter case, the poles become entangled and it is necessary to use multivariate residue machinery. It is evident that we face such situations with (97) and (98).

It is possible to compute integrals (A1) in the general multivariate case with a formula analogous to (A10). Such a formula is of course more abstract, so, before presenting the result, let us first point to certain features of the univariate formula that should be translated into the multivariate case. The basic idea behind (A10) is to use the straight path of integration of the original integral as a part of a larger closed contour. The integral along such a contour can be computed with residues. The region to which the contour is closed is chosen such that contribution from the part of the contour that is additional to the original straight path vanishes. Since the original integration path is infinite and the contour is closed, then the additional parts are infinite too and must be placed in a region where the integrand vanishes, at least asymptotically. Such a region is ultimately determined by $\lambda$ in the univariate case and by $\mathbf{\Delta}$ for the multivariate ones of (A1). Therefore, one would expect the relevant poles of the multivariate case to be the ones in $\Pi_{\mathbf{\Delta}}$, just as the relevant ones for (A10) are in the upper half of $\mathbb{C}$ for $\lambda > 0$.

Now we need to introduce the definition of multivariate poles and residues. These are slightly different from the univariate case. Let us consider the following general function:

$$f(z) = \frac{\eta(z)}{\phi_1(z)...\phi_n(z)} \, , \tag{A11}$$

where $z = (z_1, ..., z_n) \in \mathbb{C}^n$. A naive univariate generalization would tell us that $f(z)$ has poles in any $z_0$ such that $\phi_j(z_0) = 0$ for at least one $j \in \{1, ..., n\}$, as long as $\eta(z_0) \neq 0$. Instead, the correct definition states that $f(z)$ has poles in any $z_0$ such that $\boldsymbol{\phi}(z_0) = (\phi_1(z_0), ..., \phi_n(z_0)) = 0$, as long as $\eta(z_0) \neq 0$. This definition is not as odd as it may seem; if we could arrange a variable change such that each $\phi_j$ becomes univariate, then we would disentangle the multivariate poles and the closed integral of $f(z)$ would become a product of univariate integrals. For some integration contours, such a product would be equal to zero if not all $\phi_j$ had zeros at the same point.

There is one more rather peculiar feature of the definition of poles that we have just given; it leaves space for ambiguities with respect to the way in which singular factors $\phi_j$ are grouped together. For example, let us consider the following function $f(z_1, z_2)$:

$$f(z_1, z_2) = \frac{\eta(z_1, z_2)}{z_1(z_1 - z_2 + 1)(z_1 + z_2)} \, . \tag{A12}$$

There is no obvious way to define the singular functions. Three of the possibilities are:

$$
\begin{aligned}
\phi_1 &= z_1(z_1 - z_2 + 1) \, , & \phi_2 &= (z_1 + z_2) \, , \\
\phi_1 &= (z_1 - z_2 + 1) \, , & \phi_2 &= z_1(z_1 + z_2) \, , \\
\phi_1 &= z_1 \, , & \phi_2 &= (z_1 - z_2 + 1)(z_1 + z_2) \, .
\end{aligned}
\tag{A13}
$$

Each of these three combinations has different poles and they may have even different residues in the poles that they share (An explicit computation of an example of the latter case is given in [111]). Furthermore, even if there were only two singular factors, the order in which they are defined introduces a sign ambiguity, as we will see later. Hence, any residue formula must clearly specify the singular functions with respect to which its poles are defined. Each set of singular points defined by the condition $\phi_j(z) = 0$ is called a divisor and we represent them with $F_j$. Consequently, the set $F_1 \cap F_2 \cap ... \cap F_n$ contains the poles of $f(z)$ with respect to a certain set of divisors $\{F_j\}$.

Now let us consider the residues of $f(z)$ in these poles:

$$Res_{\{F_1,...,F_n\},z_0} f(z) = \frac{1}{(2\pi i)^n} \oint_{C_\epsilon} \frac{\eta(z) dz_1...dz_n}{\phi_1(z)...\phi_n(z)} \, , \tag{A14}$$

where $C_\epsilon \{z \in \mathbb{C}^P | \, |\phi_i(z)| = \epsilon_i\}$ is called a cycle and $\epsilon_i$ has infinitesimal positive value. The orientation of the integration path $C_\epsilon$ is defined such that the change in the argument of

every $\phi_j$ is always possible, which is analogous to the usual clockwise orientation, although this time it refers to the functions $\phi_j$ rather than the integration variables $z_j$. Note that due to the definition of the orientation of $C_\epsilon$, one sees that residues are skew-symmetric with respect to the permutations of $\phi_j$. Equation (A14) defines local Grothendieck residues, which are a multivariate generalization of the univariate ones and are commonly used in the context of algebraic geometry [123].

Now we are able to state the the formal mathematical generalization of (A10) for multiple variables, which is called "multidimensional abstract Jordan lemma" [122]. It asserts that for a complex variable function $f(\boldsymbol{z})$:

$$\frac{1}{(2\pi i)^n} \int_\sigma f(\boldsymbol{z}) dz_1 ... dz_n = \sum_{a \in \Pi} Res_a f(\boldsymbol{z}) . \tag{A15}$$

Let $\Pi$ be a polyhedron and $\sigma$ be the "skeleton" of $\Pi$, that is, the structure formed by the vertices and edges of $\Pi$. The residues in $\Pi$ are defined in terms of divisors $\{F_j\}$ such that each of them does not intersect one specific face of the polyhedron, that is, the polyhedron has $n$ faces $\sigma_n$ and the set of divisors verifies the condition $F_j \cap \sigma_j = \varnothing$ for each $j = 1, ..., n$. This is referred to as "compatibility" between divisors and the polyhedron.

In general, $\Pi$ may be bounded or not; however, we want to identify the edges in $\sigma$ with the infinite straight integration paths of (A1), so we are interested in the unbounded case. In this context, there is an additional condition for the validity of (A15), which is essentially a multivariate generalization of the asymptotic behavior condition on $f(z)$ when there is an infinite set of poles, which we omitted when discussing (A10) and we omit for this case, too, because it is not crucial for our analysis [121,122].

Applying (A15) to integrals of the type shown in (A1), one obtains the following result [121,124]:

$$J(\{\boldsymbol{e}_j\}, \boldsymbol{f}; \{\boldsymbol{g}_j\}, h; u_1, ..., u_P) = \sum_{a \in \Pi_\Delta} Res_a J . \tag{A16}$$

In addition, $Res_a J$ represents the residue of the integrand in its pole $a$. The compatibility condition for the divisors and the polyhedron is of course still required for (A16) to be valid. For $\boldsymbol{\Delta} = 0$, one sees that there is no preferred region of the $\mathbb{C}^P$ space, and hence, such integrals are usually called "degenerate". In fact, in such cases, Formula (A16) remains valid for any $\Pi_\Delta$.

The analogy of this result with the standard one-dimensional Jordan lemma is more apparent in the one-dimensional case of (A1). In there, $l_\Delta$ is just $\gamma + i\mathbb{R}$. Hence, when $\Delta > 0$, the sum of residues from the poles enclosed in the negative real half of the complex plane constitutes a series representation convergent for any value of $u$, while the sum of residues from the other half forms a divergent asymptotic expansion [125]. For $\Delta < 0$, the roles of these two halves of the complex plane are inverted, while for $\Delta = 0$, one obtains two different series for each half that converge in non-overlapping complementary regions of the $u$ complex plane. If $\alpha > 0$, then they are an analytical continuation of each other. In this way, one can see the analogy of $\boldsymbol{\Delta}$ with the role of the time coordinate and its sign in Fourier transforms. Regarding the compatibility between the divisor and the polyhedron of integration, note that the face of $\Pi_\Delta$ is just the integration path $\gamma + i\mathbb{R}$; therefore, one sees that such a prescription is just the multidimensional generalization of the requirement for (A10) that no poles lie on the integration path.

Now that we have presented the multivariate generalization of Jordan's lemma, we need to show how to compute Grothendieck residues of the integrand of (A1) with respect to the poles and divisors that fulfill the requirements of (A16).

Let us first start with poles. The ones that we are interested in exist at points where $P$ gamma functions become singular, that is, the intersection of $P$ singular hyperplanes of the gamma functions in the numerator. For example, in the case of the three-point function (98), we must have an intersection of three two-dimensional planes. Each gamma

function in the numerator of the integrand generates a family $L^j$ with countably infinite singular hyperplanes $L_n^j$ defined as $L_n^j = \{s \in \mathbb{C}^P \,|\, e_j \cdot s + g_j = -n\}$ for every $n \in \mathbb{N}$. One sees that each $e_j$ is the normal vector of the family of singular hyperplanes of a given gamma function. They give us information about intersection of singular planes, and therefore, they are key to identifying poles of the integrand. If a set of $P$ vectors $\{e_{j_1}, ..., e_{j_P}\}$ is linearly independent, then for any $n_i \in \mathbb{N}$, the set $L_{n_1}^{j_1} \cap ... \cap L_{n_P}^{j_P}$ always has only one element $z_0 \in \mathbb{C}^P$, which constitutes a pole of the Mellin–Barnes integrand. Moreover, if each singular plane $L^{j_i}$ belongs to a different divisor $F_{j_i}$, then $z_0$ is a relevant pole for (A16). With this definition of poles, the formula (A16) requires us to:

- Group the singular planes of the gamma functions in the numerator of (A1) in $P$ divisors $F_j$ that satisfy the compatibility condition with respect to the faces of $\Pi_\Delta$.
- Study all possible $P$ combinations of gamma functions in the numerator of (A1) such that each gamma function belongs to a different divisor $F_j$.
- Determine which of these combinations have isolated intersection points, that is, poles.
- Discard all poles that do not belong to $\Pi_\Delta$.
- Compute the residues of the integrand of (A1) for all relevant poles.

In addition, there are situations in which things are more complicated. It is possible, and in fact it happens for the three-point function, that more than $P$ singular hyperplanes coincide at certain points. These cases are the multivariate versions of higher-multiplicity poles and they are called "resonant" or "logarithmic" due to the logarithms that appear in the resulting series because of the derivatives of the terms $(-u)^s$ that are involved. Later in the section, we present a useful tool to deal with such cases.

There is another subtlety that we have not addressed. The half space $\Pi_\Delta$ plays a key role in the computation, but it seems to be ill-defined for $\Delta$, which is actually true for all the integrals that we need. The solution to this issue is very simple; one may define $\Pi_\Delta$ arbitrarily. However, the key point for (A15) and (A16) is that one computes an integral along the skeleton of a polyhedron in terms of the poles that lie within the polyhedron. Hence, the polyhedron $\Pi$ that one chooses for the computation must have $\gamma$ as one of its vertices and $\gamma + i\mathbb{R}^P$ as one of its edges. For a given $\gamma$, there are still infinitely many options to define $\Pi_\Delta$. Nevertheless, there are still only a finite number of series representations for (A1) that, since $\alpha > 0$, are analytic continuations of each other for different values of $|u_i|$. Once the residues have been computed and the corresponding series representation has been obtained, one can identify the convergence region of the series obtained by applying Horn's theorem [84,113,115]. It is even possible to determine the convergence region of a series before performing the full computation [112] in order to compute only the series representation that converges for the kinematic regime that one is interested in.

Now that we have studied the poles that we need to compute (A1), we have only left to consider how to compute the residues on the right-hand side of (A16). As happens in the single-variable case, the formal definition (A14) usually is not the most appropriate tool.

Let us begin with the simple case in which there is a straightforward connection between univariate residues and multivariate ones. For this, let us consider again the general function $f(z)$ of (A11). If the Jacobian determinant evaluated at the pole $z_0$:

$$det\left(\frac{\partial \phi_j}{\partial z_i}\right)\Big|_{z=z_0} \tag{A17}$$

is not equal to zero, then one can perform the variable change $w_i \equiv \phi_i$, which disentangles the poles and hence allows for the multivariate integral to become a product of univariate integrals. The latter can be evaluated by the usual methods. These are called "simple poles" [126]. As we mentioned previously, this is usually not the case for (A1).

When the Jacobian (A17) is zero, then one has to use another formula called the "Transformation law" for multivariate residues (see page 20 of [126]), which is valid for residues of any function $f(z)$ irrespective of the value of its Jacobian determinant. For a function $f(z)$ with an isolated pole at $z = z_0$, one has:

$$Res_{z_0} \frac{\eta(z)}{\phi_1(z)...\phi_n(z)} = Res_{z_0} \frac{\eta(z)det\hat{A}}{\rho_1(z)...\rho_n(z)} \, , \tag{A18}$$

such that:

$$\rho_i(z) = \sum_j a_{ij}(z)\phi_j(z) \qquad \longrightarrow \qquad \boldsymbol{\rho}(z) = \hat{A}(z)\boldsymbol{\phi}(z) \, , \tag{A19}$$

where the coefficients $a_{ij}(z)$ are holomorphic functions that form the matrix $\hat{A}$ and $\boldsymbol{\rho} = (\rho_1, ..., \rho_n)$. The holomorphy condition for these matrix elements is important to ensure that they do not cancel zeros in any $\phi_j$. Another requisite for (A18) to hold is that all the poles of $\boldsymbol{\rho}$ and $\boldsymbol{\phi}$ are isolated (In the mathematical literature, this result is often presented in terms of ideals noted as $\langle \phi_1, ..., \phi_n \rangle$ and $\langle \rho_1, ..., \rho_n \rangle$. The condition of isolation for the poles is equivalent to the assertion that these two are *zero-dimensional* ideals.) The transformation law is useful to compute multivariate residues as long as one is able to find a set $\{\rho_j\}$ such that each element is a univariate function, because then one may factorize the integrals and use the standard univariate machinery for residue computation. From this formula, it is also easy to see that even a change in the order of the denominators $\phi_j$ introduces a minus sign from $A$, which illustrates the importance of properly taking into account the orientation of the cycles in multivariate integrals.

## Appendix B. Triangle Scalar Loop Integrals in Arbitrary Dimensions

In Equation (104), we quoted the result of [114] for scalar triangle loop integrals in arbitrary space-time dimensions with unit propagator powers, $J_3^{(d)}$. In this appendix, we complete the definition of relevant quantities that we used and present the results in a way that clearly shows the appearance of logarithms and ultraviolet singularities.

First, let us define the Cayley determinant $S_3$:

$$S_3 = \begin{vmatrix} 2m_1^2 & -p_1^2 + m_1^2 + m_2^2 & -p_3^2 + m_1^2 + m_3^2 \\ -p_1^2 + m_1^2 + m_2^2 & 2m_2^2 & -p_2^2 + m_2^2 + m_3^2 \\ -p_3^2 + m_1^2 + m_3^2 & -p_2^2 + m_2^2 + m_3^2 & 2m_3^2 \end{vmatrix} . \tag{A20}$$

In the same fashion, we define the Cayley determinants $S_{ij}$ of self-energy integrals, which are obtained by suppressing one of the three propagators in the triangle:

$$S_{ij} = \begin{vmatrix} 2m_i^2 & -p_i^2 + m_i^2 + m_j^2 \\ -p_i^2 + m_i^2 + m_j^2 & 2m_j^2 \end{vmatrix} = -\lambda(p_i^2, m_i^2, m_j^2) \, . \tag{A21}$$

Similarly, for the Gram determinants, we have:

$$G_3 = -8 \begin{vmatrix} p_1^2 & p_1 \cdot p_2 \\ p_1 \cdot p_2 & p_2^2 \end{vmatrix} = 2\lambda(p_1^2, p_2^2, p_3^2) \, ,$$
$$G_{12} = -4p_1^2 \, , \qquad G_{13} = -4p_3^2 \, , \qquad G_{23} = -4p_2^2 \, , \tag{A22}$$

where $p_3 = -p_1 - p_2$. Finally, we have $M_3 = S_3/G_3$ and $M_{ij} = S_{ij}/S_3$.

Now let us define $x_k$:

$$x_1 = 1 - \frac{D - E\beta + 2(C - B\beta)}{2(1 - \beta)(C - B\beta)} \, , \qquad x_2 = 1 + \frac{D - E\beta}{2(C - B\beta)} \, , \qquad x_3 = -\frac{D - E\beta}{2\beta(C - B\beta)} \, , \tag{A23}$$

where

$$A = p_1^2 \, , \qquad B = p_3^2 \, , \qquad C = -p_1 \cdot p_3 \, , \qquad D = -(p_1^2 + m_1^2 - m_2^2) \, ,$$
$$E = -(p_3^2 + m_1^2 - m_3^2) \, , \qquad F = m_1^2 \, , \qquad \beta = \frac{C + \sqrt{C^2 - AB}}{B} \, . \tag{A24}$$

$x_{ij}$ and $x_k$ fulfill the following relevant identities:

$$p_i^2 x_{ij}^2 = m_i^2 - M_{ij}, \qquad p_i^2 (x_k - x_{ij})^2 = M_3 - M_{ij}. \tag{A25}$$

Finally, we will present the formulas for the case $d = 4 + 2k - 2\epsilon$ with $k \in \mathbb{N}$, which are relevant for our computation. Keeping full $\epsilon$ dependence, we have:

$$
\begin{aligned}
J_3^{(4+2k-2\epsilon)} &\times \frac{(4\pi)^{2+k} \lambda^{1/2}(p_1^2, p_2^2, p_3^2)}{i(4\pi)^\epsilon} \\
&= \Gamma(-k+\epsilon)\left( M_3^k \left(\frac{\mu^2}{M_3}\right)^\epsilon - M_{ij}^k \left(\frac{\mu^2}{M_{ij}}\right)^\epsilon \right) \sum_{n_1=1} \frac{1}{n_1} \left( -\frac{x_{ij}}{x_k - x_{ij}} \right)^{n_1} \\
&\quad - M_{ij}^k \left(\frac{\mu^2}{M_{ij}}\right)^\epsilon \left[ \sum_{\substack{n_1=1 \\ n_2=1}}^{n_2=k} \frac{\Gamma(-k+n_2+\epsilon)}{(n_1+2n_2)n_2!} \left( -\frac{x_{ij}}{x_k - x_{ij}} \right)^{n_1} \left( -\frac{p_i^2 x_{ij}^2}{M_{ij}} \right)^{n_2} \right. \\
&\quad \left. + \sum_{\substack{n_1=1 \\ n_2=k+1}} \frac{\Gamma(-k+n_2)}{(n_1+2n_2)n_2!} \left( -\frac{x_{ij}}{x_k - x_{ij}} \right)^{n_1} \left( -\frac{p_i^2 x_{ij}^2}{M_{ij}} \right)^{n_2} \right] \\
&\quad - \left\{ x_{ij} \to 1 - x_{ij} \; ; \; x_k \to 1 - x_k \right\}.
\end{aligned}
\tag{A26}
$$

Note that we neglected the infinitesimal term $i\epsilon$ that gives the Feynman prescription, because it is not relevant in the deep space-like region that we are interested in. Nevertheless, it can be easily reinstated by replacing $M_{ij} \to M_{ij} - i\epsilon$ and $M_3 \to M_3 - i\epsilon$. Taking the limit $\epsilon \to 0$, we have:

$$
\begin{aligned}
J_3^{(4+2k)} &\times \frac{(4\pi)^{2+k} \lambda_-^{1/2}}{i} \\
&= \frac{(-1)^k}{k!} \left\{ M_3^k \left( \frac{1}{\hat{\epsilon}} + \ln\left\{\frac{\mu^2}{M_3}\right\} \right) - M_{ij}^k \left( \frac{1}{\hat{\epsilon}} + \ln\left\{\frac{\mu^2}{M_{ij}}\right\} \right) \right\} \sum_{n_1=1} \frac{1}{n_1} \left( -\frac{x_{ij}}{x_k - x_{ij}} \right)^{n_1} \\
&\quad + \frac{(-1)^k}{k!} \sum_{j=1}^k \frac{1}{j} \left( M_3^k - M_{ij}^k \right) \sum_{n_1=1} \frac{1}{n_1} \left( -\frac{x_{ij}}{x_k - x_{ij}} \right)^{n_1} \\
&\quad - M_{ij}^k \left( \frac{1}{\hat{\epsilon}} + \ln\left\{\frac{\mu^2}{M_{ij}}\right\} \right) \left[ \sum_{\substack{n_1=1 \\ n_2=1}}^{n_2=k} \frac{(-1)^{k-n_2}}{(n_1+2n_2)(k-n_2)!n_2!} \left( -\frac{x_{ij}}{x_k - x_{ij}} \right)^{n_1} \left( -\frac{p_i^2 x_{ij}^2}{M_{ij}} \right)^{n_2} \right] \\
&\quad - M_{ij}^k \left[ \sum_{\substack{n_1=1 \\ n_2=1}}^{n_2=k} \frac{1}{(n_1+2n_2)n_2!} \left( -\frac{x_{ij}}{x_k - x_{ij}} \right)^{n_1} \left( -\frac{p_i^2 x_{ij}^2}{M_{ij}} \right)^{n_2} \frac{(-1)^{k-n_2}}{(k-n_2)!} \sum_{j=1}^{k-n_2} \frac{1}{j} \right. \\
&\quad \left. + \sum_{\substack{n_1=1 \\ n_2=k+1}} \frac{\Gamma(-k+n_2)}{(n_1+2n_2)n_2!} \left( -\frac{x_{ij}}{x_k - x_{ij}} \right)^{n_1} \left( -\frac{p_i^2 x_{ij}^2}{M_{ij}} \right)^{n_2} \right] \\
&\quad - \left\{ x_{ij} \to 1 - x_{ij} \; ; \; x_k \to 1 - x_k \right\},
\end{aligned}
\tag{A27}
$$

where a sum over the three permutations $(i,j,k) \to (1,2,3) \to (2,3,1) \to (3,1,2)$ is implied. $\mu$ is the renormalization scale. The singular terms $1/\hat{\epsilon}$ vanish and dependence on $\mu$ disappears when powers of the propagators grow high enough via derivatives with respect to the masses.

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
