# Peer review of "Hadronic Light-by-Light Corrections to the Muon Anomalous Magnetic Moment"

_2571-712X, doi:10.3390/particles7020020_

Round 1

Reviewer 1 Report

Comments and Suggestions for Authors

The manuscript is about calculating the hadronic light by light (HLbL) contribution to the muon anomalous magnetic moment in the short-distance regime.

The abstract claims it is a review of HLbL as well but that aspect is quite small and simply adds a few references compared to the content of Ref. 3. The introduction and section 3 is essentially a summary of Ref. 3.

Sections 4,5,6,7 are in essence a reproduction of the work in Refs. 55 and 56, providing a few more (in my opinion rather obvious) intermediate steps.

Section 8 uses an alternative method for the pure quark loop contribution (the fully analytic result was already obtained earlier in 55,56) and does the expansion in quark masses in a somewhat different manner than in Refs. 55-57.

The whole article is written at a rather basic level (appropriate for a master thesis but not for a scientific paper) with a large amount of details.

In conclusion the abstract is rather misleading as regards the content and it is certainly not clear on the fact that it basically is a recalculation of existing work. It is stated in the introduction and it becomes clear while reading the paper that there is no really new content but a recalculation of already known results. That the results of earlier work are reproduced is worth doing but it seems rather unnecessary to devote a 67 page paper to this.

Reviewer 2 Report

Comments and Suggestions for Authors

Referee Report

Journal Particles (ISSN 2571-712X)

Manuscript ID: particles-2746954

Type: Review

Title:  “Hadronic Light by Light Corrections to the Muon Anomalous Magnetic Moment”

Authors: Daniel Gerardo Melo Porras *  , Edilson Alfonso Reyes Rojas , Angelo Raffaele Fazio 

Special Issue: Feature Papers for Particles 2023

The manuscript presents a comprehensive and insightful review of the present status of  research on the Hadronic Light-by-Light corrections to the muon anomalous magnetic moment and  I believe it can  be useful to those  entering in this research field as well as  to those  already acquainted to the subject.

It appears timely, in in moment in which  there is   tension on the theory side,  particularly shaken by the apparent disagreement between the results on the Hadronic Vacuum Polarization (HVP) obtained in  lattice and data-driven dispersive approaches. This theoretical conundrum is particularly unwelcome, because it makes it difficult the  comparison with recent experimental data with their  increased precision.

The authors have skillfully synthesized a vast body of scientific literature concarning   another important hadronic contribution to the anomalous magnetic moment of the muon, i.e. the Hadronic Light-by Light (HLbL) contribution. Before the HVP crisis, HLbL was the hadronic contribution with the larger theorethical uncentainty. Indeed, it receives significant contributions from all energy scales, where different description of hadronic physics have to  used: chiral Lagrangian, narrow resonance exchanges, perturbative QCD, to say a few.

In recent years, the data-driven approach, using dispersion relation to consistently  account for  low energy data has become the leading approach in  making theoretical predictions for  HLbL. However,  the study of the role of short distance constraints  (SDC) from perturbative QCD has  gained momentum, and much effort has been devoted to their evaluation  in order to pinpoint the uncertainity of the theoretical predictions  of the HLbL contribution. 

The review focus is on the SDC coming from the maximally symmetric high-energy limit  in the quark loop approximation. The authors have evaluated the tensor decomposition of the HLbL amplitude, using the kinematic singularity free formalism introduced  in the data-driven approach. Using less assumptions, compared to those of  previous papers, they have provided an independent  check of the previous results.

The paper is well-structured, and a big  amount  of information is well explained, facilitatating a clear comprehension of the subject matter. The inclusion of tables, figures, and graphical representations enhances the overall readability and effectiveness of the paper. The existing literature is discussed with great detail, often spelling out some tricky technical  points, which the interested reader would  otherwise find only after a careful study of the existing literature.  The language used is clear, and as I have already said, it cater to both experts in the field and to those less familiar with its subject.

In conclusion, I recommend the acceptance of this review paper for publication  in its current form.

Author Response

The reviewer recommends the work for publication without revision, therefore we have no replies.

Reviewer 3 Report

Comments and Suggestions for Authors

The manuscript submitted by D. Melo, E. Reyes and R. Fazio is a review of the contribution from hadronic light-by-light (HLbL) to the anomalous magnetic moment og the muon. This subject has been an active field of research for many years now, and remains an important component in the evaluation of the the anomalous magnetic moment of the muon within the standard model.

The manuscript first reviews the present situation of this field of research and introduces the theoretical aspects pertaining to the HLbL contribution. The authors establish the link with the muon's Pauli form factor and describe how it can be projected out of the full vertex function. The description of the HLbL component of this form factor in terms of the fourth-rank vacuum polarization tensor is also introduced. The authors then describe the dispersive approach that has been devised and exploited in recent theoretical calculation, before moving to the discussion of the short-distance constraints that the fourth-rank vacuum-polarization tensor has to satisfy. The main focus of their study consists of a  particular class of constraints arising from the symmetric high-energy regime (all three virtual photon momenta become simultaneously large in the Euclidian domain) within the framework of background gauge fixing. The authors provide a detailed introduction to the technical tools needed for this study. In particular, they describe and implement a different method to handle tensor loop integrals, which allows to show explicitly that the resulting expression decomposes on the tensor basis introduced for the dispersive approach. For the reader interested in the technical details of the calculation of the short-distance properties this alternative presentation may be useful.

The review (and the list of references) is rather complete in the sense that everything that one expects to find in it seems to be there. I would therefore recommend its publication in ``Particles", but only conditionally. Indeed, there are a certain number of points and issues that require more careful formulations. They are listed hereafter:

1) I think that the statement in line 27 about the discrepancy between the experimental measurement of the anomalous magnetic moment of the electron and its theoretical prediction is misleading. The authors should specify which value of the fine-structure constant they refer to and why they do not consider a more recent measurement for which the central value of the discrepancy is only 4.8 (instead of 8.8) in the same units as given in the text. Also the uncertainty on this discrepancy should be given as well. As a general remark, numbers quoted without uncertainties have only a limited meaning (experimentalists would even say that they have no meaning), cf. lines 51 and 52.

2) The statement made in lines 72-73 is highly misleading. What has been confirmed so far is a part only of the total HVP contribution from the BMW collaboration, coming from the so-called intermediate window. This partial result, however, corresponds only to ~30% of the total value. The remaining 70% of the BMWc value have not yet been confirmed by another lattice collaboration.

3) On line 83 it should be recalled that the value given for the HLbL contribution is the one to be found in ref. [7].

4) The statement made on line 88 is not correct: in four dimensions, the number of independent tensors is 41, a number that can easily be understood upon counting the independent helicity amplitudes.

5) In the text covered by the lines 100-104 the authors seem to make a difference between dispersive approaches and hadronic models. Could they please explain why they put refs. [35,36,37,40] into the first class (this is how I read the text) and not into the second?

6) I do not understand how, from what they have written, the authors go from (26) to (28), especially since they mention ``differentiating the Ward identity" without, seemingly, having specified the Ward identity in question at this stage.

7) I do not understand the statement of the authors in line 292, namely that lattice computations of HLbL ``still are not competitive with dispersive ones". Only certain (dominant) contributions (pseudo-scalar poles,...) have been evaluated dispersively, whereas lattice calculations include, in principle, all the contributions. If one completes the dispersive contributions with the evaluation/estimate of the other ones, which is what leads to the number given in [7], then one finds that recent lattice results are quite competitive, at least as far as the total HLbL component is concerned.

8) Many references contain typos and/or are with incomplete bibliographical data (e.g. missing journal number or page number, or arXiv number). The content of the references definitely needs to be checked seriously.

9) On line 1941 I think that the replacement ``through" -> ``thorough" is required? There is a typo on line 762.

Comments on the Quality of English Language

Generally correct. A systematic search for typographical errors would be welcome.

Author Response

Please see the attachment. The cover letter refers to a revised version of the manuscript that we have submitted.

Round 2

Reviewer 1 Report

Comments and Suggestions for Authors

I stand by my previous comments, the authors clearly disagree. The background field method is sufficiently well known in my opinion that having all intermediate steps carefully given is not necessary. The same goes for a Mellin-Barnes evaluation of integrals which I would have simply relegated to an appendix. One-loop integrals and methods are sufficiently well known they do not need extensive  explanations. In point 3, it should be remembered that the higher order in the quark masses depend on the definitions of the operators used in the nonperturbative part, as explained in [56], thus having the quark loop result by itself to all orders is not very relevant (and numerically negligible anyway). For the review parts, the changes compared to the big white paper are rather minimal and have been given in many places already.
